# Characterization of Excess Risk for Locally Strongly Convex Population Risk

**Mingyang Yi**[1,2,3]*, **Ruoyu Wang**[1,2]*, **Zhi-Ming Ma**[1,2]
[1]University of Chinese Academy of Sciences
[2]Academy of Mathematics and Systems Science, Chinese Academy of Sciences
[3]Huawei Noah's Ark Lab
{yimingyang17, wangruoyu17}@mails.ucas.edu.cn
mazm@amt.ac.cn

## Abstract

We establish upper bounds for the expected excess risk of models trained by proper iterative algorithms which approximate the local minima. Unlike the results built upon the strong globally strongly convexity or global growth conditions e.g., PL-inequality, we only require the population risk to be *locally* strongly convex around its local minima. Concretely, our bound under convex problems is of order $\tilde{\mathcal{O}}(1/n)$. For non-convex problems with $d$ model parameters such that $d/n$ is smaller than a threshold independent of $n$, the order of $\tilde{\mathcal{O}}(1/n)$ can be maintained if the empirical risk has no spurious local minima with high probability. Moreover, the bound for non-convex problem becomes $\tilde{\mathcal{O}}(1/\sqrt{n})$ without such assumption. Our results are derived via algorithmic stability and characterization of the empirical risk's landscape. Compared with the existing algorithmic stability based results, our bounds are dimensional insensitive and without restrictions on the algorithm's implementation, learning rate, and the number of iterations. Our bounds underscore that with locally strongly convex population risk, the models trained by any proper iterative algorithm can generalize well, even for non-convex problems, and $d$ is large.

## 1 Introduction

The core problem in machine learning is obtaining a model that generalizes well on unseen test data. The excess risk decides the model's performance on these unseen data, and it can be decomposed into optimization and generalization errors. The tool of algorithmic stability [9, 10] has been proven to be a suitable tool for exploring the excess risk. Roughly speaking, the output of a stable algorithm is robust to a slight change in the algorithm's input, i.e., training set. The output of a stable algorithm has been proved to have controlled excess risk in [9], and the result has been further developed under some specific algorithms [32, 79, 14, 16, 50, 20] e.g., stochastic gradient descent [61] (SGD). However, these results have some limitations. The results in [79, 14, 50, 47] are obtained under the assumption of either global strong convexity or global growth conditions (PL-inequality [42]). On the other hand, the results in [32, 20] are only applicable to a specific algorithm, i.e., SGD, and their bounds of generalization error diverge across training which is inconsistent with the observation that "train longer, generalize better" [34].

To improve these, we provide a unified analysis of the expected excess risk for a generic class of iterative algorithms without any strong global conditions, i.e., global strong convexity or global growth conditions in [79, 14, 50]. Concretely, we substitute the strong global conditions with weaker

---

*equal contribution

36th Conference on Neural Information Processing Systems (NeurIPS 2022).

local strong convexity (see Section 2) of population risk around its local minima. The substitution is based on the fact that the nice strong convexity property can be locally (though not globally) satisfied by many important problems, e.g., PCA [30], ICA [27], and matrix completion [28]. We derive our results via algorithmic stability and characterize the empirical risk's landscape. For both convex and non-convex problems, our results can be applied to any proper algorithms that approximate local minima. Moreover, our generalization upper bounds do not diverge with the number of training steps.

Technically, we upper bound both generalization and optimization errors to control the excess risk. We first show a fact that the locally strongly convexity around the local minima of population risk (population local minima) can be generalized to the local minima of empirical risk (empirical local minima), and the empirical local minima would concentrate around population local minima. Then for convex problems, we establish the generalization upper bound of the iterates of any proper algorithm via algorithmic stability by leveraging the facts of iterates will converge to empirical local minima, which concentrate around population local minima. For non-convex problems, our generalization error analysis includes three steps. 1) By applying similar arguments under the convex problem, we upper bound the generalization error of those empirical local minima around population local minima. 2) Then, we prove that, with high probability, there are no extra empirical local minima except for those concentrated around population local minima with guaranteed generalization capability. 3) Finally, we extrapolate the upper bound of the generalization error to the iterates obtained by the proper algorithm as they converge to empirical local minima.

After controlling the generalization error, the excess risk is directly implied by characterizing the optimization error. By the proved local strong convexity of empirical risk and the convergence results of proper algorithms, the optimization error can be controlled as in [12, 29, 64, 27, 37].

Concretely, we establish an upper bound of order $\tilde{\mathcal{O}}(1/n)$ ($\tilde{\mathcal{O}}(\cdot)$ defined in Section 2) for the expected excess risk of iterates obtained by any proper algorithm under convex problems. Here $n$ is the number of training samples. For non-convex problems with $d$ parameters of model, we establish an upper bound of order $\tilde{\mathcal{O}}(1/\sqrt{n} + \exp(-n(c_1 - d/n)))$ where $c_1$ is a constant independent of $n$ and $d$. Noticeably, the exponential term in the bound can be ignored when $d/n \leq c_1$, then our bound becomes $\tilde{\mathcal{O}}(1/\sqrt{n})$. The bound can be applied to high-dimensional problems such that $d$ is in the same order of $n$. The result significantly improves the classical one of order $\mathcal{O}(\sqrt{d/n})$ [63], which has polynomial dependence on $d$. Moreover, our bound of order $\tilde{\mathcal{O}}(1/\sqrt{n})$ can be improved to $\tilde{\mathcal{O}}(1/n)$ if the empirical risk has no spurious local minima with high probability, which can be satisfied for many important non-convex problems [30, 28, 1].

Our upper bounds to the excess risk underscore that, for both convex and non-convex problems satisfying our regularity conditions, the model trained by an algorithm can generalize on test data even when $d$ is large. Our improvements over existing classical results are summarized as follows.

• For convex problems, our bound improves the standard upper bound of the expected excess risk in the order of $\mathcal{O}(\sqrt{1/n})$ [32] to $\tilde{\mathcal{O}}(1/n)$, under an extra locally strongly convex assumption.

• For non-convex problems, we relax the dimensional-dependence in the standard excess risk bound of order $\mathcal{O}(\sqrt{d/n})$ [63], under local strong convexity assumption.

• In contrast to the existing algorithmic stability based works [32, 79, 14], our results can be applied to any algorithms that approximate local minima without restrictions on the implementation of algorithms, learning rate, and the number of iterations.

## 2  Preliminaries

### 2.1  Notations and Assumptions

In this subsection, we collect our (mostly standard) notations and assumptions. We use $\|\cdot\|$ to denote $\ell_2$-norm for vectors and spectral norm for matrices. $B_p(\boldsymbol{w}, r)$ is $\ell_p$-ball with radius $r$ around $\boldsymbol{w} \in \mathbb{R}^d$. Let dataset $\{\boldsymbol{z}_1, \cdots, \boldsymbol{z}_n, \boldsymbol{z}_1', \cdots, \boldsymbol{z}_n'\}$ be $2n$ i.i.d samples from an unknown distribution, and $\boldsymbol{S} = \{\boldsymbol{z}_1, \cdots, \boldsymbol{z}_n\}$ is the training set, $\boldsymbol{S}^i = \{\boldsymbol{z}_1, \cdots, \boldsymbol{z}_{i-1}, \boldsymbol{z}_i', \boldsymbol{z}_{i+1}, \cdots, \boldsymbol{z}_n\}$ and $\boldsymbol{S}' = \boldsymbol{S}^1$. Throughout this paper, we assume without further mention that the loss function $f(\boldsymbol{w}, \boldsymbol{z})$ is differentiable w.r.t. to parameter $\boldsymbol{w}$ for any $\boldsymbol{z}$, $0 \leq f(\boldsymbol{w}, \boldsymbol{z}) \leq M$, and the parameter space $\mathcal{W} \subseteq \mathbb{R}^d$

is a convex compact set. Thus $\|\boldsymbol{w}_1 - \boldsymbol{w}_2\| \leq D$ for $\boldsymbol{w}_1, \boldsymbol{w}_2 \in \mathcal{W}$ and some positive constant $D$. The population risk is $R(\boldsymbol{w}) = \mathbb{E}_{\boldsymbol{z}}[f(\boldsymbol{w}, \boldsymbol{z})]$ and its empirical counterpart on the training set $\boldsymbol{S}$ is $R_{\boldsymbol{S}}(\boldsymbol{w}) = n^{-1} \sum_{i=1}^{n} f(\boldsymbol{w}, \boldsymbol{z}_i)$. Let $\boldsymbol{w}_{\boldsymbol{S}}^* \in \arg\min_{\boldsymbol{w}} R_{\boldsymbol{S}}(\boldsymbol{w})$ and $\boldsymbol{w}^* \in \arg\min_{\boldsymbol{w}} R(\boldsymbol{w})$, The projection operator $\mathcal{P}_{\mathcal{W}}(\cdot)$ is defined as $\mathcal{P}_{\mathcal{W}}(\boldsymbol{v}) = \arg\min_{\boldsymbol{w} \in \mathcal{W}} \{\|\boldsymbol{w} - \boldsymbol{v}\|\}$. During our analysis, the order of sample size $n$ can go to infinity, and $d$ can diverge to infinity with $n$. But we assume the other quantities are universal constant independent of $n$. The symbol $\mathcal{O}(\cdot)$ is the order of a number, while $\tilde{\mathcal{O}}(\cdot)$ hides a poly-logarithmic factor in the number of model parameters $d$. The following two assumptions on loss function $f(\boldsymbol{w}, \boldsymbol{z})$ are imposed on the population risk.

**Assumption 1** (Smoothness). *For $0 \leq j \leq 2$, each $\boldsymbol{z}$ and any $\boldsymbol{w}_1, \boldsymbol{w}_2 \in \mathcal{W}$,*

$$\left\| \nabla^j f(\boldsymbol{w}_1, \boldsymbol{z}) - \nabla^j f(\boldsymbol{w}_2, \boldsymbol{z}) \right\| \leq L_j \|\boldsymbol{w}_1 - \boldsymbol{w}_2\|, \tag{1}$$

*where $\nabla^j f(\boldsymbol{w}, \boldsymbol{z})$ are respectively loss function, gradient, and Hessian at $\boldsymbol{w}$ for $j = 0, 1, 2$.*

**Assumption 2** (Non-Degenerate Local Minima). *For $\boldsymbol{w}_{\text{local}}^*$ in the set of local minima of population risk $R(\boldsymbol{w})$, $\nabla^2 R(\boldsymbol{w}_{\text{local}}^*) \succeq \lambda > 0$, i.e., $\nabla^2 R(\boldsymbol{w}_{\text{local}}^*) - \lambda \boldsymbol{I}_d$ is a semi-positive definite matrix.*

Assumption 1 says that the loss function should be smooth enough, which is a mild assumption and has been adopted in [32, 81, 30]. Assumption 1 and 2 together imply that the population risk is locally strongly convex around its local minima. The rationale behind the imposed local strong convexity is as follows. Though the strong global conditions (e.g., global strong convexity) in [32, 79, 14, 16, 50, 20] do not hold in many problems, the weaker locally strongly convex condition can be satisfied by many important problems, e.g., generalized linear regression [49], robust regression [49], PCA [30], ICA [27], and matrix completion [28]. The detailed examples of import problems that satisfy the assumptions imposed in this paper are in Appendix F.

## 2.2 Stability and Generalization

**Definition 1** (Proper Algorithm). *The algorithm $\mathcal{A}$ is proper if it approximates local minima* [2] *of empirical risk $R_{\boldsymbol{S}}(\boldsymbol{w})$.*

This is a rough definition of the discussed proper algorithm. The sense in which algorithms approximate local minima will be made clear in our formal theoretical results. Let $\mathcal{A}(\boldsymbol{S})$ be the parameters obtained by an algorithm $\mathcal{A}$, e.g., SGD, on the training set $\boldsymbol{S}$. The performance of model on unseen data is determined by the excess risk $R(\mathcal{A}(\boldsymbol{S})) - \inf_{\boldsymbol{w}} R(\boldsymbol{w})$, which is the gap of population risk between the current model and the optimal one. In this paper, we explore the expected excess risk $\mathbb{E}_{\mathcal{A}, \boldsymbol{S}}[R(\mathcal{A}(\boldsymbol{S})) - \inf_{\boldsymbol{w}} R(\boldsymbol{w})]$ where $\mathbb{E}_{\mathcal{A}, \boldsymbol{S}}[\cdot]$ means the expectation is taken over the randomized algorithm $\mathcal{A}$ and the training set $\boldsymbol{S}$. We may neglect the subscript if there is no obfuscation. Since $R_{\boldsymbol{S}}(\boldsymbol{w}_{\boldsymbol{S}}^*) \leq R_{\boldsymbol{S}}(\boldsymbol{w}^*)$, we have the following decomposition.

$$
\begin{aligned}
\mathbb{E}_{\mathcal{A}, \boldsymbol{S}}[R(\mathcal{A}(\boldsymbol{S})) - R(\boldsymbol{w}^*)] &= \mathbb{E}_{\mathcal{A}, \boldsymbol{S}}[R(\mathcal{A}(\boldsymbol{S})) - R_{\boldsymbol{S}}(\boldsymbol{w}^*)] \leq \mathbb{E}_{\mathcal{A}, \boldsymbol{S}}[R(\mathcal{A}(\boldsymbol{S})) - R_{\boldsymbol{S}}(\boldsymbol{w}_{\boldsymbol{S}}^*)] \\
&= \mathbb{E}_{\mathcal{A}, \boldsymbol{S}}[R_{\boldsymbol{S}}(\mathcal{A}(\boldsymbol{S})) - R_{\boldsymbol{S}}(\boldsymbol{w}_{\boldsymbol{S}}^*)] + \mathbb{E}_{\mathcal{A}, \boldsymbol{S}}[R(\mathcal{A}(\boldsymbol{S})) - R_{\boldsymbol{S}}(\mathcal{A}(\boldsymbol{S}))] \\
&\leq \underbrace{\mathbb{E}_{\mathcal{A}, \boldsymbol{S}}[R_{\boldsymbol{S}}(\mathcal{A}(\boldsymbol{S})) - R_{\boldsymbol{S}}(\boldsymbol{w}_{\boldsymbol{S}}^*)]}_{\mathcal{E}_{\text{opt}}} + \underbrace{|\mathbb{E}_{\mathcal{A}, \boldsymbol{S}}[R(\mathcal{A}(\boldsymbol{S})) - R_{\boldsymbol{S}}(\mathcal{A}(\boldsymbol{S}))]|}_{\mathcal{E}_{\text{gen}}}.
\end{aligned}
\tag{2}
$$

The expected excess risk is upper bounded by the sum of optimization error $\mathcal{E}_{\text{opt}}$ and generalization error $\mathcal{E}_{\text{gen}}$. $\mathcal{E}_{\text{opt}}$ is decided by the convergence rate of the algorithm $\mathcal{A}$ [12, 29]. The generalization error $\mathcal{E}_{\text{gen}}$ can be controlled by algorithmic stability [9] as follows.

**Definition 2.** *An algorithm $\mathcal{A}$ is $\epsilon$-uniformly stable, if*

$$\epsilon_{\text{stab}} = \mathbb{E}_{\boldsymbol{S}, \boldsymbol{S}'} \left[ \sup_{\boldsymbol{z}} |\mathbb{E}_{\mathcal{A}}[f(\mathcal{A}(\boldsymbol{S}), \boldsymbol{z}) - f(\mathcal{A}(\boldsymbol{S}'), \boldsymbol{z})]| \right] \leq \epsilon, \tag{3}$$

*where $\boldsymbol{S}$ and $\boldsymbol{S}'$ are defined at the beginning of Section 2.1.*

The $\epsilon$-uniformly stable is different from the one in [32], which does not take expectation over training sets $\boldsymbol{S}$ and $\boldsymbol{S}'$. The next theorem shows that the uniform stability implies the expected generalization of the model, i.e., $\mathcal{E}_{\text{gen}} \leq \epsilon_{\text{stab}}$. The idea of Theorem 1 is similar to the ones in [9, 32, 14], and its proof is in Appendix A.

---

[2]Please notice that local minima are all global minima for convex problem.

**Theorem 1.** *If $\mathcal{A}$ is $\epsilon$-uniformly stable, then*

$$\mathcal{E}_{\text{gen}} = |\mathbb{E}_{\mathcal{A},\boldsymbol{S}}\left[R(\mathcal{A}(\boldsymbol{S})) - R_{\boldsymbol{S}}(\mathcal{A}(\boldsymbol{S}))\right]| \leq \epsilon. \tag{4}$$

Please note that all the analysis in this paper is applicable to the practically infeasible empirical risk minimization "algorithm" such that $\mathcal{A}(\mathcal{S}) = \boldsymbol{w}_{\boldsymbol{S}}^*$. However, to make our results more practical, we suppose $\mathcal{A}$ as iterative algorithms in the sequel. For any given iterative algorithm $\mathcal{A}$, let $\boldsymbol{w}_t$ and $\boldsymbol{w}_t'$ denote the output of the algorithm when $\mathcal{A}$ is iterated $t$ steps on the training set $\boldsymbol{S}$ and $\boldsymbol{S}'$ respectively.

## 3 Excess Risk under Convex Problems

In this section, we propose upper bounds of the expected excess risk for convex problems. We impose the following convexity assumption throughout this section.

**Assumption 3** (Convexity). *For each $\boldsymbol{z}$ and any $\boldsymbol{w}_1, \boldsymbol{w}_2 \in \mathcal{W}$, $f(\boldsymbol{w}, \boldsymbol{z})$ satisfies*

$$f(\boldsymbol{w}_1, \boldsymbol{z}) - f(\boldsymbol{w}_2, \boldsymbol{z}) \leq \langle \nabla f(\boldsymbol{w}_1, \boldsymbol{z}), \boldsymbol{w}_1 - \boldsymbol{w}_2 \rangle. \tag{5}$$

### 3.1 Generalization Error under Convex Problems

As we have discussed, in the existing literature [32, 79, 14, 16, 50, 20], researchers have explored the excess risk via the algorithmic stability to control the error generalization. However, the obtained generalization upper bounds of order $\mathcal{O}(1/n)$ in [32, 79, 14, 50] are built upon the strong assumptions of either global strong convexity or global growth conditions, e.g., PL-inequality [42]. On the other hand, the generalization upper bounds in [32, 20] are only applied to SGD, and they diverge as the number of iterations grows. For example, Theorem 3.8 in [32] establishes an upper bound $2L_0^2 \sum_{k=0}^{t-1} \eta_k/n$ to the algorithmic stability of SGD with learning rate $\eta_k$, which diverges when $t \to \infty$, as the convergence of SGD requires $\sum_{k=0}^{\infty} \eta_k = \infty$ [8]. Thus the bound can not explain the observation that the generalization error of SGD trained model converges to a constant [8, 34].

To mitigate the drawbacks in the existing literature, we propose the following new upper bound of algorithmic stability (Theorem 2). Our bound can be applied on the top of any proper algorithm defined in Definition 1, and it remains small for an arbitrary number of iterations as long as the sample size $n$ is large. Under convexity Assumption 3, the proper algorithm means that $\mathbb{E}\left[R_{\boldsymbol{S}}(\boldsymbol{w}_t) - R_{\boldsymbol{S}}(\boldsymbol{w}_{\boldsymbol{S}}^*)\right] \to 0$ as $t \to \infty$. Our theorem is based on the following intuition. Due to the locally strongly convex property discussed after Assumption 2, there exists (with high probability) the unique global minimum $\boldsymbol{w}_{\boldsymbol{S}}^*$ of $R_{\boldsymbol{S}}(\cdot)$ and $\boldsymbol{w}_{\boldsymbol{S}'}^*$ of $R_{\boldsymbol{S}'}(\cdot)$ that concentrate around the unique (the uniqueness is from Assumption 2) population global minimum $\boldsymbol{w}^*$. Then, the provable convergence results of $\boldsymbol{w}_t \to \boldsymbol{w}_{\boldsymbol{S}}^*$ and $\boldsymbol{w}_t' \to \boldsymbol{w}_{\boldsymbol{S}'}^*$ imply the algorithmic stability (see Lemma 3 in Appendix).

**Theorem 2.** *Under Assumption 1-3,*

$$\epsilon_{\text{stab}}(t) \leq \frac{4\sqrt{2}L_0(\lambda + 4DL_2)}{\lambda^{\frac{3}{2}}}\sqrt{\epsilon(t)} + \frac{8L_0}{n\lambda}\left(L_0 + \frac{64L_0^2L_2^2D}{\lambda^3}\right) + \frac{128L_0L_1^2D}{n\lambda^2}\left(5\sqrt{\log d} + \frac{4e\log d}{\sqrt{n}}\right)^2$$

$$= \tilde{\mathcal{O}}(\sqrt{\epsilon(t)} + 1/n),$$

$$\tag{6}$$

*where $\epsilon_{\text{stab}}(t) = \mathbb{E}_{\boldsymbol{S},\boldsymbol{S}'}\left[\sup_{\boldsymbol{z}} |\mathbb{E}_{\mathcal{A}}[f(\boldsymbol{w}_t, \boldsymbol{z}) - f(\boldsymbol{w}_t', \boldsymbol{z})]|\right]$ is the stability of $\boldsymbol{w}_t$, and $\epsilon(t) = \mathbb{E}\left[R_{\boldsymbol{S}}(\boldsymbol{w}_t) - R_{\boldsymbol{S}}(\boldsymbol{w}_{\boldsymbol{S}}^*)\right]$, $\boldsymbol{w}_{\boldsymbol{S}}^*$ is the global minimum of $R_{\boldsymbol{S}}(\cdot)$.*

The proof of this theorem is in Appendix B.1. The expected generalization error of $\boldsymbol{w}_t$ is upper bounded by the right hand side of (6) due to Theorem 1. Compared with the existing result [32], the extra term related to $\sqrt{\epsilon(t)}$ in our bound originates from our proof technique, and it seems to be unavoidable according to [63]. Since for proper algorithms, e.g., GD and SGD, $\epsilon(t) \to 0$ as $t \to \infty$ the leading term of the upper bound (6) is $C^* \log d/n = \tilde{\mathcal{O}}(1/n)$ with $C^* = 3200L_0L_1^2D/\lambda^2$.

In summary, the local strong convexity (Assumption 2) enables us to establish an algorithmic stability based generalization bound (6). The bound improves the classical result of SGD $2L_0^2 \sum_{k=0}^{t-1} \eta_k/n$ in [32] as it can be applied to any proper algorithm with any learning rate and number of iterations.

### 3.2 Excess Risk Under Convex Problems

According to (2), we can upper bound the expected excess risk by combining the generalization upper bound (6) with the convergence results in convex optimization.

**Theorem 3.** *For $\boldsymbol{w}_{\boldsymbol{S}}^* \in \arg\min_{\boldsymbol{w}} R_{\boldsymbol{S}}(\boldsymbol{w})$, and $\boldsymbol{w}^* \in \arg\min_{\boldsymbol{w}} R(\boldsymbol{w})$, under Assumption 1-3,*

$$\mathbb{E}\left[R(\boldsymbol{w}_t) - R(\boldsymbol{w}^*)\right] \le \epsilon(t) + \frac{4\sqrt{2}L_0(\lambda + 4DL_2)}{\lambda^{\frac{3}{2}}}\sqrt{\epsilon(t)} + \frac{8L_0}{n\lambda}\left(L_0 + \frac{64L_0^2L_2^2D}{\lambda^3}\right)$$
$$+ \frac{128L_0L_1^2D}{n\lambda^2}\left(5\sqrt{\log d} + \frac{4e\log d}{\sqrt{n}}\right)^2 \tag{7}$$
$$= \tilde{\mathcal{O}}(\sqrt{\epsilon(t)} + 1/n),$$

*where $\epsilon(t) = \mathbb{E}\left[R_{\boldsymbol{S}}(\boldsymbol{w}_t) - R_{\boldsymbol{S}}(\boldsymbol{w}_{\boldsymbol{S}}^*)\right]$.*

This theorem provides an upper bound of the expected excess risk. The bound decreases with the number of training steps $t$, and is of order $\tilde{\mathcal{O}}(1/n)$ if $t$ is sufficiently large.

**Comparison.** Under the extra local strong convexity assumption, our result significantly improves the bound of order $\mathcal{O}(1/\sqrt{n})$ in [32]. On the other hand, our bound matches (in order) the result under strongly convex problem [63, 81]. It seems our result has a worse dependence on the strong convex parameter $\lambda$, i.e., from $1/\lambda$ to $1/\lambda^4$. The worse dependence is acceptable as local strong convexity is weaker than strong convexity. Moreover, our bound is not necessarily weaker compared to the current results [63, 81] under global strongly convex problem. This is because $\lambda$ in our bound is the local strongly convex parameter restricted around the minimum point, which is larger than the global one over the whole parameter space appears in [81]. Improving the dependence on $\lambda$ without sacrificing the order of $n$ seems to be infeasible based on our techniques[3]. It might be a meaningful topic to be explored in the future. Finally, our result has no conflict with the lower bound for general convex problem in the order of $\mathcal{O}(\sqrt{d/n})$ [25]. This is because Assumption 1 and 2 restrict our result to a smaller class of distributions and functions, which rules out the counter-examples in [25].

To make our results concrete, we apply them to GD and SGD as examples. Note that $R_{\boldsymbol{S}}(\boldsymbol{w}) = n^{-1}\sum_{i=1}^n f(\boldsymbol{w}, \boldsymbol{z}_i)$, the GD and SGD respectively start from $\boldsymbol{w}_0$ follow the update rules of

$$\boldsymbol{w}_{t+1} = \mathcal{P}_{\mathcal{W}}\left(\boldsymbol{w}_t - \eta_t \nabla R_{\boldsymbol{S}}(\boldsymbol{w}_t)\right), \tag{8}$$

and

$$\boldsymbol{w}_{t+1} = \mathcal{P}_{\mathcal{W}}\left(\boldsymbol{w}_t - \eta_t \nabla f(\boldsymbol{w}_t, \boldsymbol{z}_{i_t})\right), \tag{9}$$

where $i_t$ is randomly sampled from 1 to $n$. Note the convergence rate of $\boldsymbol{w}_t$ updated by GD and SGD are respectively $\mathcal{O}(1/t)$ [12] and $\tilde{\mathcal{O}}(1/\sqrt{t})$ [64], we have the following two corollaries declare the converged expected excess risks whose proofs appear in Appendix B.2.

**Corollary 1.** *Under Assumption 1-3, if $\boldsymbol{w}_t$ is updated by GD in (8) with $\eta_t = 1/L_1$, then*

$$R(\boldsymbol{w}_t) - R(\boldsymbol{w}^*) \le \tilde{\mathcal{O}}\left(\frac{1}{\sqrt{t}} + \frac{1}{n}\right). \tag{10}$$

**Corollary 2.** *Under Assumption 1-3, if $\boldsymbol{w}_t$ is updated by SGD in (9) with $\eta_t = D/(L_1\sqrt{t+1})$, then*

$$\mathbb{E}\left[R(\boldsymbol{w}_t) - R(\boldsymbol{w}^*)\right] \le \tilde{\mathcal{O}}\left(\frac{1}{t^{\frac{1}{4}}} + \frac{1}{n}\right). \tag{11}$$

## 4 Excess Risk Under Non-Convex Problems

In this section, we present the upper bounds of the expected excess risk of iterates obtained by proper algorithms that approximate local minima under non-convex problems.

### 4.1 Generalization Error Under Non-Convex Problems

In this subsection, we study the generalization error under non-convex problems. Unfortunately, the analysis in Section 3 can not be directly generalized here due to the following reason. The generalization error under convex problems relies on the fact that there exists the *unique* empirical local minima $\boldsymbol{w}_{\boldsymbol{S}}^*$ of $R_{\boldsymbol{S}}(\cdot)$ and $\boldsymbol{w}_{\boldsymbol{S}'}^*$ of $R_{\boldsymbol{S}'}(\cdot)$ that concentrate around the *unique* population local minimum $\boldsymbol{w}^*$ of $R(\cdot)$. Under non-convex problems, there can be many empirical and population

---

[3]The dependence can be improved to $1/\lambda^2$ with a worse order of $n$ (from $1/n$ to $1/\sqrt{n}$).

local minima. The iterates obtained on $\boldsymbol{S}$ and $\boldsymbol{S}'$ may converge to different empirical local minima away from each other, which invalidates our methods used in convex problems.

Fortunately, we can prove that for each population local minimum, there is an empirical local minimum concentrated around it with high probability. If the generalization upper bound for these local minima is established, and there are no extra empirical local minima, the convergence results of the iterates obtained by proper algorithms imply their generalization ability. Next, we prove our results following this road map.

First, we establish the generalization upper bound for the empirical local minima around the population local minima. According to Proposition 1 in the Appendix C.1, there are only finite population local minima, thus the non-convex problems with local minima consists of a manifold [48] is not considered in this paper. Let $\mathcal{M} = \{\boldsymbol{w}_1^*, \cdots, \boldsymbol{w}_K^*\}$ be the set of population local minima. The number of local minima $K$ may depend on the problem of interest. In many important non-convex problems, $K$ can be quite small, e.g., $K = 2$ for PCA [30] and $K = 1$ for robust regression [49].

Then, we notice that the population risk is strongly convex in $B_2(\boldsymbol{w}_k^*, \lambda/(4L_2))$. Similar to the scenario under convex problems, we can verify that the empirical risk is locally strongly convex in $B_2(\boldsymbol{w}_k^*, (\lambda/4L_2))$ with high probability. Next, we consider the following points

$$\boldsymbol{w}_{\boldsymbol{S},k}^* = \underset{\boldsymbol{w} \in B_2(\boldsymbol{w}_k^*, \frac{\lambda}{4L_2})}{\arg\min} R_{\boldsymbol{S}}(\boldsymbol{w}), \tag{12}$$

for $k = 1 \ldots, K$. We show that $\boldsymbol{w}_{\boldsymbol{S},k}^*$ is a local minimum of $R_{\boldsymbol{S}}(\cdot)$ with high probability and present the generalization bound of it. Note that in Theorem 1, $\mathcal{A}$ can be infeasible. We construct an auxiliary sequence $\boldsymbol{w}_t$ via an infeasible algorithm.

$$\boldsymbol{w}_{t+1} = \mathcal{P}_{B_2(\boldsymbol{w}_k^*, \frac{\lambda}{4L_2})}\left(\boldsymbol{w}_t - \frac{1}{L_1}\nabla R_{\boldsymbol{S}}(\boldsymbol{w}_t)\right). \tag{13}$$

Then, as $\boldsymbol{w}_t$ locates in $B_2(\boldsymbol{w}_k^*, \lambda/(4L_2))$ in which $R_{\boldsymbol{S}}(\cdot)$ is strongly convex with high probability, we can establish the algorithmic stability bound of the $\boldsymbol{w}_t$. Combining this with the convergence result of $\boldsymbol{w}_t$ to $\boldsymbol{w}_{\boldsymbol{S},k}^*$ implies the generalization ability of $\boldsymbol{w}_{\boldsymbol{S},k}^*$. The following lemma states our result rigorously.

**Lemma 1.** *Under Assumption 1 and 4, for $k = 1, \ldots, K$, with probability at least*

$$1 - \frac{512L_0^2L_2^2}{n\lambda^4} - \frac{128L_1^2}{n\lambda^2}\left(5\sqrt{\log d} + \frac{4e\log d}{\sqrt{n}}\right)^2, \tag{14}$$

$\boldsymbol{w}_{\boldsymbol{S},k}^*$ [4] *is a local minimum of $R_{\boldsymbol{S}}(\cdot)$. Moreover, for such $\boldsymbol{w}_{\boldsymbol{S},k}^*$, we have*

$$\begin{aligned}|\mathbb{E}_{\boldsymbol{S}}[R_{\boldsymbol{S}}(\boldsymbol{w}_{\boldsymbol{S},k}^*) - R(\boldsymbol{w}_{\boldsymbol{S},k}^*)]| &\leq \frac{8L_0}{n\lambda}\left(L_0 + \frac{64L_0^2L_2^2}{\lambda^3}\right)\min\left\{3D, \frac{3\lambda}{2L_2}\right\} \\ &+ \frac{128L_0L_1^2}{n\lambda^2}\left(5\sqrt{\log d} + \frac{4e\log d}{\sqrt{n}}\right)^2 \min\left\{3D, \frac{3\lambda}{2L_2}\right\}.\end{aligned} \tag{15}$$

The lemma is proved in Appendix C.1.1., and it guarantees the generalization ability of those empirical local minima located around population local minima. The expected generalization error on these local minima is of order $\tilde{\mathcal{O}}(1/n)$ as in convex problems. In the sequel, we show that there are no extra empirical local minima expected for these $\boldsymbol{w}_{\boldsymbol{S},k}^*$ with high probability, under the following mild assumption, which also appears in [49, 30].

**Assumption 4** (Strict saddle). *There exists $\alpha, \lambda > 0$ such that $\|\nabla R(\boldsymbol{w})\| > \alpha$ on the boundary of $\mathcal{W}$, and*

$$\|\nabla R(\boldsymbol{w})\| \leq \alpha \Rightarrow |\sigma_{\min}(\nabla^2 R(\boldsymbol{w}))| \geq \lambda, \tag{16}$$

*where $\sigma_{\min}(\nabla^2 R(\boldsymbol{w}))$ is $\nabla^2 R(\boldsymbol{w})$'s smallest eigenvalue.*

The Assumption 4 is a generalized version of local strong convexity Assumption 2 (can be implied by Assumption 4). A vast vary of machine learning problems satisfy this assumption, e.g., generalized linear regression, robust regression, normal mixture model, tensor decomposition, matrix completion, PCA, and ICA [30, 49, 81]. We refer readers to [30, 27, 28, 49] for more details of this assumption.

Let $\mathcal{M}_{\boldsymbol{S}} = \{\boldsymbol{w} : \boldsymbol{w} \text{ is a local minimum of } R_{\boldsymbol{S}}(\cdot)\}$ be the set consists of all the local minima of empirical risk $R_{\boldsymbol{S}}(\cdot)$. Then we establish the following non-asymptotic probability bound.

---

[4] Please note the definition of $\boldsymbol{w}_{\boldsymbol{S},k}^*$ in (12) which is not necessary to be a local minimum.

**Lemma 2.** *Under Assumption 1 and 4, for $r = \min\left\{\frac{\lambda}{8L_2}, \frac{\alpha^2}{16L_0L_1}\right\}$, with probability at least*

$$1 - 2\left(\frac{3D}{r}\right)^d \exp\left(-\frac{n\alpha^4}{128L_0^4}\right) - 4d\left(\frac{3D}{r}\right)^d \exp\left(-\frac{n\lambda^2}{128L_1^2}\right)$$

$$- K\left\{\frac{512L_0^2L_2^2}{n\lambda^4} + \frac{128L_1^2}{n\lambda^2}\left(5\sqrt{\log d} + \frac{4e\log d}{\sqrt{n}}\right)^2\right\}, \tag{17}$$

*we have*

    *i:* $\mathcal{M}_{\boldsymbol{S}} = \{\boldsymbol{w}_{\boldsymbol{S},1}^*, \ldots, \boldsymbol{w}_{\boldsymbol{S},K}^*\}$;

    *ii: for any $\boldsymbol{w} \in \mathcal{W}$, if $\|\nabla R_{\boldsymbol{S}}(\boldsymbol{w})\| < \alpha^2/(2L_0)$ and $\nabla^2 R_{\boldsymbol{S}}(\boldsymbol{w}) \succ -\lambda/2$, then $\|\boldsymbol{w} - \mathcal{P}_{\mathcal{M}_{\boldsymbol{S}}}(\boldsymbol{w})\| \leq \lambda\|\nabla R_{\boldsymbol{S}}(\boldsymbol{w})\|/4$,*

*where $\nabla^2 R_{\boldsymbol{S}}(\boldsymbol{w}) \succ -\lambda/2$ means $\nabla^2 R_{\boldsymbol{S}}(\boldsymbol{w}) + \lambda/2 \boldsymbol{I}_d$ is a positive definite matrix.*

The first conclusion in this lemma states that there are no extra empirical local minima except for those $\boldsymbol{w}_{\boldsymbol{S},k}^*$ concentrate around population local minima, which have guaranteed generalization ability (by Theorem 1). The second result is that the empirical risk is "error bound" (see [42] for its definition) around its local minima, with high probability. The "error bound" is a nice property in optimization [42]. Proof of the lemma is in Appendix C.2.1. The probability bound (17) will appear in the generalization bound of iterates obtained by proper algorithms accounting for the existence of those empirical local minima away from population local minima. We defer the discussion to the bound after providing our generalization upper bound in Theorem 4.

We move forward to derive the generalization upper bound of those iterates obtained by the proper algorithm that approximates the local minima under non-convex problems. Under strict saddle Assumption 4, the proper algorithm $\mathcal{A}$ approximates the second-order stationary point (SOSP) [5], that says with probability at least $1 - \delta$ ($\delta$ is a constant that can be arbitrary small),

$$\|\nabla R_{\boldsymbol{S}}(\boldsymbol{w}_t)\| \leq \zeta(t), \qquad \nabla^2 R_{\boldsymbol{S}}(\boldsymbol{w}_t) \succeq -\rho(t) \tag{18}$$

where $\boldsymbol{w}_t$ is updated by the algorithm $\mathcal{A}$, and $\zeta(t), \rho(t) \to 0$ (which may have poly-logarithmic dependence on $\delta$ [37]) as $t \to \infty$.

To instantiate such proper algorithms, we construct an algorithm that satisfies (18) in Appendix D. The following theorem establishes a generalization upper bound of $\boldsymbol{w}_t$ obtained by such $\mathcal{A}$.

**Theorem 4.** *Under Assumption 1, 2 and 4, if $\boldsymbol{w}_t$ satisfies (18) and $r$ defined in Lemma 2, by choosing $t$ such that $\zeta(t) < \alpha^2/(2L_0)$ and $\rho(t) < \lambda/2$ we have*

$$|\mathbb{E}_{\mathcal{A},\boldsymbol{S}}\left[R(\boldsymbol{w}_t) - R_{\boldsymbol{S}}(\boldsymbol{w}_t)\right]| \leq \frac{8L_0}{\lambda}\zeta(t) + 2L_0D\delta + \frac{2KM}{\sqrt{n}} + \frac{8KL_0^2}{n\lambda}$$

$$+ \left(L_0\min\left\{3D, \frac{3\lambda}{2L_2}\right\} + 2M\right)\xi_{n,1} + 2M\xi_{n,2} \tag{19}$$

$$= \tilde{\mathcal{O}}\left(\zeta(t) + \frac{1}{\sqrt{n}}\right) \qquad (d/n \leq \mathcal{O}(1)),$$

*where*

$$\xi_{n,1} = K\left\{\frac{512L_0^2L_2^2}{n\lambda^4} + \frac{128L_1^2}{n\lambda^2}\left(5\sqrt{\log d} + \frac{4e\log d}{\sqrt{n}}\right)^2\right\}, \tag{20}$$

*and*

$$\xi_{n,2} = 2\left(\frac{3D}{r}\right)^d \exp\left(-\frac{n\alpha^4}{128L_0^4}\right) + 4d\left(\frac{3D}{r}\right)^d \exp\left(-\frac{n\lambda^2}{128L_1^2}\right). \tag{21}$$

*If with probability at least $1 - \delta'$ ($\delta'$ can be arbitrary small), $R_{\boldsymbol{S}}(\cdot)$ has no spurious local minimum, then*

$$|\mathbb{E}_{\mathcal{A},\boldsymbol{S}}\left[R(\boldsymbol{w}_t) - R_{\boldsymbol{S}}(\boldsymbol{w}_t)\right]| \leq \frac{8L_0}{\lambda}\zeta(t) + 2L_0D\delta + 6M\delta' + \frac{8(K+4)L_0^2}{n\lambda}$$

$$+ \left(\frac{(K+4)L_0}{K}\min\left\{3D, \frac{3\lambda}{2L_2}\right\} + 6M\right)\xi_{n,1} + 6M\xi_{n,2} \tag{22}$$

$$= \tilde{\mathcal{O}}\left(\zeta(t) + \frac{1}{n}\right) \qquad (d/n \leq \mathcal{O}(1)).$$

---

[5]$\boldsymbol{w}$ is a $(\epsilon, \gamma)$-second-order stationary point (SOSP) if $\|\nabla R_{\boldsymbol{S}}(\boldsymbol{w})\| \leq \epsilon$ and $\nabla^2 R_{\boldsymbol{S}}(\boldsymbol{w}) \succeq -\gamma$

This theorem is proved in Appendix C.3, and it provides upper bounds of the expected generalization error of iterates obtained by any proper algorithm that approximates SOSP. We present an explanation of each term in it as follows. The $2DL_0\delta$ is of order $\mathcal{O}(1/\sqrt{n})$ or $\mathcal{O}(1/n)$ as we take the corresponded $\delta = 1/\sqrt{n}$ or $1/n$, and $8L_0\zeta(t)/\lambda$ can be arbitrary small if we take a sufficiently large $t$. Since $\xi_{n,1}$ is of order $\tilde{\mathcal{O}}(1/n)$, we next explore $\xi_{n,2}$. The leading term in $\xi_{n,2}$ is

$$4d\left(\frac{3D}{r}\right)^d \exp\left(-\frac{n\lambda^2}{128L_1^2}\right) = \exp\left(\log 4d + d\log\left(\frac{3D}{r}\right) - \frac{n\lambda^2}{128L_1^2}\right). \tag{23}$$

If $d$ is large enough to make $\log 4d \leq d\log(3D/r)$, then $\xi_{n,2} \leq \exp(-c_2 n(c_1 - \frac{d}{n}))$, where $c_1 = \lambda^2/(256L_1^2\log(3D/r))$ and $c_2 = 2\log(3D/r)$. Thus $\xi_{n,2} \ll \tilde{\mathcal{O}}(1/n)$ provided by $d/n < c_1$. In this case, the $2KM/\sqrt{n}$ appears in bound (19) implies it is of order $\tilde{\mathcal{O}}(1/\sqrt{n})$, even under high-dimensional problems such that $d$ is in the same order of $n$. The $K$ can be small here for many non-convex problems, as previously discussed. Moreover, the bound (22) improves the result in (19) to $\tilde{\mathcal{O}}(1/n)$, under the condition of empirical risk has no spurious local minima with high probability (i.e. $\delta' \leq \tilde{\mathcal{O}}(1/n)$). The condition has been proven to be satisfied by many important non-convex optimization problems e.g., PCA [30], matrix completion [28], and over-parameterized neural network [43, 1, 21].

**Comparison.** Under the extra strictly saddle Assumption 4, our bounds (no matter whether imposing the no spurious local minima assumption) improve the classical results of order $\mathcal{O}(\sqrt{d/n})$ based on the uniform convergence theory [63] or the one of order $\mathcal{O}(t^c/n)$ for a positive $c$ [32, 79] based on algorithmic stability. [30] get the result of order $\tilde{\mathcal{O}}(d/n)$ under the same Assumptions 1 and 4 imposed in this paper. However, their bound has a linear dependence on $d$, thus can not be non-vacuous like ours when $d$ is in the same order of $n$.

Specifically, if the parameter space satisfies some sparsity conditions [6, 80, 35, 36, 22, 72], we can extrapolate Theorem 4 to ultrahigh-dimensional problem such that $d \gg n$. For example, suppose the parameter space $\mathcal{W}$ is contained in a $\ell_1$-ball, i.e., $\|\boldsymbol{w}_1 - \boldsymbol{w}_2\|_1 \leq D'$ for some positive $D'$. Note that the covering number (defined in [72]) of polytopes (Corollary 0.0.4 in [71]) is much smaller than that of $\ell_2$-ball. Then, applying the similar proof of Theorem 4 establishes the same upper bound of generalization error w.r.t. $\boldsymbol{w}_t$ with $\xi_{n,2}$ in Theorem 4 replaced by

$$2(2d)^{(2D'/r)^2+1}\exp\left(-\frac{n\alpha^4}{128L_0^4}\right) + 2(2d)^{(2D'/r)^2+2}\exp\left(-\frac{n\lambda^2}{128L_1^2}\right) \ll \tilde{\mathcal{O}}\left(\frac{1}{n}\right), \tag{24}$$

where the much smaller relationship is valid as long as $\log(d)/n \to 0$.

### 4.2 Excess Risk Under Non-Convex Problems

In this subsection, we establish upper bounds for the expected excess risk of iterates obtained by proper algorithms under non-convex problems. In contrast to convex optimization, the proper algorithm under non-convex problems is not guaranteed to find the global minimum, as it only approximates SOSP. Hence the optimization error may not vanish as in Theorem 3. The following theorem proved in Appendix C.4 establishes an upper bound of the expected excess risk.

**Theorem 5.** *Under Assumption 1, 2 and 4, if $\boldsymbol{w}_t$ satisfies (18), by choosing $t$ in (18) such that $\zeta(t) < \alpha^2/(2L_0)$ and $\rho(t) < \lambda/2$, we have*

$$\begin{aligned}
\mathbb{E}_{\mathcal{A},\boldsymbol{S}}\left[R(\boldsymbol{w}_t) - R(\boldsymbol{w}^*)\right] \leq{} & \frac{4L_0}{\lambda}\zeta(t) + L_0 D\delta + \frac{2KM}{\sqrt{n}} \\
& + \frac{8KL_0^2}{n\lambda} + \left(L_0\min\left\{3D, \frac{3\lambda}{2L_2}\right\} + 2M\right)\xi_{n,1} + 2M\xi_{n,2} \\
& + \mathbb{E}_{\mathcal{A},\boldsymbol{S}}[R_{\boldsymbol{S}}(\mathcal{P}_{\mathcal{M}_{\boldsymbol{S}}}(\boldsymbol{w}_t)) - R_{\boldsymbol{S}}(\boldsymbol{w}_{\boldsymbol{S}}^*)] \\
={} & \mathbb{E}_{\mathcal{A},\boldsymbol{S}}[R_{\boldsymbol{S}}(\mathcal{P}_{\mathcal{M}_{\boldsymbol{S}}}(\boldsymbol{w}_t)) - R_{\boldsymbol{S}}(\boldsymbol{w}_{\boldsymbol{S}}^*)] + \tilde{\mathcal{O}}\left(\zeta(t) + \frac{1}{\sqrt{n}}\right) \qquad (d/n \leq \mathcal{O}(1)),
\end{aligned}$$

$$\tag{25}$$

*where $\boldsymbol{w}^*$ is the global minimum of the population risk. If with probability at least $1 - \delta'$ ($\delta'$ can be arbitrary small), $R_{\boldsymbol{S}}(\cdot)$ has no spurious local minimum, then*

$$\mathbb{E}_{\mathcal{A}, \boldsymbol{S}}\left[R(\boldsymbol{w}_t) - R(\boldsymbol{w}^*)\right] \leq \frac{4L_0}{\lambda}\zeta(t) + L_0 D\delta + 8M\delta' + \frac{8(K+4)L_0^2}{n\lambda}$$
$$+ \left(\frac{(K+4)L_0}{K}\min\left\{3D, \frac{3\lambda}{2L_2}\right\} + 8M\right)\xi_{n,1} + 8M\xi_{n,2} \qquad (26)$$
$$= \tilde{\mathcal{O}}\left(\zeta(t) + \frac{1}{n}\right) \qquad (d/n \leq \mathcal{O}(1)),$$

*where $\xi_{n,1}$ and $\xi_{n,2}$ are defined in Theorem 4, and $\boldsymbol{w}_{\boldsymbol{S}}^*$ is the global minimum of $R_{\boldsymbol{S}}(\cdot)$.*

From the discussions in the last section, the bound (25) and (26) become $\mathcal{O}(1/\sqrt{n})$ and $\tilde{\mathcal{O}}(1/n)$, respectively, when $d$ is in the same order of $n$ and $t \to \infty$. Besides that, in (25), expected for the order of convergence rate $\mathcal{O}(\zeta(t))$ and the generalization bound of order $\tilde{\mathcal{O}}(1/\sqrt{n} + \exp(-c_2 n(c_1 - d/n))$ [6], there is an extra $\mathbb{E}_{\mathcal{A}, \boldsymbol{S}}[R_{\boldsymbol{S}}(\mathcal{P}_{\mathcal{M}_{\boldsymbol{S}}}(\boldsymbol{w}_t)) - R_{\boldsymbol{S}}(\boldsymbol{w}_{\boldsymbol{S}}^*)]$ in the bound (25), compared with the result of convex problems in Theorem 3. This is the gap between the empirical global minimum and the algorithmic approximated empirical local minimum. The gap seems necessary as the proper algorithm is not guaranteed to find the global minima, and if so, the gap becomes zero.

The bound (26) of order $\tilde{\mathcal{O}}(1/n)$ is obtained under empirical risk without spurious local minima, which is proven to be hold on many important non-convex problems e.g., PCA [30], matrix completion [28], and over-parameterized neural network [43, 1, 21, 85].

## 5 Related Works

**Generalization**   The generalization error is the gap between the model's performance on training and unseen test data. One of the central tools to bound the generalization error in statistical learning is uniform convergence theory. However, this method is unavoidably related to the capacity of hypothesis space e.g., VC dimension [7, 17, 59, 31], Rademacher complexity [3, 51, 55], covering number [73, 82, 65], or entropy integral [72]. Thus, these results are not well suited for high-dimensional hypothesis spaces, which makes the mentioned measures to be large.

The generalization error of the iterates obtained by some algorithms, e.g., GD or SGD, is often of more interest. There are plenty of papers working on this topic via the tool of algorithmic stability [9, 26, 10, 30, 63], differential privacy [18, 41], robustness of model [75, 66, 77], and information theory [74, 67, 11]. However, these tools either depend heavily on algorithm implementation (algorithmic stability and information theory) or require unverifiable conditions (robustness and differential privacy). This paper combines the technique of characterizing empirical loss landscape and algorithmic stability to explore the generalization under both convex and non-convex problems. Our methods develop a new way to use algorithmic stability, which can be applied without restrictions on the algorithm, learning rate, and the number of iterations.

**Optimization**   Results in this paper are related to both convex and non-convex problems.

For convex problems, [12] summarizes most of the classical algorithms in convex optimization. Some other novel methods [40, 62, 56] with lower computational complexity have also been extensively explored. Recently, the non-convex optimization has attracted quite a lot attentions owing to the development of deep learning [33, 70]. But most of the existing algorithms [29, 2, 57, 15, 23, 78] approximate the first-order stationary point instead of local minima.

Under non-convex problem, the algorithm that approximates SOSP is proper (approximate local minima) in this paper. We refer readers for recent progress in the topic of developing algorithms approximating SOSP to [27, 24, 19, 37, 39, 76, 52, 84, 38]. The discussed proper algorithms in this paper have constrained parameter space which is different from the ones in [5, 13, 52]. To resolve this, we also develop an algorithm that approximates SOSP under our constraints in Appendix D.

---

[6]The difference in the coefficients of the convergence rate term $\zeta(t)$ between the bounds in Theorem 4 and 5 is due to a technique issue and not essential.

**Excess Risk** A straightforward way to characterize the excess risk is by controlling the generalization and optimization errors, respectively, as we did in this paper. Thus, for this problem, the used tools are similar to the ones in analyzing generalization, e.g., uniform convergence theory [69, 81, 25], algorithmic stability [32, 14, 16, 79, 20], information theory [53, 54]. However, the discussed drawbacks of these tools also appeared. Our results are built upon the combination of characterizing empirical risk's landscape and algorithmic stability. Moreover, they are dimensional insensitive, independent of algorithm's implementation, and they improve the order of existing results under both convex and non-convex problems.

## 6   Conclusion

This paper provides a unified analysis of the expected excess risk of models trained by proper algorithms under convex and non-convex problems. Our primary techniques are algorithmic stability and the non-asymptotic characterization of the empirical risk's landscape.

Under the conditions of local strong convexity around population local minima and some other mild regularity conditions, we establish the upper bounds of the expected excess risk in the order of $\tilde{\mathcal{O}}(1/n)$ and $\tilde{\mathcal{O}}(1/\sqrt{n})$ (can be improved to $\tilde{\mathcal{O}}(1/n)$ when empirical risk has no spurious local minima with high probability) under convex and non-convex problems respectively.

The presented results improve the existing results in many aspects. For convex problems, our results improve the standard excess risk bound of order $\mathcal{O}(\sqrt{1/n})$ [32] to $\tilde{\mathcal{O}}(1/n)$ under locally convex assumption. For non-convex problems, our results significantly improve the standard uniform convergence bound in the order of $\mathcal{O}(\sqrt{d/n})$ [63] when $d/n$ is smaller than a universal constant. Moreover, our results can be generally applied to algorithms that approximate local minima, and they have no restrictions on the algorithm, learning rate, and number of iterations.

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
