# A  Proof of Theorem 1

*Proof.* Recall that $\{z_1, \cdots, z_n, z_1', \cdots, z_n'\}$ are $2n$ i.i.d samples from the target population, $\boldsymbol{S} = \{z_1, \cdots, z_n\}$, $\boldsymbol{S}^i = \{z_1, \cdots, z_{i-1}, z_i', z_{i+1}, \cdots, z_n\}$, and $\boldsymbol{S}' = \boldsymbol{S}^1$. We have

$$\mathbb{E}_{\mathcal{A}, \boldsymbol{S}}\left[R(\mathcal{A}(\boldsymbol{S})) - R_{\boldsymbol{S}}(\mathcal{A}(\boldsymbol{S}))\right] = \mathbb{E}_{\mathcal{A}, \boldsymbol{S}, \boldsymbol{z}}\left[\frac{1}{n}\sum_{i=1}^{n}(f(\mathcal{A}(\boldsymbol{S}), \boldsymbol{z}) - f(\mathcal{A}(\boldsymbol{S}), \boldsymbol{z}_i))\right]$$

$$= \mathbb{E}_{\mathcal{A}, \boldsymbol{S}, \boldsymbol{S}^i}\left[\frac{1}{n}\sum_{i=1}^{n}\Big(f(\mathcal{A}(\boldsymbol{S}^i), \boldsymbol{z}_i) - f(\mathcal{A}(\boldsymbol{S}), \boldsymbol{z}_i)\Big)\right] \qquad (27)$$

$$= \frac{1}{n}\sum_{i=1}^{n}\mathbb{E}_{\mathcal{A}, \boldsymbol{S}, \boldsymbol{S}^i}\left[f(\mathcal{A}(\boldsymbol{S}^i), \boldsymbol{z}_i) - f(\mathcal{A}(\boldsymbol{S}), \boldsymbol{z}_i)\right].$$

Thus

$$|\mathbb{E}_{\mathcal{A}, \boldsymbol{S}}\left[R(\mathcal{A}(\boldsymbol{S})) - R_{\boldsymbol{S}}(\mathcal{A}(\boldsymbol{S}))\right]| \leq \frac{1}{n}\sum_{i=1}^{n}\mathbb{E}_{\boldsymbol{S}, \boldsymbol{S}^i}\left|\mathbb{E}_{\mathcal{A}}\left[f(\mathcal{A}(\boldsymbol{S}^i), \boldsymbol{z}_i) - f(\mathcal{A}(\boldsymbol{S}), \boldsymbol{z}_i)\right]\right|$$

$$\leq \mathbb{E}_{\boldsymbol{S}, \boldsymbol{S}'}\left[\sup_{\boldsymbol{z}}|\mathbb{E}_{\mathcal{A}}[f(\mathcal{A}(\boldsymbol{S}'), \boldsymbol{z}) - f(\mathcal{A}(\boldsymbol{S}), \boldsymbol{z})]|\right]$$

$$\leq \epsilon,$$

where the last inequality is due to the $\epsilon$-uniform stability. $\qquad\square$

# B  Proofs in Section 3

Throughout this and the following proofs, for any symmetric matrix $\boldsymbol{A}$, we denote its smallest and largest eigenvalue by $\sigma_{\min}(\boldsymbol{A})$ and $\sigma_{\max}(\boldsymbol{A})$, respectively.

## B.1  Proofs in Section 3.1

Before providing the proof of Theorem 2, we need several lemmas. First we define two "good events"

$$E_1 = \left\{\|\nabla R_{\boldsymbol{S}}(\boldsymbol{w}^*)\| \leq \frac{\lambda^2}{16L_2}, \|\nabla R_{\boldsymbol{S}'}(\boldsymbol{w}^*)\| \leq \frac{\lambda^2}{16L_2}\right\}$$

$$E_2 = \left\{\|\nabla^2 R_{\boldsymbol{S}}(\boldsymbol{w}^*) - \nabla^2 R(\boldsymbol{w}^*)\| \leq \frac{\lambda}{4}, \|\nabla^2 R_{\boldsymbol{S}'}(\boldsymbol{w}^*) - \nabla^2 R(\boldsymbol{w}^*)\| \leq \frac{\lambda}{4}\right\} \qquad (28)$$

The following lemma is based on the fact that on event $E_1 \bigcap E_2$ the empirical global minimum is around the population global minimum.

**Lemma 3.** *Under Assumptions 1-3, there exists global minimum $\boldsymbol{w}_{\boldsymbol{S}}^*$ and $\boldsymbol{w}_{\boldsymbol{S}'}^*$ of $R_{\boldsymbol{S}}(\cdot)$ and $R_{\boldsymbol{S}'}(\cdot)$ such that*

$$\mathbb{E}\left[\|\boldsymbol{w}_{\boldsymbol{S}}^* - \boldsymbol{w}_{\boldsymbol{S}'}^*\|\mathbf{1}_{E_1 \bigcap E_2}\right] \leq \frac{8L_0}{n\lambda}, \qquad (29)$$

*where $\mathbf{1}_{(\cdot)}$ is the indicative function and $\boldsymbol{w}^*$ is the sole global minimum of $R(\cdot)$.*

*Proof.* To begin with, we show $R_{\boldsymbol{S}}(\cdot)$ is locally strongly convex around $\boldsymbol{w}^*$ with high probability. Then, by providing that there exists $\boldsymbol{w}_{\boldsymbol{S}}^*$ and $\boldsymbol{w}_{\boldsymbol{S}'}^*$ locates in the region, we get the conclusion.

We claim that if the event $E_1 \bigcap E_2$ happens, then $\nabla^2 R_{\boldsymbol{S}}(\boldsymbol{w}) \succeq \frac{\lambda}{2}$ for any $\boldsymbol{w} \in B_2(\boldsymbol{w}^*, \frac{\lambda}{4L_2})$. Since

$$\sigma_{\min}(\nabla^2 R_{\boldsymbol{S}}(\boldsymbol{w})) = \sigma_{\min}\left(\nabla^2 R_{\boldsymbol{S}}(\boldsymbol{w}) - \nabla^2 R_{\boldsymbol{S}}(\boldsymbol{w}^*) + \nabla^2 R_{\boldsymbol{S}}(\boldsymbol{w}^*) - \nabla^2 R(\boldsymbol{w}^*) + \nabla^2 R(\boldsymbol{w}^*)\right)$$

$$\geq \sigma_{\min}(\nabla^2 R(\boldsymbol{w}^*)) - \|\nabla^2 R_{\boldsymbol{S}}(\boldsymbol{w}) - \nabla^2 R_{\boldsymbol{S}}(\boldsymbol{w}^*)\| - \|\nabla^2 R_{\boldsymbol{S}}(\boldsymbol{w}^*) - \nabla^2 R(\boldsymbol{w}^*)\| \quad (30)$$

$$\geq \lambda - L_2\|\boldsymbol{w} - \boldsymbol{w}^*\| - \frac{\lambda}{4} \geq \frac{\lambda}{2},$$

where the last inequality is due to the Lipschitz Hessian and event $E_2$. After that, we show that both $\boldsymbol{w}_{\boldsymbol{S}}^*$ and $\boldsymbol{w}_{\boldsymbol{S}'}^*$ locate in $B_2(\boldsymbol{w}^*, \frac{\lambda}{4L_2})$, when $E_1, E_2$ hold. Let $\boldsymbol{w} = \gamma \boldsymbol{w}_{\boldsymbol{S}}^* + (1 - \gamma)\boldsymbol{w}^*$, with $\gamma = \frac{\lambda}{4L_2\|\boldsymbol{w}_{\boldsymbol{S}}^* - \boldsymbol{w}^*\|}$ then

$$\|\boldsymbol{w} - \boldsymbol{w}^*\| = \gamma\|\boldsymbol{w}_{\boldsymbol{S}}^* - \boldsymbol{w}^*\|. \qquad (31)$$

One can see $\boldsymbol{w} \in S_2(\boldsymbol{w}^*, \frac{\lambda}{4L_2})$. Thus by the strong convexity,

$$\|\boldsymbol{w} - \boldsymbol{w}^*\|^2 \le \frac{4}{\lambda}(R_{\boldsymbol{S}}(\boldsymbol{w}) - R_{\boldsymbol{S}}(\boldsymbol{w}^*) + \langle \nabla R_{\boldsymbol{S}}(\boldsymbol{w}^*), \boldsymbol{w} - \boldsymbol{w}^* \rangle) < \frac{4}{\lambda}\|\nabla R_{\boldsymbol{S}}(\boldsymbol{w}^*)\| \|\boldsymbol{w} - \boldsymbol{w}^*\|, \quad (32)$$

where the last inequality is due to the convexity such that

$$R_{\boldsymbol{S}}(\boldsymbol{w}) - R_{\boldsymbol{S}}(\boldsymbol{w}^*) = R_{\boldsymbol{S}}(\gamma \boldsymbol{w}_{\boldsymbol{S}}^* + (1-\gamma)\boldsymbol{w}^*) - R_{\boldsymbol{S}}(\boldsymbol{w}^*) \le \gamma(R_{\boldsymbol{S}}(\boldsymbol{w}_{\boldsymbol{S}}^*) - R_{\boldsymbol{S}}(\boldsymbol{w}^*)) < 0 \quad (33)$$

and Schwarz inequality. Then,

$$\frac{\lambda}{4}\|\boldsymbol{w} - \boldsymbol{w}^*\| = \frac{\lambda^2}{16L_2\|\boldsymbol{w}_{\boldsymbol{S}}^* - \boldsymbol{w}^*\|}\|\boldsymbol{w}_{\boldsymbol{S}}^* - \boldsymbol{w}^*\| = \frac{\lambda}{16L_2^2} < \|\nabla R_{\boldsymbol{S}}(\boldsymbol{w}^*)\|, \quad (34)$$

which leads to a contraction to event $E_1$. Thus, we conclude that $\boldsymbol{w}_{\boldsymbol{S}}^* \in B_2(\boldsymbol{w}^*, \frac{\lambda}{4L_2})$. Identically, one can verify that $\boldsymbol{w}_{\boldsymbol{S}'}^* \in B_2(\boldsymbol{w}^*, \frac{\lambda}{4L_2})$.

Since both $\boldsymbol{w}_{\boldsymbol{S}}^*$ and $\boldsymbol{w}_{\boldsymbol{S}'}^*$ are in $B_2(\boldsymbol{w}^*, \frac{\lambda}{4L_2})$ on event $E_1 \bigcap E_2$, $\boldsymbol{S}$ and $\boldsymbol{S}'$ differs in $\boldsymbol{z}_1$, then we have

$$\begin{aligned}
\|\boldsymbol{w}_{\boldsymbol{S}}^* - \boldsymbol{w}_{\boldsymbol{S}'}^*\| &\le \frac{4}{\lambda}\|\nabla R_{\boldsymbol{S}}(\boldsymbol{w}_{\boldsymbol{S}'}^*)\| \\
&= \frac{4}{\lambda}\left\|\frac{1}{n}\sum_{\boldsymbol{z} \in \boldsymbol{S}} \nabla f(\boldsymbol{w}_{\boldsymbol{S}'}^*, \boldsymbol{z})\right\| \\
&= \frac{4}{n\lambda}\|\nabla f(\boldsymbol{w}_{\boldsymbol{S}'}^*, \boldsymbol{z}_1) - \nabla f(\boldsymbol{w}_{\boldsymbol{S}'}^*, \boldsymbol{z}_1')\| \\
&\le \frac{8L_0}{n\lambda},
\end{aligned} \quad (35)$$

where the last equality is due to $\boldsymbol{w}_{\boldsymbol{S}'}^*$ is the minimum of $R_{\boldsymbol{S}'}(\cdot)$. The lemma follows from the fact

$$\mathbb{E}\left[\|\boldsymbol{w}_{\boldsymbol{S}}^* - \boldsymbol{w}_{\boldsymbol{S}'}^*\| \mathbf{1}_{E_1 \cap E_2}\right] \le \frac{8L_0}{n\lambda}\mathbb{P}(E_1 \bigcap E_2) \le \frac{8L_0}{n\lambda}. \quad (36)$$

$\square$

Next, we show that the "good event" happens with high probability.

**Lemma 4.** *Under Assumption 1,*

$$\mathbb{P}(E_1^c \bigcup E_2^c) \le \mathbb{P}(E_1^c) + \mathbb{P}(E_2^c) \le \frac{512L_0^2L_2^2}{n\lambda^4} + \frac{128L_1^2}{n\lambda^2}\left(5\sqrt{\log d} + \frac{4e\log d}{\sqrt{n}}\right)^2, \quad (37)$$

*where $E_k^c$ is the complementary of $E_k$ for $k = 1, 2$.*

*Proof.* By Assumption 1, we have $\|\nabla f(\boldsymbol{w}, \boldsymbol{z})\| \le L_0$ and $\|\nabla^2 f(\boldsymbol{w}, \boldsymbol{z})\| \le L_1$ for any $\boldsymbol{w} \in \mathcal{W}$ and $\boldsymbol{z}$. Thus $\mathbb{E}_{\boldsymbol{z}}[\|\nabla f(\boldsymbol{w}^*, \boldsymbol{z})\|^2] \le L_0^2$ and $\mathbb{E}[\|\nabla^2 f(\boldsymbol{w}, \boldsymbol{z}) - \nabla^2 R(\boldsymbol{w})\|^2] \le 4L_1^2$. For $E^c$, a simple Markov's inequality implies

$$\begin{aligned}
\mathbb{P}(E^c) = \mathbb{P}\left(E_1^c \bigcup E_2^c\right) &\le \mathbb{P}(E_1^c) + \mathbb{P}(E_2^c) \\
&= 2\mathbb{P}\left(\|\nabla R_{\boldsymbol{S}}(\boldsymbol{w}^*)\| > \frac{\lambda^2}{16L_2}\right) + 2\mathbb{P}\left(\|\nabla^2 R_{\boldsymbol{S}}(\boldsymbol{w}^*) - \nabla^2 R(\boldsymbol{w}^*)\| > \frac{\lambda}{4}\right) \\
&\le \frac{512L_2^2}{\lambda^4}\mathbb{E}\left[\|\nabla R_{\boldsymbol{S}}(\boldsymbol{w}^*)\|^2\right] + \frac{32}{\lambda^2}\mathbb{E}\left[\|\nabla^2 R_{\boldsymbol{S}}(\boldsymbol{w}^*) - \nabla^2 R(\boldsymbol{w}^*)\|^2\right].
\end{aligned} \quad (38)$$

By similar arguments as in the proof of Lemma 7 in [83], we have

$$\mathbb{E}\left[\|\nabla R_{\boldsymbol{S}}(\boldsymbol{w}^*)\|^2\right] \le \frac{L_0^2}{n}, \quad (39)$$

and

$$\mathbb{E}\left[\|\nabla^2 R_{\boldsymbol{S}}(\boldsymbol{w}^*) - \nabla^2 R(\boldsymbol{w}^*)\|^2\right] \le \frac{1}{n}\left(10\sqrt{\log d}L_1 + \frac{8e\log d L_1}{\sqrt{n}}\right)^2. \quad (40)$$

Combining these with (38), we have

$$\mathbb{P}(E^c) \le \mathbb{P}(E_1^c) + \mathbb{P}(E_2^c) \le \frac{512L_0^2L_2^2}{n\lambda^4} + \frac{128L_1^2}{n\lambda^2}\left(5\sqrt{\log d} + \frac{4e\log d}{\sqrt{n}}\right)^2. \quad (41)$$

Then Lemma 3 follows from (35) and (37). $\square$

This lemma shows the fact that there exists empirical global minimum on the training set $S$ and $S'$ concentrate around population global minimum $w^*$, so the two empirical global minimum are close with each other. Besides that, the empirical risk is locally strongly convex around this global minimum with high probability.

To present the algorithmic stability, we need to show the convergence of $w_t$ to $w_S^*$ with $w_t$ trained on the training set $S$. However, there is no convergence rate of $\|w_t - w_S^*\|$ under general convex problems, because the quadratic growth condition [7] only holds for strongly convex problems [8]. Fortunately, the local strong convexity of $R_S(\cdot)$ and $R_{S'}(\cdot)$ enables us to upper bound $\|w_t - w_S^*\|$ and $\|w_t' - w_{S'}^*\|$ after a certain number of iterations.

**Lemma 5.** *Under Assumption 1 and 3, for any global minimum $w_S^*$ of $R_S(\cdot)$, define event*

$$E_{0,r} = \left\{ \nabla^2 R_S(w) \succeq \frac{\lambda}{4} : \forall w \in B_2(w_S^*, r) \right\} \tag{42}$$

*for some $r > 0$ and the training set $S$. Then*

$$\mathbb{E}[\|w_t - w_S^*\| \mathbf{1}_{E_{0,r}}] \leq \frac{2\sqrt{2}(r+D)}{r\sqrt{\lambda}} \mathbb{E}\left[R_S(w_t) - R_S(w_S^*)\right]^{\frac{1}{2}}. \tag{43}$$

*Proof.* Define event

$$E_{1,r} = \left\{ R_S(w_t) - R_S(w_S^*) < \frac{\lambda r^2}{8} \right\}. \tag{44}$$

First, we prove on event $E_{0,r} \bigcap E_{1,r}$ we have $\nabla^2 R_S(w_t) \succeq \frac{\lambda}{4}$. If $E_{0,r}$ holds and $w_t \in B_2(w_S^*, r)$, the conclusion is full-filled. On the other hand, if $w_t \notin B_2(w_S^*, r)$ and $E_{0,r} \bigcap E_{1,r}$ happens, for any $w$ with $\|w - w_S^*\| = r$, we have

$$R_S(w) - R_S(w_S^*) \geq \frac{\lambda r^2}{8}, \tag{45}$$

since $E_{0,r}$ holds. Then, let $w = \gamma w_t + (1 - \gamma) w_S^*$ with $\gamma = \frac{r}{\|w_t - w_S^*\|}$. Due to $w \in B_2(w_S^*, r)$ and the convexity of $R_S(\cdot)$,

$$R_S(w) - R_S(w_S^*) \leq \gamma (R_S(w_t) - R_S(w_S^*)) < \frac{\lambda r^2}{8}, \tag{46}$$

which leads to a contraction to (45). Hence, we conclude that on $E_{0,r} \bigcap E_{1,r}$,

$$\|w_t - w_S^*\| \leq \frac{2\sqrt{2}}{\sqrt{\lambda}} (R_S(w_t) - R_S(w_S^*))^{\frac{1}{2}}, \tag{47}$$

due to the local strong convexity. With all these derivations, we see that

$$
\begin{aligned}
\mathbb{E}\left[\|w_t - w_S^*\| \mathbf{1}_{E_{0,r}}\right] &= \mathbb{E}\left[\mathbf{1}_{E_{0,r} \bigcap E_{1,r}} \|w_t - w_S^*\|\right] + \mathbb{E}\left[\mathbf{1}_{E_{0,r} \bigcap E_{1,r}^c} \|w_t - w_S^*\|\right] \\
&\overset{a}{\leq} \frac{2\sqrt{2}}{\sqrt{\lambda}} \mathbb{E}\left[R_S(w_t) - R_S(w_S^*)\right]^{\frac{1}{2}} + D\mathbb{P}(E_{1,r}^c) \\
&\leq \frac{2\sqrt{2}}{\sqrt{\lambda}} \mathbb{E}\left[R_S(w_t) - R_S(w_S^*)\right]^{\frac{1}{2}} + D\frac{2\sqrt{2}}{r\sqrt{\lambda}} \mathbb{E}\left[(R_S(w_t) - R_S(w_S^*))^{\frac{1}{2}}\right] \\
&\leq \frac{2\sqrt{2}(r+D)}{r\sqrt{\lambda}} \mathbb{E}\left[R_S(w_t) - R_S(w_S^*)\right]^{\frac{1}{2}},
\end{aligned} \tag{48}
$$

where $a$ is due to (47) and Jesen's inequality. Thus, we get the conclusion. $\square$

### B.1.1 Proof of Theorem 2

With all these lemmas, we are now ready to prove the Theorem 2.

---

[7]For $f : \mathbb{R}^d \to \mathbb{R}$, quadratic growth means $\frac{\mu}{2}\|w - w^*\|^2 \leq f(w) - f(w^*)$ for some $\mu > 0$, where $w^*$ is the global minimum.

[8]For $f : \mathbb{R}^d \to \mathbb{R}$, strongly convex means $f(w_1) - f(w_2) \leq \langle \nabla f(w_1), w_1 - w_2 \rangle - \frac{\lambda}{2}\|w_1 - w_2\|$ for some $\lambda > 0$ and any $w_1, w_2 \in \mathbb{R}^d$.

**Restate of Theorem 2**  *Under Assumption 1-3, we have*

$$\epsilon_{\text{stab}}(t) \leq \frac{4\sqrt{2}L_0(\lambda + 4DL_2)}{\lambda^{\frac{3}{2}}}\sqrt{\epsilon(t)} + \frac{8L_0}{n\lambda}\left\{L_0 + \frac{64L_0^2L_2^2D}{\lambda^3} + \frac{16L_1^2D}{\lambda}\left(5\sqrt{\log d} + \frac{4e\log d}{\sqrt{n}}\right)^2\right\},$$

(49)

*where $\epsilon_{\text{stab}}(t) = \mathbb{E}_{\boldsymbol{S},\boldsymbol{S}'}\left[\sup_{\boldsymbol{z}}|\mathbb{E}_{\mathcal{A}}[f(\boldsymbol{w}_t,\boldsymbol{z}) - f(\boldsymbol{w}'_t,\boldsymbol{z})]|\right]$ is the stability of the output in the $t$-th step, and $\epsilon(t) = \mathbb{E}[R_{\boldsymbol{S}}(\boldsymbol{w}_t) - R_{\boldsymbol{S}}(\boldsymbol{w}^*_{\boldsymbol{S}})]$ with $\boldsymbol{w}^*_{\boldsymbol{S}}$ as global minimum of $R_{\boldsymbol{S}}(\cdot)$.*

*Proof.* At first glance,

$$\begin{aligned}|f(\boldsymbol{w}_t,\boldsymbol{z}) - f(\boldsymbol{w}'_t,\boldsymbol{z})| &\leq L_0\|\boldsymbol{w}_t - \boldsymbol{w}'_t\| \\ &\leq L_0(\|\boldsymbol{w}_t - \boldsymbol{w}^*_{\boldsymbol{S}}\| + \|\boldsymbol{w}'_t - \boldsymbol{w}^*_{\boldsymbol{S}'}\| + \|\boldsymbol{w}^*_{\boldsymbol{S}} - \boldsymbol{w}^*_{\boldsymbol{S}'}\|).\end{aligned}$$

(50)

We respectively bound these three terms. An upper bound of the third term can be verified by Lemma 3. As proven in Lemma 3, when the two events

$$E_1 = \left\{\|\nabla R_{\boldsymbol{S}}(\boldsymbol{w}^*)\| \leq \frac{\lambda^2}{16L_2}, \|\nabla R_{\boldsymbol{S}'}(\boldsymbol{w}^*)\| \leq \frac{\lambda^2}{16L_2}\right\}$$

$$E_2 = \left\{\|\nabla^2 R_{\boldsymbol{S}}(\boldsymbol{w}^*) - \nabla^2 R(\boldsymbol{w}^*)\| \leq \frac{\lambda}{4}, \|\nabla^2 R_{\boldsymbol{S}'}(\boldsymbol{w}^*) - \nabla^2 R(\boldsymbol{w}^*)\| \leq \frac{\lambda}{4}\right\}$$

(51)

hold, there exists empirical global minimum $\boldsymbol{w}^*_{\boldsymbol{S}}$ and $\boldsymbol{w}^*_{\boldsymbol{S}'}$ such that $\nabla^2 R_{\boldsymbol{S}}(\boldsymbol{w}^*_{\boldsymbol{S}}) \succeq \frac{\lambda}{2}$ and $\nabla^2 R_{\boldsymbol{S}'}(\boldsymbol{w}^*_{\boldsymbol{S}'}) \succeq \frac{\lambda}{2}$. Thus for $\|\boldsymbol{w} - \boldsymbol{w}^*_{\boldsymbol{S}}\| \leq \frac{\lambda}{4L_2}$, we have

$$\sigma_{\min}(\nabla^2 R_{\boldsymbol{S}}(\boldsymbol{w})) \geq \sigma_{\min}(\nabla^2 R_{\boldsymbol{S}}(\boldsymbol{w}^*_{\boldsymbol{S}})) - \|\nabla^2 R_{\boldsymbol{S}}(\boldsymbol{w}) - \nabla^2 R_{\boldsymbol{S}}(\boldsymbol{w}^*_{\boldsymbol{S}})\| \geq \frac{\lambda}{2} - L_2\|\boldsymbol{w} - \boldsymbol{w}^*_{\boldsymbol{S}}\| \geq \frac{\lambda}{4}.$$

(52)

Hence, we conclude that event $E_1 \bigcap E_2 \subseteq E_{\boldsymbol{S}} \bigcap E_{\boldsymbol{S}'}$ with

$$E_{\boldsymbol{S}} = \left\{\nabla^2 R_{\boldsymbol{S}}(\boldsymbol{w}) \succeq \frac{\lambda}{4} : \boldsymbol{w} \in B_2(\boldsymbol{w}^*_{\boldsymbol{S}}, \frac{\lambda}{4L_2})\right\}$$

$$E_{\boldsymbol{S}'} = \left\{\nabla^2 R_{\boldsymbol{S}'}(\boldsymbol{w}) \succeq \frac{\lambda}{4} : \boldsymbol{w} \in B_2(\boldsymbol{w}^*_{\boldsymbol{S}'}, \frac{\lambda}{4L_2})\right\}.$$

(53)

By choosing $r = \frac{\lambda}{4L_2}$ in Lemma 5,

$$\mathbb{E}\left[\|\boldsymbol{w}_t - \boldsymbol{w}^*_{\boldsymbol{S}}\|\mathbf{1}_{E_{\boldsymbol{S}}} + \|\boldsymbol{w}'_t - \boldsymbol{w}^*_{\boldsymbol{S}'}\|\mathbf{1}_{E_{\boldsymbol{S}'}}\right] \leq \left(\frac{4\sqrt{2}}{\sqrt{\lambda}} + \frac{16\sqrt{2}DL_2}{\lambda^{\frac{3}{2}}}\right)\sqrt{\epsilon(t)}.$$

(54)

Note that $E^c_{\boldsymbol{S}} \bigcup E^c_{\boldsymbol{S}'} \subseteq E^c_1 \bigcup E^c_2$ and on the event $E^c_1 \bigcup E^c_2$ we still have

$$|f(\boldsymbol{w}_t,\boldsymbol{z}) - f(\boldsymbol{w}'_t,\boldsymbol{z})| \leq L_0\|\boldsymbol{w}_t - \boldsymbol{w}'_t\| \leq L_0D.$$

(55)

Combining this with (29), (37), (50) and (54), we get the conclusion. $\square$

## B.2   Proofs in Section 3.2

We now respectively prove the convergence results of GD and SGD w.r.t the terminal point in Section 3.2. The two convergence results imply the conclusion of the two Corollaries in Section 3.2.

**Lemma 6.** *Under Assumption 1 and 3, we have*

$$R_{\boldsymbol{S}}(\boldsymbol{w}_t) - R_{\boldsymbol{S}}(\boldsymbol{w}^*_{\boldsymbol{S}}) \leq \frac{D^2 L_1}{2t},$$

(56)

*where $\boldsymbol{w}_t$ is updated by GD in (8) with $\eta_t = 1/L_1$.*

*Proof.* The following descent equation holds due to the Lipschitz gradient,

$$R_{\boldsymbol{S}}(\boldsymbol{w}_k) - R_{\boldsymbol{S}}(\boldsymbol{w}_{k-1}) \leq \langle\nabla R_{\boldsymbol{S}}(\boldsymbol{w}_{k-1}), \boldsymbol{w}_k - \boldsymbol{w}_{k-1}\rangle + \frac{L_1}{2}\|\boldsymbol{w}_k - \boldsymbol{w}_{k-1}\|^2 \leq -\frac{1}{2L_1}\|\boldsymbol{w}_k - \boldsymbol{w}_{k-1}\|^2,$$

(57)

where the last inequality is because the property of projection. On the other hand, we have

$$\begin{aligned}\|\boldsymbol{w}_k - \boldsymbol{w}^*_{\boldsymbol{S}}\|^2 &= \|\boldsymbol{w}_k - \boldsymbol{w}_{k-1} + \boldsymbol{w}_{k-1} - \boldsymbol{w}^*_{\boldsymbol{S}}\|^2 \\ &\leq \|\boldsymbol{w}_k - \boldsymbol{w}_{k-1}\|^2 + 2\langle\boldsymbol{w}_k - \boldsymbol{w}_{k-1}, \boldsymbol{w}_{k-1} - \boldsymbol{w}^*_{\boldsymbol{S}}\rangle + \|\boldsymbol{w}_{k-1} - \boldsymbol{w}^*_{\boldsymbol{S}}\|^2.\end{aligned}$$

(58)

Then, due to the co-coercive of $R_S(\cdot)$ (see Lemma 3.5 in [12]), we have

$$\sum_{k=1}^{t}(R_S(\boldsymbol{w}_k) - R_S(\boldsymbol{w}_S^*)) \leq \sum_{k=1}^{t} L_1\left(\langle \boldsymbol{w}_{k-1} - \boldsymbol{w}_k, \boldsymbol{w}_{k-1} - \boldsymbol{w}_S^*\rangle - \frac{1}{2}\|\boldsymbol{w}_k - \boldsymbol{w}_{k-1}\|^2\right)$$

$$\overset{a}{\leq} \sum_{k=1}^{t}\frac{L_1}{2}\left(\|\boldsymbol{w}_{k-1} - \boldsymbol{w}_S^*\|^2 - \|\boldsymbol{w}_k - \boldsymbol{w}_S^*\|^2\right) \tag{59}$$

$$\leq \frac{D^2 L_1}{2},$$

where $a$ is due to (58). The descent equation shows

$$R_S(\boldsymbol{w}_t) - R_S(\boldsymbol{w}_S^*) \leq \frac{1}{t}\sum_{k=1}^{t}(R_S(\boldsymbol{w}_k) - R_S(\boldsymbol{w}_S^*)) \leq \frac{D^2 L_1}{2t}. \tag{60}$$

Thus, we get the conclusion. $\qquad\square$

For SGD, the following convergence result holds for the terminal point. This conclusion is Theorem 2 in [64], we give the proof of it to make this paper self-contained.

**Lemma 7.** *Under Assumption 1 and 3,*

$$\mathbb{E}[R_S(\boldsymbol{w}_t) - R_S(\boldsymbol{w}_S^*)] \leq \frac{D(L_1^2 + 2L_0^2)}{2L_1\sqrt{t+1}}(1 + \log(t+1)), \tag{61}$$

*for $\boldsymbol{w}_t$ updated by SGD in (9) with $\eta_t = \frac{D}{L_1\sqrt{t+1}}$.*

*Proof.* By the convexity of $R_S(\cdot)$,

$$\sum_{k=j}^{t}\mathbb{E}\left[(R_S(\boldsymbol{w}_k) - R_S(\boldsymbol{w}))\right] \leq \sum_{k=j}^{t}\mathbb{E}\left[\langle\nabla R_S(\boldsymbol{w}_k), \boldsymbol{w}_k - \boldsymbol{w}\rangle\right]$$

$$\leq \frac{1}{2D}\sum_{k=j}^{t}L_1\sqrt{k+1}\mathbb{E}\left[\|\boldsymbol{w}_k - \boldsymbol{w}\|^2 - \|\boldsymbol{w}_{k+1} - \boldsymbol{w}\|^2 + \frac{D^2}{L_1^2(k+1)}\|\nabla f(\boldsymbol{w}_k, \boldsymbol{z}_{i_k})\|^2\right]$$

$$\leq \frac{\sqrt{j+1}L_1}{2D}\|\boldsymbol{w}_j - \boldsymbol{w}\|^2 + \frac{L_1}{2D}\sum_{k=j+1}^{t}\left(\sqrt{k+1} - \sqrt{k}\right)\|\boldsymbol{w}_k - \boldsymbol{w}\|^2 + \frac{DL_0^2}{2L_1}\sum_{k=j}^{t}\frac{1}{\sqrt{k+1}} \tag{62}$$

$$\leq \frac{\sqrt{j+1}L_1}{2D}\|\boldsymbol{w}_j - \boldsymbol{w}\|^2 + \frac{DL_1}{2}\left(\sqrt{t+1} - \sqrt{j+1}\right) + \frac{DL_0^2}{2L_1}\sum_{k=j}^{t}\frac{1}{\sqrt{k+1}}$$

for any $0 \leq j \leq t$ and $\boldsymbol{w}$, where the second inequality is due to the property of projection. By choosing $\boldsymbol{w} = \boldsymbol{w}_j$, one can see

$$\sum_{k=j}^{t}\mathbb{E}\left[(R_S(\boldsymbol{w}_k) - R_S(\boldsymbol{w}_j))\right] \leq \frac{DL_1}{2}\left(\sqrt{t+1} - \sqrt{j+1}\right) + \frac{DL_0^2}{L_1}(\sqrt{t+1} - \sqrt{j})$$

$$\leq \frac{D(L_1^2 + 2L_0^2)}{2L_1}(\sqrt{t+1} - \sqrt{j}). \tag{63}$$

Here we use the inequality $\sum_{k=j}^{t} 1/\sqrt{k+1} \leq 2(\sqrt{t+1} - \sqrt{j})$. Let $S_j = \frac{1}{t-j+1}\sum_{k=j}^{t}\mathbb{E}[R_S(\boldsymbol{w}_k)]$, we have

$$(t-j)S_{j+1} - (t-j+1)S_j = -\mathbb{E}[R_S(\boldsymbol{w}_j)] \leq -S_j + \frac{D(L_1^2 + 2L_0^2)}{2L_1(t-j+1)}\left(\sqrt{t+1} - \sqrt{j}\right)$$

$$\leq -S_j + \frac{D(L_1^2 + 2L_0^2)}{2L_1(\sqrt{t+1} + \sqrt{j})} \tag{64}$$

$$\leq -S_j + \frac{D(L_1^2 + 2L_0^2)}{2L_1\sqrt{t+1}},$$

which concludes

$$S_{j+1} - S_j \leq \frac{D(L_1^2 + 2L_0^2)}{2L_1(t-j)\sqrt{t+1}}. \tag{65}$$

Thus

$$\mathbb{E}[R_{\boldsymbol{S}}(\boldsymbol{w}_t)] = S_t \leq S_0 + \frac{D(L_1^2 + 2L_0^2)}{2L_1\sqrt{t+1}} \sum_{j=0}^{t-1} \frac{1}{t-j} \leq S_0 + \frac{D(L_1^2 + 2L_0^2)}{2L_1\sqrt{t+1}}(1 + \log(t+1)). \qquad (66)$$

Here we use the inequality $\sum_{k=1}^t 1/k \leq 1 + \log(t+1)$. By taking $\boldsymbol{w} = \boldsymbol{w}_{\boldsymbol{S}}^*, j = 0$ in (62) and dividing $t+1$ in both side of the above equation, we have

$$S_0 - R_{\boldsymbol{S}}(\boldsymbol{w}_{\boldsymbol{S}}^*) \leq \frac{DL_1}{2\sqrt{t+1}} + \frac{DL_0^2}{L_1\sqrt{t+1}} = \frac{D(L_1^2 + 2L_0^2)}{2L_1\sqrt{t+1}}. \qquad (67)$$

Combining this with (66), the proof is completed. $\qquad\square$

In convex optimization, the convergence results are usually on the running average scheme i.e., $\bar{\boldsymbol{w}}_t = (\boldsymbol{w}_0 + \cdots + \boldsymbol{w}_t)/t$, especially for the randomized algorithm [12]. In this case, we can take $\bar{\boldsymbol{w}}_t$ to be the output of the algorithm after $t$ update steps. One can prove the convergence rate of order $\mathcal{O}(1/\sqrt{t})$ for $\bar{\boldsymbol{w}}_t$ from (67). But Lemma 7 gives the nearly optimal convergence result for the terminal point $\boldsymbol{w}_t$ without involving average.

Combining the convergence result of $\bar{\boldsymbol{w}}_t$ and our Theorem 3, we conclude that the expected excess risk of $\bar{\boldsymbol{w}}_t$ obtained by SGD is also upper bounded by $\tilde{\mathcal{O}}\left(t^{-1/4} + n^{-1}\right)$.

## C  Proof in Section 4

### C.1  Generalization Error on Empirical Local Minima

To begin our discussion, we give a proposition to the finiteness of population local minima.

**Proposition 1.** *Let $\boldsymbol{w}_i^*$ and $\boldsymbol{w}_j^*$ be two local minima of $R(\cdot)$. Then $\|\boldsymbol{w}_i^* - \boldsymbol{w}_j^*\| \geq 4\lambda/L_2$.*

*Proof.* Denote $c = \|\boldsymbol{w}_i^* - \boldsymbol{w}_j^*\|$ and define

$$g(t) = \frac{\mathrm{d}}{\mathrm{d}t} R\left(\boldsymbol{v}^* + \frac{t}{c}(\boldsymbol{w}^* - \boldsymbol{v}^*)\right). \qquad (68)$$

Then $g(0) = g(c) = 0$, $g'(0) \geq \lambda$ and $g'(c) \geq \lambda$. By Assumption 1, $g'(\cdot)$ is Liptchitz continuous with Liptchitz constant $L_2$ and hence $g'(t) \geq \lambda - L_2 \min\{t, c-t\}$ for $t \in [0, c]$. Thus

$$0 = \int_0^c g'(t)dt \geq c\lambda - L_2 \int_0^c \min\{t, c-t\}dt = c\lambda - L_2\frac{c^2}{4}, \qquad (69)$$

and this implies $c \geq 4\lambda/L_2$. $\qquad\square$

Due to the parameter space $\mathcal{W} \subseteq \mathbb{R}^d$ is compact set, Heine–Borel Theorem and the above proposition implies that there only exists finite population local minima. The following lemma is needed in the sequel.

**Lemma 8.** *Under Assumption 1, 2, for any local minimum $\boldsymbol{w}_k^*$ of $R(\cdot)$ with $1 \leq k \leq K$ and the two training sets $\boldsymbol{S}$ and $\boldsymbol{S}'$, $\boldsymbol{w}_{\boldsymbol{S},k}^*$ and $\boldsymbol{w}_{\boldsymbol{S}',k}^*$ are empirical local minimum of $R_{\boldsymbol{S}}(\cdot)$ and $R_{\boldsymbol{S}}(\cdot)$ respectively on the event $E_k$, where*

$$E_k = E_{1,k} \bigcap E_{2,k} \qquad (70)$$

*with*

$$E_{1,k} = \left\{ \|\nabla R_{\boldsymbol{S}}(\boldsymbol{w}_k^*)\| < \frac{\lambda^2}{16L_2}, \|\nabla R_{\boldsymbol{S}'}(\boldsymbol{w}_k^*)\| < \frac{\lambda^2}{16L_2} \right\}$$

$$E_{2,k} = \left\{ \|\nabla^2 R_{\boldsymbol{S}}(\boldsymbol{w}_k^*) - \nabla^2 R(\boldsymbol{w}_k^*)\| \leq \frac{\lambda}{4}, \|\nabla^2 R_{\boldsymbol{S}'}(\boldsymbol{w}_k^*) - \nabla^2 R(\boldsymbol{w}_k^*)\| \leq \frac{\lambda}{4} \right\}, \qquad (71)$$

*and*

$$\mathbb{P}\left(E_k^c\right) \leq \frac{512L_0^2L_2^2}{n\lambda^4} + \frac{128L_1^2}{n\lambda^2}\left(5\sqrt{\log d} + \frac{4e\log d}{\sqrt{n}}\right)^2, \qquad (72)$$

*for any $k$.*

*Proof.* First, as in the proof of Lemma 3, we have $\nabla^2 R_{\boldsymbol{S}}(\boldsymbol{w}) \succeq \frac{\lambda}{2}, \nabla^2 R_{\boldsymbol{S}'}(\boldsymbol{w}) \succeq \frac{\lambda}{2}$ for $\boldsymbol{w} \in B_2(\boldsymbol{w}_k^*, \frac{\lambda}{4L_2})$ when the event $E_{2,k}$ holds. This is due to $\boldsymbol{w}_k^*$ is a local minimum of $R(\cdot)$. Then for any $\boldsymbol{w} \in B_2(\boldsymbol{w}_k^*, \frac{\lambda}{4L_2})$ with $\|\boldsymbol{w}\| = \frac{\lambda}{4L_2}$, we have

$$
\begin{aligned}
R_{\boldsymbol{S}}(\boldsymbol{w}) - R_{\boldsymbol{S}}(\boldsymbol{w}_k^*) &\geq \langle \nabla R_{\boldsymbol{S}}(\boldsymbol{w}_k^*), \boldsymbol{w} - \boldsymbol{w}_k^* \rangle + \frac{\lambda}{4} \|\boldsymbol{w} - \boldsymbol{w}_k^*\|^2 \\
&\geq -\|\nabla R_{\boldsymbol{S}}(\boldsymbol{w}_k^*)\|\|\boldsymbol{w} - \boldsymbol{w}_k^*\| + \frac{\lambda}{4}\|\boldsymbol{w} - \boldsymbol{w}_k^*\|^2 \\
&\geq \left( \frac{\lambda}{4}\|\boldsymbol{w} - \boldsymbol{w}_k^*\| - \|\nabla R_{\boldsymbol{S}}(\boldsymbol{w}_k^*)\| \right) \|\boldsymbol{w} - \boldsymbol{w}_k^*\| \\
&= \left( \frac{\lambda^2}{16L_2} - \|\nabla R_{\boldsymbol{S}}(\boldsymbol{w}_k^*)\| \right) \|\boldsymbol{w} - \boldsymbol{w}_k^*\| > 0,
\end{aligned}
\tag{73}
$$

when event $E_k$ holds. Then the function $R_{\boldsymbol{S}}(\cdot)$ has at least one local minimum in the inner of $B_2(\boldsymbol{w}_k^*, \frac{\lambda}{4L_2})$. Remind that

$$
\boldsymbol{w}_{\boldsymbol{S},k}^* = \underset{\boldsymbol{w} \in B_2(\boldsymbol{w}_k^*, \frac{\lambda}{4L_2})}{\arg\min} R_{\boldsymbol{S}}(\boldsymbol{w}),
\tag{74}
$$

then $\boldsymbol{w}_{\boldsymbol{S},k}^*$ is a local minimum of $R_{\boldsymbol{S}}(\cdot)$. Similarly, $\boldsymbol{w}_{\boldsymbol{S}',k}^*$ is a local minimum of $R_{\boldsymbol{S}'}(\cdot)$. Thus we get the conclusion by event probability upper bound (38). $\qquad\square$

This lemma implies that $R_{\boldsymbol{S}}(\cdot)$ is locally strongly convex around those local minima close to population local minima with high probability. Now, we are ready to give the proof of Lemma 1.

### C.1.1 Proof of Lemma 1

**Restate of Lemma 1** *Under Assumption 1 and 4, for $k = 1, \ldots, K$, with probability at least*

$$
1 - \frac{512L_0^2L_2^2}{n\lambda^4} - \frac{128L_1^2}{n\lambda^2}\left( 5\sqrt{\log d} + \frac{4e\log d}{\sqrt{n}} \right)^2,
\tag{75}
$$

$\boldsymbol{w}_{\boldsymbol{S},k}^*$ [9] *is a local minimum of $R_{\boldsymbol{S}}(\cdot)$. Moreover, for such $\boldsymbol{w}_{\boldsymbol{S},k}^*$, we have*

$$
\begin{aligned}
&\left| \mathbb{E}_{\boldsymbol{S}}[R_{\boldsymbol{S}}(\boldsymbol{w}_{\boldsymbol{S},k}^*) - R(\boldsymbol{w}_{\boldsymbol{S},k}^*)] \right| \\
&\leq \frac{8L_0}{n\lambda}\left[ L_0 + \left\{ \frac{64L_0^2L_2^2}{\lambda^3} + \frac{16L_1^2}{\lambda}\left( 5\sqrt{\log d} + \frac{4e\log d}{\sqrt{n}} \right)^2 \right\} \min\left\{ 3D, \frac{3\lambda}{2L_2} \right\} \right].
\end{aligned}
\tag{76}
$$

*Proof.* The first statement of this Theorem follows from Lemma 8. We prove (76) via the stability of the proposed auxiliary sequence in Section 4.1. Let $\mathcal{A}_{0,k}$ on the training set $\boldsymbol{S}$ and $\boldsymbol{S}'$ be the following auxiliary projected gradient descent algorithm that follow the update rule

$$
\begin{aligned}
\boldsymbol{w}_{t+1,k} &= \mathcal{P}_{B_2(\boldsymbol{w}_k^*, \frac{\lambda}{4L_2})}\left( \boldsymbol{w}_{t,k} - \frac{1}{L_1}\nabla R_{\boldsymbol{S}}(\boldsymbol{w}_{t,k}) \right), \\
\boldsymbol{w}_{t+1,k}' &= \mathcal{P}_{B_2(\boldsymbol{w}_k^*, \frac{\lambda}{4L_2})}\left( \boldsymbol{w}_{t,k}' - \frac{1}{L_1}\nabla R_{\boldsymbol{S}'}(\boldsymbol{w}_{t,k}') \right),
\end{aligned}
\tag{77}
$$

start from $\boldsymbol{w}_{0,k} = \boldsymbol{w}_{0,k}' = \boldsymbol{w}_k^*$. Although this sequence is infeasible, the generalization bounds based on the stability of it are valid. First note that

$$
\|\boldsymbol{w}_{t,k} - \boldsymbol{w}_{t,k}'\| \leq \|\boldsymbol{w}_{t,k} - \boldsymbol{w}_{\boldsymbol{S},k}^*\| + \|\boldsymbol{w}_{t,k}' - \boldsymbol{w}_{\boldsymbol{S}',k}^*\| + \|\boldsymbol{w}_{\boldsymbol{S},k}^* - \boldsymbol{w}_{\boldsymbol{S}',k}^*\|.
\tag{78}
$$

If event $E_k$ defined in (70) holds, due to Lemma 8, $\boldsymbol{w}_{\boldsymbol{S},k}^*$ and $\boldsymbol{w}_{\boldsymbol{S}',k}^*$ are respectively empirical local minimum of $R_{\boldsymbol{S}}(\cdot)$ and $R_{\boldsymbol{S}'}(\cdot)$, and the two empirical risk are $\lambda/2$-strongly convex in $B_2(\boldsymbol{w}_k^*, \frac{\lambda}{4L_2})$. As in Lemma 3, we have

$$
\|\boldsymbol{w}_{\boldsymbol{S},k}^* - \boldsymbol{w}_{\boldsymbol{S}',k}^*\| \leq \frac{8L_0}{n\lambda}
\tag{79}
$$

and

$$
\mathbb{P}(E_k^c) \leq \frac{512L_0^2L_2^2}{n\lambda^4} + \frac{128L_1^2}{n\lambda^2}\left( 5\sqrt{\log d} + \frac{4e\log d}{\sqrt{n}} \right)^2.
\tag{80}
$$

---

[9] Please note the definition of $\boldsymbol{w}_{\boldsymbol{S},k}^*$ in (12) which is not necessary to be a local minimum.

By the standard convergence rate of projected gradient descent i.e., Theorem 3.10 in [12], we have

$$\|\boldsymbol{w}_{t,k} - \boldsymbol{w}^*_{\boldsymbol{S},k}\| \leq \exp\left(-\frac{\lambda t}{4L_1}\right)\frac{\lambda}{4L_2}, \tag{81}$$

and

$$\|\boldsymbol{w}'_{t,k} - \boldsymbol{w}^*_{\boldsymbol{S}',k}\| \leq \exp\left(-\frac{\lambda t}{4L_1}\right)\frac{\lambda}{4L_2}. \tag{82}$$

on event $E_k$. Since $\mathcal{A}_{0,k}$ is a deterministic algorithm, similar to the proof of Lemma 3, we see

$$
\begin{aligned}
\epsilon_{\text{stab}}(t) &= \mathbb{E}_{\boldsymbol{S}}\mathbb{E}_{\boldsymbol{S}'}\left[\sup_{\boldsymbol{z}}|f(\boldsymbol{w}_{t,k},\boldsymbol{z}) - f(\boldsymbol{w}'_{t,k},\boldsymbol{z})|\right] \\
&\leq L_0\mathbb{E}_{\boldsymbol{S}}\mathbb{E}_{\boldsymbol{S}'}\left[\|\boldsymbol{w}_{t,k} - \boldsymbol{w}'_{t,k}\|\right] \\
&\leq L_0\left(\frac{8L_0}{n\lambda} + 2\exp\left(-\frac{\lambda t}{4L_1}\right)\frac{\lambda}{4L_2}\right)\mathbb{P}(E_k) + L_0\min\left\{D, \frac{\lambda}{2L_2}\right\}P(E_k^c) \\
&\leq L_0\left(\frac{8L_0}{n\lambda} + \exp\left(-\frac{\lambda t}{4L_1}\right)\frac{\lambda}{2L_2}\right) \\
&\quad + L_0\left\{\frac{512L_0^2L_2^2}{n\lambda^4} + \frac{128L_1^2}{n\lambda^2}\left(5\sqrt{\log d} + \frac{4e\log d}{\sqrt{n}}\right)^2\right\}\min\left\{D, \frac{\lambda}{2L_2}\right\}.
\end{aligned} \tag{83}
$$

Then, according to Theorem 1,

$$|\mathbb{E}[R_{\boldsymbol{S}}(\boldsymbol{w}_{t,k}) - R(\boldsymbol{w}_{t,k})]| \leq \epsilon_{\text{stab}}(t). \tag{84}$$

Because

$$
\begin{aligned}
&|\mathbb{E}[R_{\boldsymbol{S}}(\boldsymbol{w}^*_{\boldsymbol{S},k}) - R(\boldsymbol{w}^*_{\boldsymbol{S},k})] - \mathbb{E}[R_{\boldsymbol{S}}(\boldsymbol{w}_{t,k}) - R(\boldsymbol{w}_{t,k})]| \\
&\leq 2L_0\mathbb{E}\left[\|\boldsymbol{w}_{t,k} - \boldsymbol{w}^*_{\boldsymbol{S},k}\|\right] \\
&\leq L_0\exp\left(-\frac{\lambda t}{4L_1}\right)\frac{\lambda}{2L_2} + L_0\left\{\frac{512L_0^2L_2^2}{n\lambda^4} + \frac{128L_1^2}{n\lambda^2}\left(5\sqrt{\log d} + \frac{4e\log d}{\sqrt{n}}\right)^2\right\}\min\{2D, \frac{\lambda}{L_2}\},
\end{aligned} \tag{85}
$$

we have

$$
\begin{aligned}
|\mathbb{E}[R_{\boldsymbol{S}}(\boldsymbol{w}^*_{\boldsymbol{S},k}) - R(\boldsymbol{w}^*_{\boldsymbol{S},k})]| &\leq L_0\left(\frac{8L_0}{n\lambda} + \exp\left(-\frac{\lambda t}{4L_1}\right)\frac{\lambda}{L_2}\right) \\
&\quad + L_0\left\{\frac{512L_0^2L_2^2}{n\lambda^4} + \frac{128L_1^2}{n\lambda^2}\left(5\sqrt{\log d} + \frac{4e\log d}{\sqrt{n}}\right)^2\right\}\min\left\{3D, \frac{3\lambda}{2L_2}\right\}.
\end{aligned} \tag{86}
$$

Since $t$ is arbitrary, the inequality in the theorem follows by invoking $t \to \infty$. $\qquad\square$

## C.2   No Extra Empirical Local Minima

To justify the statement in the main body of this paper, we need to introduce some definitions and results in random matrix theory. We refer readers to [72] for more details of this topic. Remind that for any deterministic matrix $\boldsymbol{Q}$, $\exp(\boldsymbol{Q})$ is defined as

$$\exp(\boldsymbol{Q}) = \sum_{k=0}^{\infty}\frac{1}{k!}\boldsymbol{Q}^k. \tag{87}$$

Then, for random matrix $\boldsymbol{Q}$, $\mathbb{E}[\exp(\boldsymbol{Q})]$ is defined as

$$\mathbb{E}[\exp(\boldsymbol{Q})] = \sum_{k=0}^{\infty}\frac{1}{k!}\mathbb{E}\boldsymbol{Q}^k. \tag{88}$$

**Definition 3** (Sub-Gaussian random matrix). *A zero-mean symmetric random matrix $\boldsymbol{M} \in \mathbb{R}^{p \times p}$ is Sub-Gaussian with matrix parameters $\boldsymbol{V} \in \mathbb{R}^{p \times p}$ if*

$$\mathbb{E}[\exp(c\boldsymbol{M})] \preceq \exp\left(\frac{c^2\boldsymbol{V}}{2}\right), \tag{89}$$

*for all $c \in \mathbb{R}$.*

Note that when $p = 1$, Definition 3 becomes the definition of sub-Gaussian random variable.

**Lemma 9.** *Let $\theta \in \{-1, +1\}$ be a Rademacher random variable independent of $z$. Under Assumption 1, for any $w \in \mathcal{W}$, $\theta \langle \nabla f(w, z), \nabla R(w) \rangle$ and $\theta \nabla^2 f(w, z)$ are Sub-Gaussian with parameter $L_0^4$ and $L_1^2 I_d$ respectively.*

*Proof.* According to Assumption 1, we have $\|\nabla f(w, z)\| \leq L_0$ and $\|\nabla^2 f(w, z)\| \leq L_1$. Because $\nabla R(w) = \mathbb{E}[\nabla f(w, z)]$, we have $\|\nabla R(w)\| \leq L_0$ and

$$|\langle \nabla f(w, z), \nabla R(w) \rangle| \leq \|\nabla f(w, z)\| \|\nabla R(w)\| \leq L_0^2. \tag{90}$$

Hence

$$
\begin{aligned}
\mathbb{E}[\exp(c\theta \langle \nabla f(w, z), \nabla R(w) \rangle) \mid z] &= \sum_{k=0}^{\infty} \frac{(c \langle \nabla f(w, z), \nabla R(w) \rangle)^k}{k!} \mathbb{E}[\theta^k] \\
&\stackrel{a}{=} \sum_{k=0}^{\infty} \frac{(c \langle \nabla f(w, z), \nabla R(w) \rangle)^{2k}}{2k!} \\
&\leq \sum_{k=0}^{\infty} \frac{(cL_0^2)^{2k}}{2k!} \\
&= \exp\left(\frac{L_0^4 c^2}{2}\right),
\end{aligned}
\tag{91}
$$

where $a$ is due to $\mathbb{E}\theta^k = 0$ for all odd $k$. This implies

$$\mathbb{E}[\exp(c\theta \langle \nabla f(w, z), \nabla R(w) \rangle)] \leq \exp\left(\frac{L_0^4 c^2}{2}\right), \tag{92}$$

then $\theta \langle \nabla f(w, z), \nabla R(w) \rangle$ is Sub-Gaussian with parameter $L_0^4$. Similar arguments can show $\theta \nabla^2 f(w, z)$ is Sub-Gaussian matrix with parameter $L_1^2 I_d$, since $\|\nabla^2 f(w, z)\| \leq L_1$. $\square$

We have the following concentration results for the gradient and Hessian of empirical risk.

**Lemma 10.** *For any $\delta > 0$,*

$$\mathbb{P}\left(\left|\frac{1}{n}\sum_{i=1}^n \langle \nabla f(w, z_i), \nabla R(w) \rangle - \|\nabla R(w)\|^2\right| \geq \delta\right) \leq 2 \exp\left(-\frac{n\delta^2}{8L_0^4}\right), \tag{93}$$

*and*

$$\mathbb{P}\left(\left\|\frac{1}{n}\sum_{i=1}^n \nabla^2 f(w, z_i) - \nabla^2 R(w)\right\| \geq \delta\right) \leq 2d \exp\left(-\frac{n\delta^2}{8L_1^2}\right). \tag{94}$$

*Proof.* Note that $\mathbb{E}[\langle \nabla f(w, z_i), \nabla R(w) \rangle] = \|\nabla R(w)\|^2$ and $\mathbb{E}[\nabla^2 f(w, z_i)] = \nabla^2 R(w)$. According to symmetrization inequality (Proposition 4.1.1 (b) in [72]), for any $c \in \mathbb{R}$

$$\mathbb{E}\left[\exp\left(\left|\frac{c}{n}\sum_{i=1}^n \langle \nabla f(w, z_i), \nabla R(w) \rangle - \|\nabla R(w)\|^2\right|\right)\right] \leq \mathbb{E}\left[\exp\left(\left|\frac{2c}{n}\sum_{i=1}^n \theta_i \langle \nabla f(w, z_i), \nabla R(w) \rangle\right|\right)\right], \tag{95}$$

and

$$
\begin{aligned}
&\mathbb{E}\left[\exp\left(\sup_{\|u\|=1} cu^T \left(\frac{1}{n}\sum_{i=1}^n \nabla^2 f(w, z_i) - \nabla^2 R(w)\right) u\right)\right] \\
&\leq \mathbb{E}\left[\exp\left(\sup_{\|u\|=1} 2cu^T \left(\frac{1}{n}\sum_{i=1}^n \theta_i \nabla^2 f(w, z_i)\right) u\right)\right],
\end{aligned}
\tag{96}
$$

where $\theta_1, \ldots, \theta_n$ are i.i.d. Rademacher random variables independent of $z_1, \ldots, z_n$.

Because $\theta_i \langle \nabla f(w, z_i), \nabla R(w) \rangle$ is Sub-Gaussian with parameter $L_0^4$,

$$
\begin{aligned}
&\mathbb{E}\left[\exp\left(2c\left|\frac{1}{n}\sum_{i=1}^n \theta_i \langle \nabla f(w, z_i) \nabla R(w) \rangle\right|\right)\right] \\
&\leq \mathbb{E}\left[\exp\left(\frac{2c}{n}\sum_{i=1}^n \theta_i \langle \nabla f(w, z_i), \nabla R(w) \rangle\right)\right] + \mathbb{E}\left[\exp\left(-\frac{2c}{n}\sum_{i=1}^n \theta_i \langle \nabla f(w, z_i), \nabla R(w) \rangle\right)\right] \\
&\leq 2 \exp\left(\frac{2L_0^4 c^2}{n}\right).
\end{aligned}
\tag{97}
$$

Thus by Markov's inequality,

$$\mathbb{P}\left(\left|\frac{1}{n}\sum_{i=1}^{n}\langle\nabla f(\boldsymbol{w},\boldsymbol{z}_i),\nabla R(\boldsymbol{w})\rangle-\|\nabla R(\boldsymbol{w})\|^2\right|\geq\delta\right)\leq 2\exp\left(-c\delta+\frac{2L_0^4c^2}{n}\right). \tag{98}$$

Taking $c=n\delta/(4L_0^4)$, the first inequality is full-filled. By the spectral mapping property of the matrix exponential function and Sub-Gaussian property of $\theta_i\nabla^2 f(\boldsymbol{w},\boldsymbol{z}_i)$,

$$\begin{aligned}\mathbb{E}\left[\exp\left(\sup_{\|\boldsymbol{u}\|=1}\boldsymbol{u}^T\left(\frac{2c}{n}\sum_{i=1}^{n}\theta_i\nabla^2 f(\boldsymbol{w},\boldsymbol{z}_i)\right)\boldsymbol{u}\right)\right]=&\mathbb{E}\left[\exp\left(\sigma_{\max}\left(\frac{2c}{n}\sum_{i=1}^{n}\theta_i\nabla^2 f(\boldsymbol{w},\boldsymbol{z}_i)\right)\right)\right]\\=&\mathbb{E}\left[\sigma_{\max}\left(\exp\left(\frac{2c}{n}\sum_{i=1}^{n}\theta_i\nabla^2 f(\boldsymbol{w},\boldsymbol{z}_i)\right)\right)\right]\\\leq&\mathrm{tr}\left\{\mathbb{E}\left[\exp\left(\frac{2c}{n}\sum_{i=1}^{n}\theta_i\nabla^2 f(\boldsymbol{w},\boldsymbol{z}_i)\right)\right]\right\}\\\leq&\mathrm{tr}\left\{\exp\left(\frac{2L_1^2c^2\boldsymbol{I}_d}{n}\right)\right\}\\=&d\exp\left(\frac{2L_1^2c^2}{n}\right).\end{aligned} \tag{99}$$

Thus

$$\begin{aligned}&\mathbb{E}\left[\exp\left(c\left\|\frac{1}{n}\sum_{i=1}^{n}\nabla^2 f(\boldsymbol{w},\boldsymbol{z}_i)-\nabla^2 R(\boldsymbol{w})\right\|\right)\right]\\&\leq\mathbb{E}\left[\exp\left(\sup_{\|\boldsymbol{u}\|=1}\boldsymbol{u}^T\left(\frac{c}{n}\sum_{i=1}^{n}\nabla^2 f(\boldsymbol{w},\boldsymbol{z}_i)-\nabla^2 R(\boldsymbol{w})\right)\boldsymbol{u}\right)\right]\\&+\mathbb{E}\left[\exp\left(\sup_{\|\boldsymbol{u}\|=1}\boldsymbol{u}^T\left(\frac{-c}{n}\sum_{i=1}^{n}\nabla^2 f(\boldsymbol{w},\boldsymbol{z}_i)-\nabla^2 R(\boldsymbol{w})\right)\boldsymbol{u}\right)\right]\\&\leq 2d\exp\left(\frac{2L_1^2c^2}{n}\right).\end{aligned} \tag{100}$$

Again by Markov's inequality

$$\mathbb{P}\left(\left\|\frac{1}{n}\sum_{i=1}^{n}\nabla^2 f(\boldsymbol{w},\boldsymbol{z}_i)-\nabla^2 R(\boldsymbol{w})\right\|\geq\delta\right)\leq 2d\exp\left(-c\delta+\frac{2L_1^2c^2}{n}\right). \tag{101}$$

Taking $c=n\delta/(4L_1^2)$, the second inequality follows. $\qquad\square$

The next lemma establishes Liptchitz property of $\langle\nabla f(\boldsymbol{w},\boldsymbol{z}),\nabla R(\boldsymbol{w})\rangle$ and $\|\nabla R(\boldsymbol{w})\|^2$.

**Lemma 11.** *For any $\boldsymbol{w},\boldsymbol{w}'\in\mathcal{W}$, we have*

$$|\langle\nabla f(\boldsymbol{w},\boldsymbol{z}),\nabla R(\boldsymbol{w})\rangle-\langle\nabla f(\boldsymbol{w}',\boldsymbol{z}),\nabla R(\boldsymbol{w}')\rangle|\leq 2L_0L_1\|\boldsymbol{w}-\boldsymbol{w}'\|, \tag{102}$$

*and*

$$|\|\nabla R(\boldsymbol{w})\|^2-\|\nabla R(\boldsymbol{w}')\|^2|\leq 2L_0L_1\|\boldsymbol{w}-\boldsymbol{w}'\|. \tag{103}$$

*Proof.* We have

$$\begin{aligned}|\langle\nabla f(\boldsymbol{w},\boldsymbol{z}),\nabla R(\boldsymbol{w})\rangle-\langle\nabla f(\boldsymbol{w}',\boldsymbol{z}),\nabla R(\boldsymbol{w}')\rangle|\leq&|\langle\nabla f(\boldsymbol{w},\boldsymbol{z})-\nabla f(\boldsymbol{w}',\boldsymbol{z}),\nabla R(\boldsymbol{w})\rangle|\\&+|\langle\nabla f(\boldsymbol{w}',\boldsymbol{z}),(\nabla R(\boldsymbol{w})-\nabla R(\boldsymbol{w}'))\rangle|\\\leq&2L_0L_1\|\boldsymbol{w}-\boldsymbol{w}'\|,\end{aligned} \tag{104}$$

and

$$|\|\nabla R(\boldsymbol{w})\|^2-\|\nabla R(\boldsymbol{w}')\|^2|=|\langle\nabla R(\boldsymbol{w})-\nabla R(\boldsymbol{w}'),\nabla R(\boldsymbol{w})+\nabla R(\boldsymbol{w}')\rangle|\leq 2L_0L_1\|\boldsymbol{w}-\boldsymbol{w}'\| \tag{105}$$

due to the Lipschitz gradient. Hence we get the conclusion. $\qquad\square$

Now, we are ready to provide the proof of Lemma 2.

### C.2.1 Proof of Lemma 2

**Restate of Lemma 2** *Under Assumption 1 and 4, for $r = \min\left\{\frac{\lambda}{8L_2}, \frac{\alpha^2}{16L_0L_1}\right\}$, with probability at least*

$$
1 - 2\left(\frac{3D}{r}\right)^d \exp\left(-\frac{n\alpha^4}{128L_0^4}\right) - 4d\left(\frac{3D}{r}\right)^d \exp\left(-\frac{n\lambda^2}{128L_1^2}\right)
$$
$$
- K\left\{\frac{512L_0^2L_2^2}{n\lambda^4} + \frac{128L_1^2}{n\lambda^2}\left(5\sqrt{\log d} + \frac{4e\log d}{\sqrt{n}}\right)^2\right\}, \tag{106}
$$

*we have*

   *i: $\mathcal{M}_{\boldsymbol{S}} = \{\boldsymbol{w}_{\boldsymbol{S},1}^*, \ldots, \boldsymbol{w}_{\boldsymbol{S},K}^*\}$;*

   *ii: for any $\boldsymbol{w} \in \mathcal{W}$, if $\|\nabla R_{\boldsymbol{S}}(\boldsymbol{w})\| < \alpha^2/(2L_0)$ and $\nabla^2 R_{\boldsymbol{S}}(\boldsymbol{w}) \succ -\lambda/2$, then $\|\boldsymbol{w} - \mathcal{P}_{\mathcal{M}_{\boldsymbol{S}}}(\boldsymbol{w})\| \le \lambda\|\nabla R_{\boldsymbol{S}}(\boldsymbol{w})\|/4$,*

*where $\nabla^2 R_{\boldsymbol{S}}(\boldsymbol{w}) \succ -\lambda/2$ means $\nabla^2 R_{\boldsymbol{S}}(\boldsymbol{w}) + \lambda/2\boldsymbol{I}_d$ is a positive definite matrix.*

*Proof.* Let

$$
r = \min\left\{\frac{\lambda}{8L_2}, \frac{\alpha^2}{16L_0L_1}\right\}, \tag{107}
$$

then according to the result of covering number of $\ell_2$-ball and covering number is increasing by inclusion (i.e., [81]), there are $N \le (3D/r)^d$ points $\boldsymbol{w}_1, \ldots, \boldsymbol{w}_N \in \mathcal{W}$ such that: $\forall \boldsymbol{w} \in \mathcal{W}$, $\exists j \in \{1, \cdots, N\}$, $\|\boldsymbol{w} - \boldsymbol{w}_j\| \le r$. Then, by Lemma 10 and Bonferroni inequality we have

$$
\mathbb{P}\left(\max_{1 \le j \le N} |\langle R_{\boldsymbol{S}}(\boldsymbol{w}_j), \nabla R(\boldsymbol{w}_j)\rangle - \|\nabla R(\boldsymbol{w}_j)\|^2| \ge \frac{\alpha^2}{4}\right) \le 2\left(\frac{3D}{r}\right)^d \exp\left(-\frac{n\alpha^4}{128L_0^4}\right), \tag{108}
$$

and

$$
\mathbb{P}\left(\max_{1 \le j \le N} \|\nabla^2 R_{\boldsymbol{S}}(\boldsymbol{w}_j) - \nabla^2 R(\boldsymbol{w}_j)\|\right) \le 4d\left(\frac{3D}{r}\right)^d \exp\left(-\frac{n\lambda^2}{128L_1^2}\right). \tag{109}
$$

Define the event

$$
H = \left\{\max_{1 \le j \le N} |\langle \nabla R_{\boldsymbol{S}}(\boldsymbol{w}_j), \nabla R(\boldsymbol{w}_j)\rangle - \|\nabla R(\boldsymbol{w}_j)\|\| \le \frac{\alpha^2}{4},\right.
$$
$$
\max_{1 \le j \le N} \|\nabla^2 R_{\boldsymbol{S}}(\boldsymbol{w}_j) - \nabla^2 R(\boldsymbol{w}_j)\| \le \frac{\lambda}{4}, \tag{110}
$$
$$
\left.\boldsymbol{w}_{\boldsymbol{S},k}^* \text{ is a local minimum of } R_{\boldsymbol{S}}(\cdot), \ k = 1, \ldots, K\right\},
$$

then combining inequalities (75), (108), (109), and Bonferroni inequality, we have

$$
\mathbb{P}(H) \ge 1 - 2\left(\frac{3D}{r}\right)^d \exp\left(-\frac{n\alpha^4}{128L_0^4}\right) - 4d\left(\frac{3D}{r}\right)^d \exp\left(-\frac{n\lambda^2}{128L_1^2}\right)
$$
$$
- K\left\{\frac{512L_0^2L_2^2}{n\lambda^4} + \frac{128L_1^2}{n\lambda^2}\left(5\sqrt{\log d} + \frac{4e\log d}{\sqrt{n}}\right)^2\right\}. \tag{111}
$$

Next, we show that on event $H$, the two statements in Lemma 2 hold. For any $\boldsymbol{w} \in \mathcal{W}$ there is $j \in \{1, \ldots, N\}$ such that $\|\boldsymbol{w} - \boldsymbol{w}_j\| \le r$. When event $H$ holds, due to Lemma 11, we have

$$
\begin{aligned}
\left|\langle \nabla R_{\boldsymbol{S}}(\boldsymbol{w}), \nabla R(\boldsymbol{w})\rangle - \|\nabla R(\boldsymbol{w})\|^2\right| &\le \left|\langle \nabla R_{\boldsymbol{S}}(\boldsymbol{w}_j), \nabla R(\boldsymbol{w}_j)\rangle - \|\nabla R(\boldsymbol{w}_j)\|^2\right| \\
&\quad + |\langle \nabla R_{\boldsymbol{S}}(\boldsymbol{w}), \nabla R(\boldsymbol{w})\rangle - \langle \nabla R_{\boldsymbol{S}}(\boldsymbol{w}_j), \nabla R(\boldsymbol{w}_j)\rangle| \\
&\quad + \left|\|\nabla R(\boldsymbol{w})\|^2 - \|\nabla R(\boldsymbol{w}_j)\|^2\right| \\
&\le \frac{\alpha^2}{4} + \frac{\alpha^2}{8} + \frac{\alpha^2}{8} \\
&= \frac{\alpha^2}{2},
\end{aligned} \tag{112}
$$

and

$$\begin{aligned}
\left\|\nabla^2 R_{\boldsymbol{S}}(\boldsymbol{w}) - \nabla^2 R(\boldsymbol{w})\right\| &\leq \left\|\nabla^2 R_{\boldsymbol{S}}(\boldsymbol{w}_j) - \nabla^2 R(\boldsymbol{w}_j)\right\| \\
&\quad + \left\|\nabla^2 R_{\boldsymbol{S}}(\boldsymbol{w}) - \nabla^2 R_{\boldsymbol{S}}(\boldsymbol{w}_j)\right\| + \left\|\nabla^2 R(\boldsymbol{w}) - \nabla^2 R(\boldsymbol{w}_j)\right\| \\
&\leq \frac{\lambda}{4} + \frac{\lambda}{8} + \frac{\lambda}{8} \\
&= \frac{\lambda}{2}.
\end{aligned} \tag{113}$$

Let $\mathcal{D} = \{\boldsymbol{w} : \|\nabla R(\boldsymbol{w})\| \leq \alpha\}$. According to Lemma 8 in the supplemental file of [49], there exists disjoint open sets $\{\mathcal{D}_k\}_{k=1}^{\infty}$ with $\mathcal{D}_k$ possibly empty for $k \geq K+1$ such that $\mathcal{D} = \cup_{k=1}^{\infty}\mathcal{D}_k$. Moreover $\boldsymbol{w}_k^* \in \mathcal{D}_k$, for $1 \leq k \leq K$ and $\sigma_{\min}(\nabla^2 R(\boldsymbol{w})) \geq \lambda$ for each $\boldsymbol{w} \in \cup_{k=1}^{K}\mathcal{D}_k$ while $\sigma_{\min}(\nabla^2 R(\boldsymbol{w})) \leq -\lambda$ for each $\boldsymbol{w} \in \cup_{k=K+1}^{\infty}\mathcal{D}_k$.

Thus when the event $H$ holds, for $\boldsymbol{w} \in \mathcal{D}^c$, we have

$$\langle \nabla R_{\boldsymbol{S}}(\boldsymbol{w}), \nabla R(\boldsymbol{w})\rangle \geq \frac{\alpha^2}{2}, \tag{114}$$

and thus $\boldsymbol{w}$ is not a critical point of the empirical risk. On the other hand, Weyl's theorem implies

$$|\sigma_{\min}(\nabla^2 R_{\boldsymbol{S}}(\boldsymbol{w})) - \sigma_{\min}(\nabla^2 R(\boldsymbol{w}))| \leq \|\nabla^2 R_{\boldsymbol{S}}(\boldsymbol{w}) - \nabla^2 R(\boldsymbol{w})\| \leq \frac{\lambda}{2}. \tag{115}$$

Hence $\sigma_{\min}(\nabla^2 R_{\boldsymbol{S}}(\boldsymbol{w})) \leq -\lambda/2$ for each $\boldsymbol{w} \in \cup_{k=K+1}^{\infty}\mathcal{D}_k$, and then $\boldsymbol{w}$ is not a empirical local minimum. Moreover, $\sigma_{\min}(\nabla^2 R_{\boldsymbol{S}}(\boldsymbol{w})) \geq \lambda/2$ for each $\boldsymbol{w} \in \cup_{k=1}^{K}\mathcal{D}_k$, thus for $k = 1, \ldots, K$, $R_{\boldsymbol{S}}(\cdot)$ is strongly convex in $\mathcal{D}_k$ and there is at most one local minimum in $\mathcal{D}_k$. Hence when $H$ holds, $R_{\boldsymbol{S}}(\cdot)$ has at most $K$ local minimum point, and $\boldsymbol{w}_{\boldsymbol{S},1}^*, \ldots, \boldsymbol{w}_{\boldsymbol{S},K}^*$ are $K$ distinct local minima. This proves $\mathcal{M}_{\boldsymbol{S}} = \{\boldsymbol{w}_{\boldsymbol{S},1}^*, \ldots, \boldsymbol{w}_{\boldsymbol{S},K}^*\}$. By inequality (114), we have

$$\frac{\alpha^2}{2} \leq \langle \nabla R_{\boldsymbol{S}}(\boldsymbol{w}), \nabla R(\boldsymbol{w})\rangle \leq \|\nabla R_{\boldsymbol{S}}(\boldsymbol{w})\|\|\nabla R(\boldsymbol{w})\| \leq L_0\|\nabla R_{\boldsymbol{S}}(\boldsymbol{w})\| \tag{116}$$

for $\boldsymbol{w} \in \mathcal{D}^c$. Thus if $\|\nabla R_{\boldsymbol{S}}(\boldsymbol{w})\| < \alpha^2/(2L_0)$ and $\nabla^2 R_{\boldsymbol{S}}(\boldsymbol{w}) \succ -\lambda/2$, then $\boldsymbol{w} \in \cup_{k=1}^{K}\mathcal{D}_k$. The second statement of Lemma 2 follows from the fact that $R_{\boldsymbol{S}}(\cdot)$ is $\lambda/2$-strongly convex on each of $\mathcal{D}_k$ for $k = 1, \ldots, K$. $\qquad\square$

## C.3 Proof of Theorem 4

The following is the proof of Theorem 4, it provides upper bound of the expected excess risk of any proper algorithm for non-convex problems that efficiently approximates SOSP. We first introduce the following lemma which is a variant of Lemma 1.

**Lemma 12.** *Under Assumptions 1 and 4*

$$\begin{aligned}
&\mathbb{E}_{\boldsymbol{S}}\left[|R_{\boldsymbol{S}}(\boldsymbol{w}_{\boldsymbol{S},k}^*) - R(\boldsymbol{w}_{\boldsymbol{S},k}^*)|\right] \\
&\leq \frac{2M}{\sqrt{n}} + \frac{8L_0}{n\lambda}\left[L_0 + \left\{\frac{64L_0^2L_2^2}{\lambda^3} + \frac{16L_1^2}{\lambda}\left(5\sqrt{\log d} + \frac{4e\log d}{\sqrt{n}}\right)^2\right\}\min\left\{3D, \frac{3\lambda}{2L_2}\right\}\right].
\end{aligned} \tag{117}$$

*Proof.* For $\boldsymbol{w} \in B_2(\boldsymbol{w}_k^*, \frac{\lambda}{4L_2})$, by Weyl's theorem (Exercise 6.1 in [72]),

$$\sigma_{\min}(\nabla^2 R(\boldsymbol{w})) \geq \sigma_{\min}(\nabla^2 R(\boldsymbol{w}_k^*)) - \|\nabla^2 R(\boldsymbol{w}) - \nabla^2 R(\boldsymbol{w}_k^*)\| \geq \lambda - L_2\|\boldsymbol{w} - \boldsymbol{w}_k^*\| \geq \frac{3\lambda}{4}. \tag{118}$$

Hence $R(\cdot)$ is strongly convex in $B_2(\boldsymbol{w}_k^*, \frac{\lambda}{4L_2})$. Then because $\boldsymbol{w}_k^*$ is a local minimum of $R(\cdot)$, we have

$$\boldsymbol{w}_k^* = \operatorname*{arg\,min}_{\boldsymbol{w} \in B_2(\boldsymbol{w}_k^*, \frac{\lambda}{4L_2})} R(\boldsymbol{w}). \tag{119}$$

Thus $R(\boldsymbol{w}_k^*) \leq R(\boldsymbol{w}_{\boldsymbol{S},k}^*)$ and $R_{\boldsymbol{S}}(\boldsymbol{w}_{\boldsymbol{S},k}^*) \leq R_{\boldsymbol{S}}(\boldsymbol{w}_k^*)$. Then

$$(R_{\boldsymbol{S}}(\boldsymbol{w}_{\boldsymbol{S},k}^*) - R(\boldsymbol{w}_{\boldsymbol{S},k}^*))_+ \leq |R_{\boldsymbol{S}}(\boldsymbol{w}_k^*) - R(\boldsymbol{w}_k^*)|, \tag{120}$$

and

$$\begin{aligned}
\mathbb{E}\left[(R_{\boldsymbol{S}}(\boldsymbol{w}_{\boldsymbol{S},k}^*) - R(\boldsymbol{w}_{\boldsymbol{S},k}^*))_+\right] &\leq \mathbb{E}\left[|R_{\boldsymbol{S}}(\boldsymbol{w}_k^*) - R(\boldsymbol{w}_k^*)|\right] \\
&\overset{a}{\leq} \left(\mathbb{E}\left[(R_{\boldsymbol{S}}(\boldsymbol{w}_k^*) - R(\boldsymbol{w}_k^*))^2\right]\right)^{\frac{1}{2}} \\
&\leq \frac{M}{\sqrt{n}},
\end{aligned} \tag{121}$$

where $a$ is due to Jensen's inequality. Hence

$$
\begin{aligned}
\mathbb{E}\left[|R_{\boldsymbol{S}}(\boldsymbol{w}_{\boldsymbol{S},k}^*) - R(\boldsymbol{w}_{\boldsymbol{S},k}^*)|\right] &= \mathbb{E}\left[(R_{\boldsymbol{S}}(\boldsymbol{w}_{\boldsymbol{S},k}^*) - R(\boldsymbol{w}_{\boldsymbol{S},k}^*))_+\right] + \mathbb{E}\left[(R_{\boldsymbol{S}}(\boldsymbol{w}_{\boldsymbol{S},k}^*) - R(\boldsymbol{w}_{\boldsymbol{S},k}^*))_-\right] \\
&= 2\mathbb{E}\left[(R_{\boldsymbol{S}}(\boldsymbol{w}_{\boldsymbol{S},k}^*) - R(\boldsymbol{w}_{\boldsymbol{S},k}^*))_+\right] - \mathbb{E}\left[R_{\boldsymbol{S}}(\boldsymbol{w}_{\boldsymbol{S},k}^*) - R(\boldsymbol{w}_{\boldsymbol{S},k}^*)\right] \\
&\leq 2\mathbb{E}\left[(R(\boldsymbol{w}_{\boldsymbol{S},k}^*) - R(\boldsymbol{w}_{\boldsymbol{S},k}^*))_+\right] + |\mathbb{E}\left[R_{\boldsymbol{S}}(\boldsymbol{w}_{\boldsymbol{S},k}^*) - R(\boldsymbol{w}_{\boldsymbol{S},k}^*)\right]| \\
&\leq \frac{2M}{\sqrt{n}} + |\mathbb{E}\left[R_{\boldsymbol{S}}(\boldsymbol{w}_{\boldsymbol{S},k}^*) - R(\boldsymbol{w}_{\boldsymbol{S},k}^*)\right]|.
\end{aligned}
\tag{122}
$$

Then (117) follows from (76). $\qquad\square$

Then we are ready to give the proof of Theorem 4.

**Restate of Theorem 4** *Under Assumption 1, 2 and 4, if $\boldsymbol{w}_t$ satisfies (18) and $r$ defined in Lemma 2, by choosing $t$ such that $\zeta(t) < \alpha^2/(2L_0)$ and $\rho(t) < \lambda/2$ we have*

$$
\begin{aligned}
|\mathbb{E}_{\mathcal{A},\boldsymbol{S}}\left[R(\boldsymbol{w}_t) - R_{\boldsymbol{S}}(\boldsymbol{w}_t)\right]| \leq\ & \frac{8L_0}{\lambda}\zeta(t) + 2L_0 D\delta + \frac{2KM}{\sqrt{n}} + \frac{8KL_0^2}{n\lambda} \\
& + \left(L_0 \min\left\{3D, \frac{3\lambda}{2L_2}\right\} + 2M\right)\xi_{n,1} + 2M\xi_{n,2},
\end{aligned}
\tag{123}
$$

*where*

$$
\xi_{n,1} = K\left\{\frac{512L_0^2 L_2^2}{n\lambda^4} + \frac{128L_1^2}{n\lambda^2}\left(5\sqrt{\log d} + \frac{4e\log d}{\sqrt{n}}\right)^2\right\},
\tag{124}
$$

*and*

$$
\xi_{n,2} = 2\left(\frac{3D}{r}\right)^d \exp\left(-\frac{n\alpha^4}{128L_0^4}\right) + 4d\left(\frac{3D}{r}\right)^d \exp\left(-\frac{n\lambda^2}{128L_1^2}\right).
\tag{125}
$$

*If with probability at least $1 - \delta'$ ($\delta'$ can be arbitrary small), $R_{\boldsymbol{S}}(\cdot)$ has no spurious local minimum, then*

$$
\begin{aligned}
|\mathbb{E}_{\mathcal{A},\boldsymbol{S}}\left[R(\boldsymbol{w}_t) - R_{\boldsymbol{S}}(\boldsymbol{w}_t)\right]| \leq\ & \frac{8L_0}{\lambda}\zeta(t) + 2L_0 D\delta + 6M\delta' + \frac{8(K+4)L_0^2}{n\lambda} \\
& + \left(\frac{(K+4)L_0}{K}\min\left\{3D, \frac{3\lambda}{2L_2}\right\} + 6M\right)\xi_{n,1} + 6M\xi_{n,2}.
\end{aligned}
\tag{126}
$$

*Proof.* Remind the event in the proof of Lemma 2

$$
\begin{aligned}
H = \Bigg\{ & \max_{1\leq j\leq N}\|\langle\nabla R_{\boldsymbol{S}}(\boldsymbol{w}_j), \nabla R(\boldsymbol{w}_j)\rangle - \|\nabla R(\boldsymbol{w}_j)\|\| \leq \frac{\alpha^2}{4}, \\
& \max_{1\leq j\leq N}\|\nabla^2 R_{\boldsymbol{S}}(\boldsymbol{w}_j) - \nabla^2 R(\boldsymbol{w}_j)\| \leq \frac{\lambda}{4}, \\
& \boldsymbol{w}_{\boldsymbol{S},k}^* \text{ is a local minimum of } R_{\boldsymbol{S}}(\cdot),\ k = 1,\ldots,K \Bigg\},
\end{aligned}
\tag{127}
$$

We have $\mathbb{P}(H^c) \leq \xi_{n,1} + \xi_{n,2}$, and on the event $H$

i: $\mathcal{M}_{\boldsymbol{S}} = \{\boldsymbol{w}_{\boldsymbol{S},1}^*, \ldots, \boldsymbol{w}_{\boldsymbol{S},K}^*\}$;

ii: For any $\boldsymbol{w} \in \mathcal{W}$, if $\|\nabla R_{\boldsymbol{S}}(\boldsymbol{w})\| < \alpha^2/(2L_0)$ and $\nabla^2 R_{\boldsymbol{S}}(\boldsymbol{w}) \succ -\lambda/2$, then $\|\boldsymbol{w} - \mathcal{P}_{\mathcal{M}_{\boldsymbol{S}}}(\boldsymbol{w})\| \leq \lambda\|\nabla R_{\boldsymbol{S}}(\boldsymbol{w})\|/4$.

By Assumption 1,

$$
\begin{aligned}
|\mathbb{E}\left[R(\boldsymbol{w}_t) - R_{\boldsymbol{S}}(\boldsymbol{w}_t)\right]| &\leq |\mathbb{E}\left[(R(\boldsymbol{w}_t) - R_{\boldsymbol{S}}(\boldsymbol{w}_t))\mathbf{1}_H\right]| + |\mathbb{E}\left[(R(\boldsymbol{w}_t) - R_{\boldsymbol{S}}(\boldsymbol{w}_t))\mathbf{1}_{H^c}\right]| \\
&\leq |\mathbb{E}\left[(R(\boldsymbol{w}_t) - R(\mathcal{P}_{\mathcal{M}_{\boldsymbol{S}}}(\boldsymbol{w}_t)))\mathbf{1}_H\right]| \\
&\quad + |\mathbb{E}\left[(R_{\boldsymbol{S}}(\boldsymbol{w}_t) - R_{\boldsymbol{S}}(\mathcal{P}_{\mathcal{M}_{\boldsymbol{S}}}(\boldsymbol{w}_t)))\mathbf{1}_H\right]| \\
&\quad + |\mathbb{E}\left[(R(\mathcal{P}_{\mathcal{M}_{\boldsymbol{S}}}(\boldsymbol{w}_t)) - R_{\boldsymbol{S}}(\mathcal{P}_{\mathcal{M}_{\boldsymbol{S}}}(\boldsymbol{w}_t)))\mathbf{1}_H\right]| + 2M\mathbb{P}(H^c) \\
&\leq 2L_0\mathbb{E}\left[\|\boldsymbol{w}_t - \mathcal{P}_{\mathcal{M}_{\boldsymbol{S}}}(\boldsymbol{w}_t)\|\mathbf{1}_H\right] \\
&\quad + |\mathbb{E}\left[(R(\mathcal{P}_{\mathcal{M}_{\boldsymbol{S}}}(\boldsymbol{w}_t)) - R_{\boldsymbol{S}}(\mathcal{P}_{\mathcal{M}_{\boldsymbol{S}}}(\boldsymbol{w}_t)))\mathbf{1}_H\right]| + 2M\mathbb{P}(H^c).
\end{aligned}
\tag{128}
$$

Because $\zeta(t) < \alpha^2/(2L_0)$, $\rho(t) < \lambda/2$ and (18), we have on event $H$

$$\mathbb{P}_{\mathcal{A}}(U) \geq 1 - \delta, \tag{129}$$

where

$$U = \left\{ \nabla R_{\boldsymbol{S}}(\boldsymbol{w}_t) < \frac{\alpha^2}{2L_0}, \nabla^2 R_{\boldsymbol{S}}(\boldsymbol{w}_t) \succ -\frac{\lambda}{2} \right\}. \tag{130}$$

Thus we have

$$\mathbb{E}[\|\boldsymbol{w}_t - \mathcal{P}_{\mathcal{M}_{\boldsymbol{S}}}(\boldsymbol{w}_t)\|\mathbf{1}_H] \leq \mathbb{E}[\|\boldsymbol{w}_t - \mathcal{P}_{\mathcal{M}_{\boldsymbol{S}}}(\boldsymbol{w}_t)\|\mathbf{1}_{H \cap U^c}] + \mathbb{E}[\|\boldsymbol{w}_t - \mathcal{P}_{\mathcal{M}_{\boldsymbol{S}}}(\boldsymbol{w}_t)\|\mathbf{1}_{H \cap U}]$$
$$\leq \frac{4}{\lambda}\zeta(t) + D\delta, \tag{131}$$

where the second inequality is due to the property $(ii)$ in Lemma 2 holds on event $H$. According to (117), we have

$$|\mathbb{E}\left[(R(\mathcal{P}_{\mathcal{M}_{\boldsymbol{S}}}(\boldsymbol{w}_t)) - R_{\boldsymbol{S}}(\mathcal{P}_{\mathcal{M}_{\boldsymbol{S}}}(\boldsymbol{w}_t)))\mathbf{1}_H\right]|$$
$$\leq \mathbb{E}\left|[(R(\mathcal{P}_{\mathcal{M}_{\boldsymbol{S}}}(\boldsymbol{w}_t)) - R_{\boldsymbol{S}}(\mathcal{P}_{\mathcal{M}_{\boldsymbol{S}}}(\boldsymbol{w}_t)))\mathbf{1}_H]\right|$$
$$\leq \mathbb{E}\left[\max_{1 \leq k \leq K} |R(\boldsymbol{w}_{\boldsymbol{S},k}^*) - R_{\boldsymbol{S}}(\boldsymbol{w}_{\boldsymbol{S},k}^*)|\right]$$
$$\leq \sum_{k=1}^{K} \mathbb{E}\left[|R(\boldsymbol{w}_{\boldsymbol{S},k}^*) - R_{\boldsymbol{S}}(\boldsymbol{w}_{\boldsymbol{S},k}^*)|\right] \tag{132}$$
$$\leq K\left[\frac{2M}{\sqrt{n}} + \frac{8L_0}{n\lambda}\left[L_0 + \left\{\frac{64L_0^2L_2^2}{\lambda^3} + \frac{16L_1^2}{\lambda}\left(5\sqrt{\log d} + \frac{4e\log d}{\sqrt{n}}\right)^2\right\} \min\left\{3D, \frac{3\lambda}{2L_2}\right\}\right]\right].$$

Combination of equations (128), (131) and (132) completes the proof of (123).

To establish (126), we bound $|\mathbb{E}\left[(R_{\boldsymbol{S}}(\mathcal{P}_{\mathcal{M}_{\boldsymbol{S}}}(\boldsymbol{w}_t)) - R(\mathcal{P}_{\mathcal{M}_{\boldsymbol{S}}}(\boldsymbol{w}_t)))\mathbf{1}_H\right]|$ in a different manner. Remind $\mathcal{M} = \{\boldsymbol{w}_1^*, \cdots, \boldsymbol{w}_K^*\}$ is the set of population local minima. Let

$$G = \{R_{\boldsymbol{S}}(\cdot) \text{ has no spurious local minimum}\}. \tag{133}$$

Then the assumption implies that $\mathbb{P}(G^c) \leq \delta'$. Note that

$$|\mathbb{E}[(R(\mathcal{P}_{\mathcal{M}_{\boldsymbol{S}}}(\boldsymbol{w}_t)) - R_{\boldsymbol{S}}(\mathcal{P}_{\mathcal{M}_{\boldsymbol{S}}}(\boldsymbol{w}_t)))\mathbf{1}_H]| \leq \left|\mathbb{E}[(R(\mathcal{P}_{\mathcal{M}_{\boldsymbol{S}}}(\boldsymbol{w}_t)) - R_{\boldsymbol{S}}(\mathcal{P}_{\mathcal{M}_{\boldsymbol{S}}}(\boldsymbol{w}_t)))\mathbf{1}_{H \cap G}]\right|$$
$$+ \left|\mathbb{E}[(R(\mathcal{P}_{\mathcal{M}_{\boldsymbol{S}}}(\boldsymbol{w}_t)) - R_{\boldsymbol{S}}(\mathcal{P}_{\mathcal{M}_{\boldsymbol{S}}}(\boldsymbol{w}_t)))\mathbf{1}_{H \cap G^c}]\right|$$
$$\leq \left|\mathbb{E}[(R(\mathcal{P}_{\mathcal{M}_{\boldsymbol{S}}}(\boldsymbol{w}_t)) - R_{\boldsymbol{S}}(\mathcal{P}_{\mathcal{M}_{\boldsymbol{S}}}(\boldsymbol{w}_t)))\mathbf{1}_{H \cap G}]\right| + 2M\delta'$$
$$= \left|\mathbb{E}[(R(\mathcal{P}_{\mathcal{M}_{\boldsymbol{S}}}(\boldsymbol{w}_t)) - R_{\boldsymbol{S}}(\boldsymbol{w}_{\boldsymbol{S},1}^*))\mathbf{1}_{H \cap G}]\right| + 2M\delta'$$
$$\leq \left|\mathbb{E}[(R(\mathcal{P}_{\mathcal{M}_{\boldsymbol{S}}}(\boldsymbol{w}_t)) - R_{\boldsymbol{S}}(\boldsymbol{w}_{\boldsymbol{S},1}^*))\mathbf{1}_H]\right| + 4M\delta', \tag{134}$$

where the last inequality is due to $\mathbb{P}(G^c) \leq \delta'$. Moreover, under Assumption 1

$$\left|\mathbb{E}[(R(\mathcal{P}_{\mathcal{M}_{\boldsymbol{S}}}(\boldsymbol{w}_t)) - R_{\boldsymbol{S}}(\boldsymbol{w}_{\boldsymbol{S},1}^*))\mathbf{1}_H]\right| \leq |\mathbb{E}[(R(\mathcal{P}_{\mathcal{M}_{\boldsymbol{S}}}(\boldsymbol{w}_t)) - R(\mathcal{P}_{\mathcal{M}}(\mathcal{P}_{\mathcal{M}_{\boldsymbol{S}}}(\boldsymbol{w}_t))))\mathbf{1}_H]|$$
$$+ |\mathbb{E}[(R(\mathcal{P}_{\mathcal{M}}(\mathcal{P}_{\mathcal{M}_{\boldsymbol{S}}}(\boldsymbol{w}_t))) - R(\boldsymbol{w}_1^*))\mathbf{1}_H]|$$
$$+ \left|\mathbb{E}[(R(\boldsymbol{w}_1^*) - R(\boldsymbol{w}_{\boldsymbol{S},1}^*))\mathbf{1}_H]\right|$$
$$+ \left|\mathbb{E}[(R(\boldsymbol{w}_{\boldsymbol{S},1}^*) - R_{\boldsymbol{S}}(\boldsymbol{w}_{\boldsymbol{S},1}^*))\mathbf{1}_H]\right| \tag{135}$$
$$\leq \left|\mathbb{E}\left[\max_k \{R(\boldsymbol{w}_{\boldsymbol{S},k}^*) - R(\boldsymbol{w}_k^*)\}\mathbf{1}_H\right]\right|$$
$$+ \max_k \{|R(\boldsymbol{w}_k^*) - R(\boldsymbol{w}_1^*)|\} + \left|\mathbb{E}[(R(\boldsymbol{w}_1^*) - R(\boldsymbol{w}_{\boldsymbol{S},1}^*))]\right|$$
$$+ \left|\mathbb{E}[(R(\boldsymbol{w}_{\boldsymbol{S},1}^*) - R_{\boldsymbol{S}}(\boldsymbol{w}_{\boldsymbol{S},1}^*))]\right| + 4M\mathbb{P}(H^c).$$

Due to Proposition 1, $R(\boldsymbol{w}_{\boldsymbol{S},k}^*) - R(\boldsymbol{w}_k^*) \geq 0$, then

$$\left|\mathbb{E}\left[\max_k \{R(\boldsymbol{w}_{\boldsymbol{S},k}^*) - R(\boldsymbol{w}_k^*)\}\mathbf{1}_H\right]\right| \leq \left|\mathbb{E}\left[\sum_{k=1}^{K}(R(\boldsymbol{w}_{\boldsymbol{S},k}^*) - R(\boldsymbol{w}_k^*))\right]\right|$$
$$\leq \sum_{k=1}^{K}\left|\mathbb{E}[(R(\boldsymbol{w}_{\boldsymbol{S},k}^*) - R(\boldsymbol{w}_k^*))]\right|. \tag{136}$$

According to Lemma 1,

$$|\mathbb{E}[R(\boldsymbol{w}_{\boldsymbol{S},k}^*) - R_{\boldsymbol{S}}(\boldsymbol{w}_{\boldsymbol{S},k}^*)]| \le \frac{8L_0}{n\lambda}\left[L_0 + \left\{\frac{64L_0^2L_2^2}{\lambda^3} + \frac{16L_1^2}{\lambda}\left(5\sqrt{\log d} + \frac{4e\log d}{\sqrt{n}}\right)^2\right\}\min\left\{3D, \frac{3\lambda}{2L_2}\right\}\right]$$

$$= \frac{8L_0^2}{n\lambda} + \frac{L_0}{K}\min\left\{3D, \frac{3\lambda}{2L_2}\right\}\xi_{n,1}.$$

(137)

Then

$$\mathbb{E}[R(\boldsymbol{w}_{\boldsymbol{S},k}^*) - R(\boldsymbol{w}_k^*)] = \mathbb{E}[R(\boldsymbol{w}_{\boldsymbol{S},k}^*) - R_{\boldsymbol{S}}(\boldsymbol{w}_{\boldsymbol{S},k}^*)] + \mathbb{E}[R_{\boldsymbol{S}}(\boldsymbol{w}_{\boldsymbol{S},k}^*) - R_{\boldsymbol{S}}(\boldsymbol{w}_k^*)]$$

$$\le \frac{8L_0^2}{n\lambda} + \frac{L_0}{K}\min\left\{3D, \frac{3\lambda}{2L_2}\right\}\xi_{n,1},$$

(138)

where the inequality is due to the definition of $\boldsymbol{w}_{\boldsymbol{S},k}^*$. (134), (135), (137) and (138) together implies

$$|\mathbb{E}[(R(\mathcal{P}_{\mathcal{M}_{\boldsymbol{S}}}(\boldsymbol{w}_t)) - R_{\boldsymbol{S}}(\mathcal{P}_{\mathcal{M}_{\boldsymbol{S}}}(\boldsymbol{w}_t)))\mathbf{1}_H]| \le \frac{8(K+2)L_0^2}{n\lambda} + \frac{(K+2)L_0}{K}\min\left\{3D, \frac{3\lambda}{2L_2}\right\}\xi_{n,1}$$

$$+ \max_k\{|R(\boldsymbol{w}_k^*) - R(\boldsymbol{w}_1^*)|\} + 4M(\delta' + \xi_{n,1} + \xi_{n,2}).$$

(139)

Now we deal with the term $\max_k\{|R(\boldsymbol{w}_k^*) - R(\boldsymbol{w}_1^*)|\}$. Note that

$$|R(\boldsymbol{w}_k^*) - R(\boldsymbol{w}_1^*)| \le |\mathbb{E}[R(\boldsymbol{w}_{\boldsymbol{S},k}^*) - R_{\boldsymbol{S}}(\boldsymbol{w}_k^*)]| + |\mathbb{E}[R(\boldsymbol{w}_{\boldsymbol{S},1}^*) - R_{\boldsymbol{S}}(\boldsymbol{w}_1^*)]|$$

$$+ |\mathbb{E}[R_{\boldsymbol{S}}(\boldsymbol{w}_{\boldsymbol{S},k}^*) - R_{\boldsymbol{S}}(\boldsymbol{w}_{\boldsymbol{S},1}^*)]|$$

$$\le \frac{16L_0^2}{n\lambda} + \frac{2L_0}{K}\min\left\{3D, \frac{3\lambda}{2L_2}\right\}\xi_{n,1} + |\mathbb{E}[R_{\boldsymbol{S}}(\boldsymbol{w}_{\boldsymbol{S},k}^*) - R_{\boldsymbol{S}}(\boldsymbol{w}_{\boldsymbol{S},1}^*)]|.$$

(140)

Because on the event $H \bigcap G$, $R_{\boldsymbol{S}}(\boldsymbol{w}_{\boldsymbol{S},k}^*) - R_{\boldsymbol{S}}(\boldsymbol{w}_{\boldsymbol{S},k}^*) = 0$,

$$|\mathbb{E}[R_{\boldsymbol{S}}(\boldsymbol{w}_{\boldsymbol{S},k}^*) - R_{\boldsymbol{S}}(\boldsymbol{w}_{\boldsymbol{S},1}^*)]| \le 2M(\mathbb{P}(H^c) + \mathbb{P}(G^c)) \le 2M(\xi_{n,1} + \xi_{n,2} + \delta').$$

(141)

Combining (139), (140) and (141), we (126).

$$|\mathbb{E}[(R(\mathcal{P}_{\mathcal{M}_{\boldsymbol{S}}}(\boldsymbol{w}_t)) - R_{\boldsymbol{S}}(\mathcal{P}_{\mathcal{M}_{\boldsymbol{S}}}(\boldsymbol{w}_t)))\mathbf{1}_H]| \le \frac{8(K+4)L_0^2}{n\lambda} + \frac{(K+4)L_0}{K}\min\left\{3D, \frac{3\lambda}{2L_2}\right\}\xi_{n,1}$$

$$+ 6M(\delta' + \xi_{n,1} + \xi_{n,2}).$$

(142)

(128), (131) and (142) implies (126). □

We notice the technique of deriving the order $\tilde{\mathcal{O}}(1/n)$ when empirical risk has no spurious local minima with high probability is very tricky. Because the obstacle is when we derive upper bound of $|\mathbb{E}[(R_{\boldsymbol{S}}(\mathcal{P}_{\mathcal{M}_{\boldsymbol{S}}}(\boldsymbol{w}_t)) - R(\mathcal{P}_{\mathcal{M}_{\boldsymbol{S}}}(\boldsymbol{w}_t)))\mathbf{1}_H]|$, the involved $\mathcal{P}_{\mathcal{M}_{\boldsymbol{S}}}(\boldsymbol{w}_t)$ is related to the proper algorithm, then it is not guaranteed to converge to a specific empirical local minima which makes us can not directly apply Lemma 1. However, if the proper algorithm is guaranteed to find a specific local minima e.g., GD finds the minimal norm solution for over-parameterized neural network, which is called "the implicit regularization of GD" [4], the order of $\tilde{\mathcal{O}}(1/n)$ can be maintained even the assumption on empirical local minima is violated.

### C.4 Proof of Theorem 5

The proof is based on the Lemma 2 in the above section.

**Restate of Theorem 5** *Under Assumption 1, 2 and 4, if $\boldsymbol{w}_t$ satisfies (18), by choosing $t$ in (18) such that $\zeta(t) < \alpha^2/(2L_0)$ and $\rho(t) < \lambda/2$, we have*

$$\mathbb{E}_{\mathcal{A},\boldsymbol{S}}[R(\boldsymbol{w}_t) - R(\boldsymbol{w}^*)] \le \frac{4L_0}{\lambda}\zeta(t) + L_0D\delta + \frac{2KM}{\sqrt{n}}$$

$$+ \frac{8KL_0^2}{n\lambda} + \left(L_0\min\left\{3D, \frac{3\lambda}{2L_2}\right\} + 2M\right)\xi_{n,1} + 2M\xi_{n,2}$$

$$+ \mathbb{E}_{\mathcal{A},\boldsymbol{S}}[R_{\boldsymbol{S}}(\mathcal{P}_{\mathcal{M}_{\boldsymbol{S}}}(\boldsymbol{w}_t)) - R_{\boldsymbol{S}}(\boldsymbol{w}_{\boldsymbol{S}}^*)],$$

(143)

*If with probability at least $1 - \delta'$ ($\delta'$ can be arbitrary small), $R_{\boldsymbol{S}}(\cdot)$ has no spurious local minimum, then*

$$\mathbb{E}_{\mathcal{A},\boldsymbol{S}}[R(\boldsymbol{w}_t) - R(\boldsymbol{w}^*)] \le \frac{4L_0}{\lambda}\zeta(t) + L_0D\delta + 8M\delta' + \frac{8(K+4)L_0^2}{n\lambda}$$

$$+ \left(\frac{(K+4)L_0}{K}\min\left\{3D, \frac{3\lambda}{2L_2}\right\} + 8M\right)\xi_{n,1} + 8M\xi_{n,2},$$

(144)

*where $\xi_{n,1}$ and $\xi_{n,2}$ are defined in Theorem 4, and $\boldsymbol{w}_{\boldsymbol{S}}^*$ is the global minimum of $R_{\boldsymbol{S}}(\cdot)$.*

*Proof.* By Assumption 1 and the relationship $R_{\boldsymbol{S}}(\boldsymbol{w}_{\boldsymbol{S}}^*) \leq R_{\boldsymbol{S}}(\boldsymbol{w}^*)$, we have the following decomposition

$$
\begin{aligned}
\mathbb{E}\left[R(\boldsymbol{w}_t) - R(\boldsymbol{w}^*)\right] &= \mathbb{E}\left[R(\boldsymbol{w}_t) - R_{\boldsymbol{S}}(\boldsymbol{w}^*)\right] \\
&\leq \mathbb{E}[R(\boldsymbol{w}_t) - R_{\boldsymbol{S}}(\boldsymbol{w}_{\boldsymbol{S}}^*)] \\
&\leq |\mathbb{E}\left[(R(\boldsymbol{w}_t) - R_{\boldsymbol{S}}(\boldsymbol{w}_{\boldsymbol{S}}^*))\mathbf{1}_H\right]| + |\mathbb{E}\left[(R(\boldsymbol{w}_t) - R_{\boldsymbol{S}}(\boldsymbol{w}_{\boldsymbol{S}}^*))\mathbf{1}_{H^c}\right]| \\
&\leq |\mathbb{E}[(R(\boldsymbol{w}_t) - R(\mathcal{P}_{\mathcal{M}_{\boldsymbol{S}}}(\boldsymbol{w}_t)))\mathbf{1}_H]| + |\mathbb{E}[(R(\mathcal{P}_{\mathcal{M}_{\boldsymbol{S}}}(\boldsymbol{w}_t)) - R_{\boldsymbol{S}}(\mathcal{P}_{\mathcal{M}_{\boldsymbol{S}}}(\boldsymbol{w}_t)))\mathbf{1}_H]| \\
&\quad + \mathbb{E}[(R_{\boldsymbol{S}}(\mathcal{P}_{\mathcal{M}_{\boldsymbol{S}}}(\boldsymbol{w}_t)) - R_{\boldsymbol{S}}(\boldsymbol{w}_{\boldsymbol{S}}^*))\mathbf{1}_H] + 2M\mathbb{P}(H^c) \\
&\leq L_0\mathbb{E}\left[\|\boldsymbol{w}_t - \mathcal{P}_{\mathcal{M}_{\boldsymbol{S}}}(\boldsymbol{w}_t)\|\mathbf{1}_H\right] + \mathbb{E}[|R(\mathcal{P}_{\mathcal{M}_{\boldsymbol{S}}}(\boldsymbol{w}_t)) - R_{\boldsymbol{S}}(\mathcal{P}_{\mathcal{M}_{\boldsymbol{S}}}(\boldsymbol{w}_t))|\mathbf{1}_H] \\
&\quad + \mathbb{E}[R_{\boldsymbol{S}}(\mathcal{P}_{\mathcal{M}_{\boldsymbol{S}}}(\boldsymbol{w}_t)) - R_{\boldsymbol{S}}(\boldsymbol{w}_{\boldsymbol{S}}^*)] + 2M\mathbb{P}(H^c).
\end{aligned}
\tag{145}
$$

The upper bound of the first and second terms in the last inequality can be easily derived from the proof of Theorem 4 which implies

$$
\begin{aligned}
L_0\mathbb{E}\left[\|\boldsymbol{w}_t - \mathcal{P}_{\mathcal{M}_{\boldsymbol{S}}}(\boldsymbol{w}_t)\|\mathbf{1}_H\right] + \mathbb{E}[|R(\mathcal{P}_{\mathcal{M}_{\boldsymbol{S}}}(\boldsymbol{w}_t)) - R_{\boldsymbol{S}}(\mathcal{P}_{\mathcal{M}_{\boldsymbol{S}}}(\boldsymbol{w}_t))|\mathbf{1}_H] & \\
\leq \frac{4L_0}{\lambda}\zeta(t) + L_0 D\delta + \frac{2KM}{\sqrt{n}} + \frac{8KL_0^2}{n\lambda} + L_0 \min\left\{3D, \frac{3\lambda}{2L_2}\right\}\xi_{n,1}. &
\end{aligned}
\tag{146}
$$

Plugging this into (145), we get (143).

Next, we move on to (144). According to (145),

$$
\begin{aligned}
\mathbb{E}\left[R(\boldsymbol{w}_t) - R(\boldsymbol{w}^*)\right] &= \mathbb{E}\left[R(\boldsymbol{w}_t) - R_{\boldsymbol{S}}(\boldsymbol{w}^*)\right] \\
&\leq \mathbb{E}[R(\boldsymbol{w}_t) - R_{\boldsymbol{S}}(\boldsymbol{w}_{\boldsymbol{S}}^*)] \\
&\leq |\mathbb{E}\left[(R(\boldsymbol{w}_t) - R_{\boldsymbol{S}}(\boldsymbol{w}_{\boldsymbol{S}}^*))\mathbf{1}_H\right]| + |\mathbb{E}\left[(R(\boldsymbol{w}_t) - R_{\boldsymbol{S}}(\boldsymbol{w}_{\boldsymbol{S}}^*))\mathbf{1}_{H^c}\right]| \\
&\leq |\mathbb{E}[(R(\boldsymbol{w}_t) - R(\mathcal{P}_{\mathcal{M}_{\boldsymbol{S}}}(\boldsymbol{w}_t)))\mathbf{1}_H]| \\
&\quad + |\mathbb{E}[(R(\mathcal{P}_{\mathcal{M}_{\boldsymbol{S}}}(\boldsymbol{w}_t)) - R_{\boldsymbol{S}}(\mathcal{P}_{\mathcal{M}_{\boldsymbol{S}}}(\boldsymbol{w}_t)))\mathbf{1}_H]| \\
&\quad + \mathbb{E}[(R_{\boldsymbol{S}}(\mathcal{P}_{\mathcal{M}_{\boldsymbol{S}}}(\boldsymbol{w}_t)) - R_{\boldsymbol{S}}(\boldsymbol{w}_{\boldsymbol{S}}^*))\mathbf{1}_H] + 2M\mathbb{P}(H^c).
\end{aligned}
\tag{147}
$$

According to (131),

$$
|\mathbb{E}[(R(\boldsymbol{w}_t) - R(\mathcal{P}_{\mathcal{M}_{\boldsymbol{S}}}(\boldsymbol{w}_t)))\mathbf{1}_H]| \leq L_0\mathbb{E}\left[\|\boldsymbol{w}_t - \mathcal{P}_{\mathcal{M}_{\boldsymbol{S}}}(\boldsymbol{w}_t)\|\mathbf{1}_H\right] \leq \frac{4L_0}{\lambda}\zeta(t) + L_0 D\delta.
\tag{148}
$$

Moreover,

$$
\begin{aligned}
\mathbb{E}[(R_{\boldsymbol{S}}(\mathcal{P}_{\mathcal{M}_{\boldsymbol{S}}}(\boldsymbol{w}_t)) - R_{\boldsymbol{S}}(\boldsymbol{w}_{\boldsymbol{S}}^*))\mathbf{1}_H] &\leq |\mathbb{E}[(R_{\boldsymbol{S}}(\mathcal{P}_{\mathcal{M}_{\boldsymbol{S}}}(\boldsymbol{w}_t)) - R_{\boldsymbol{S}}(\boldsymbol{w}_{\boldsymbol{S}}^*))\mathbf{1}_{H\bigcap G}]| \\
&\quad + |\mathbb{E}[(R_{\boldsymbol{S}}(\mathcal{P}_{\mathcal{M}_{\boldsymbol{S}}}(\boldsymbol{w}_t)) - R_{\boldsymbol{S}}(\boldsymbol{w}_{\boldsymbol{S}}^*))\mathbf{1}_{H\bigcap G^c}]|.
\end{aligned}
\tag{149}
$$

Because on the event $H\bigcap G$, $R_{\boldsymbol{S}}(\mathcal{P}_{\mathcal{M}_{\boldsymbol{S}}}(\boldsymbol{w}_t)) - R_{\boldsymbol{S}}(\boldsymbol{w}_{\boldsymbol{S}}^*) = 0$, (149) implies

$$
\mathbb{E}[(R_{\boldsymbol{S}}(\mathcal{P}_{\mathcal{M}_{\boldsymbol{S}}}(\boldsymbol{w}_t)) - R_{\boldsymbol{S}}(\boldsymbol{w}_{\boldsymbol{S}}^*))\mathbf{1}_H] \leq 2M\mathbb{P}(G^c) = 2M\delta'.
\tag{150}
$$

(147), (142), (148) and (150) implies (144). $\qquad\square$

# D An Algorithm Approximates the SOSP

For non-convex problems, as we have mentioned in the main body of this paper, we consider proper algorithm that approximates SOSP. Here, we present a detailed discussion to them, and propose such a proper algorithm to make it more concrete.

There are extensive papers about non-convex optimization working on proposing algorithms that approximate SOSP, see [27, 24, 19, 37, 39, 76, 52] for examples. However, to the best of our knowledge, theoretical guarantee of vanilla SGD approximating SOSP remains to be explored, especially for the constrained parameter space. The most related result is Theorem 11 in [27] that projected perturbed noisy gradient descent approximates a $(\epsilon, \sqrt{L_2\epsilon})$-SOSP (The definition of $(\epsilon, \gamma)$-SOSP is in the main body of this paper.) in a computational cost of $\mathcal{O}(\epsilon^{-2})$. Though this result is only applied to equality constraints.

Considering the mismatch of settings between this paper and the existing literatures, we propose a gradient-based method Algorithm 1 inspired by [52] to approximate SOSP for non-convex problems.

**Algorithm 1** Projected Gradient Descent (PGD)

---

**Input:** Parameter space $B_1(\mathbf{0}, 1)$, initial point $\mathbf{w}_0$, learning rate $\eta = \frac{1}{L_1}$, tolerance $\epsilon \leq$ $\min\left\{\frac{8\beta^3 L_2^3}{27 L_1^3}, \frac{27}{64^3 L_2^3}, \frac{\beta}{2}\right\}$,

**for** $t = 0, 1, \cdots$ **do**
  **if** $\|\nabla R_{\mathbf{S}}(\mathbf{w}_t)\| \geq \epsilon$ **then**
    **if** $\mathbf{w}_t \in B_2(\mathbf{0}, 1)$ with $\|\mathbf{w}_t\| = 1$ **then**
      $\mathbf{w}_{t+1} = \left(1 - \frac{\beta}{L_1}\right) \mathbf{w}_t$
    **else**
      $\mathbf{w}_{t+1} = \mathcal{P}_{B_2(\mathbf{0}, 1)}\left(\mathbf{w}_t - \eta \nabla R_{\mathbf{S}}(\mathbf{w}_t)\right)$
    **end if**
  **else**
    **if** $\nabla^2 R_{\mathbf{S}}(\mathbf{w}_t) \preceq -\epsilon^{\frac{1}{3}}$ **then**
      Computed $\mathbf{u}_t \in B_2(\mathbf{0}, 1)$ such that $(\mathbf{u}_t - \mathbf{w}_t)^T \nabla^2 R_{\mathbf{S}}(\mathbf{w}_t)(\mathbf{u}_t - \mathbf{w}_t) \leq -\frac{\beta^2 \epsilon^{\frac{1}{3}}}{8 L_1}$
      $\mathbf{w}_{t+1} = \sigma \mathbf{u}_t + (1 - \sigma)\mathbf{w}_t$ with $\sigma = \frac{3 L_1 \epsilon^{\frac{1}{3}}}{2\beta L_2}$.
    **else**
      Return $\mathbf{w}_{t+1}$
    **end if**
  **end if**
**end for**

---

Without loss of generality, we assume that the convex compact parameter space $\mathcal{W}$ is $B_2(\mathbf{0}, 1)$. The proposed algorithm is conducted under the following assumption which implies that there is no minimum on the boundary of the parameter space $\mathcal{W}$.

**Assumption 5.** *For any* $\mathbf{w} \in B_2(\mathbf{0}, 1)$ *with* $\|\mathbf{w}\| = 1$*, there exists* $L_1 > \beta > 0$ *such that* $\langle \nabla R_{\mathbf{S}}(\mathbf{w}), \mathbf{w} \rangle \geq \beta$.

We have following discussion to the proposed Algorithm 1 before providing its convergence rate. The involved quadratic programming can be efficiently solved under Assumption 4 [58]. In addition, we can find $\mathbf{u}_t$ in Algorithm 1 is because the minimal value of the quadratic loss is $-\beta^2 \epsilon^{1/3}/8 L_1$. The next theorem states the convergence rate of the proposed Algorithm 1.

**Theorem 6.** *Under Assumption 1 and 5, let* $\mathbf{w}_t$ *updated in Algorithm 1, by choosing*

$$\epsilon \leq \min\left\{\frac{8\beta^3 L_2^3}{27 L_1^3}, \frac{27}{64^3 L_2^3}, \frac{\beta}{2}\right\}, \tag{151}$$

*and* $\sigma = 3 L_1 \epsilon^{\frac{1}{3}}/2\beta L_2$*, the algorithm breaks at most*

$$2M \max\left\{\frac{2 L_1}{\epsilon^2}, \frac{256 L_2^2}{9\epsilon}\right\} = \mathcal{O}(\epsilon^{-2}) \tag{152}$$

*number of iterations.*

*Proof.* $\|\nabla R_{\mathbf{S}}(\mathbf{w}_t)\| \geq \epsilon$ holds for two cases.

**Case 1:** If $\mathbf{w}_t \in B_2(\mathbf{0}, 1)$ with $\|\mathbf{w}\| = 1$, then we have

$$
\begin{aligned}
R_{\mathbf{S}}(\mathbf{w}_{t+1}) - R_{\mathbf{S}}(\mathbf{w}_t) &\leq \langle \nabla R_{\mathbf{S}}(\mathbf{w}_t), \mathbf{w}_{t+1} - \mathbf{w}_t \rangle + \frac{L_1}{2}\|\mathbf{w}_{t+1} - \mathbf{w}_t\|^2 \\
&\leq -\frac{\beta^2}{L_1} + \frac{\beta^2}{2 L_1} \\
&= -\frac{\beta^2}{2 L_1} \\
&< -\frac{\epsilon^2}{2 L_1},
\end{aligned} \tag{153}
$$

due to the Assumption 5 and Lispchitz gradient.

**Case 2:** If $\boldsymbol{w}_t \in B_2(\boldsymbol{0}, 1)$ but $\|\boldsymbol{w}_t\| < 1$ then

$$R_{\boldsymbol{S}}(\boldsymbol{w}_{t+1}) - R_{\boldsymbol{S}}(\boldsymbol{w}_t) \le \langle \nabla R_{\boldsymbol{S}}(\boldsymbol{w}_t), \boldsymbol{w}_{t+1} - \boldsymbol{w}_t \rangle + \frac{L_1}{2}\|\boldsymbol{w}_{t+1} - \boldsymbol{w}_t\|^2$$

$$\overset{a}{\le} \left(-L_1 + \frac{L_1}{2}\right)\|\boldsymbol{w}_{t+1} - \boldsymbol{w}_t\|^2 \tag{154}$$

$$= -\frac{L_1}{2}\|\boldsymbol{w}_{t+1} - \boldsymbol{w}_t\|^2.$$

Here $a$ is due to the property of projection. Then, if $\|\boldsymbol{w}_{t+1}\| < 1$, one can immediately verify that

$$R_{\boldsymbol{S}}(\boldsymbol{w}_{t+1}) - R_{\boldsymbol{S}}(\boldsymbol{w}_t) \le -(1/2L_1)\|\nabla R_{\boldsymbol{S}}(\boldsymbol{w}_t)\|^2 \le -\frac{\epsilon^2}{2L_1}. \tag{155}$$

On the other hand, if $\|\boldsymbol{w}_t\| < 1$ while $\|\boldsymbol{w}_{t+1}\| = 1$, descent equation (154) implies $R_{\boldsymbol{S}}(\boldsymbol{w}_{t+1}) - R_{\boldsymbol{S}}(\boldsymbol{w}_t) \le 0$. More importantly, $\boldsymbol{w}_{t+1}$ goes back to the sphere. Then we go back to Case 1. Thus we have

$$R_{\boldsymbol{S}}(\boldsymbol{w}_{t+2}) - R_{\boldsymbol{S}}(\boldsymbol{w}_t) \le R_{\boldsymbol{S}}(\boldsymbol{w}_{t+2}) - R_{\boldsymbol{S}}(\boldsymbol{w}_{t+1}) + R_{\boldsymbol{S}}(\boldsymbol{w}_{t+1}) - R_{\boldsymbol{S}}(\boldsymbol{w}_t) \le -\frac{\epsilon^2}{2L_1} \tag{156}$$

in this situation.

Combining the results in these two cases, we have

$$-2M \le R_{\boldsymbol{S}}(\boldsymbol{w}_{2t}) - R_{\boldsymbol{S}}(\boldsymbol{w}_0) = \sum_{j=1}^{t} R_{\boldsymbol{S}}(\boldsymbol{w}_{2(j)}) - R_{\boldsymbol{S}}(\boldsymbol{w}_{2(j-1)}) \le -\frac{t\epsilon^2}{2L_1}. \tag{157}$$

Thus, $t \le 4L_1 M/\epsilon^2$. Then we can verify that $\boldsymbol{w}_t$ approximates a first-order stationary point in the number of $\mathcal{O}\left(\epsilon^{-2}\right)$ iterations.

On the other hand, when $\|\nabla R_{\boldsymbol{S}}(\boldsymbol{w}_t)\| \le \epsilon \le \beta/2$, we notice that

$$\|\nabla R_{\boldsymbol{S}}(\boldsymbol{w})\| = \|\nabla R_{\boldsymbol{S}}(\boldsymbol{w})\|\|\boldsymbol{w}\| \ge \langle \nabla R_{\boldsymbol{S}}(\boldsymbol{w}), \boldsymbol{w} \rangle \ge \beta, \tag{158}$$

for any $\boldsymbol{w} \in B_2(\boldsymbol{0}, 1)$ with $\|\boldsymbol{w}\| = 1$. Then by Lipschitz gradient, we have

$$\|\boldsymbol{w} - \boldsymbol{w}_t\| \ge \frac{1}{L_1}\|\nabla R_{\boldsymbol{S}}(\boldsymbol{w}) - \nabla R_{\boldsymbol{S}}(\boldsymbol{w}_t)\|$$

$$\ge \frac{1}{L_1}\left(\|\nabla R_{\boldsymbol{S}}(\boldsymbol{w})\| - \|\nabla R_{\boldsymbol{S}}(\boldsymbol{w}_t)\|\right) \tag{159}$$

$$\ge \frac{1}{L_1}(\beta - \epsilon)$$

$$\ge \frac{\beta}{2L_1},$$

for any $\boldsymbol{w}$ satisfies $\|\boldsymbol{w}\| = 1$. Thus we can choose the $\boldsymbol{u}_t$ in Algorithm 1, and $\boldsymbol{u}_t \in B_2(\boldsymbol{0}, 1)$. Then with the Lipschitz Hessian, by taking $\sigma = \frac{3L_1\epsilon^{\frac{1}{3}}}{2\beta L_2}$ and $\epsilon \le \min\left\{\frac{8\beta^3 L_2^3}{27L_1^3}, \frac{27}{64^3 L_2^3}\right\}$,

$$R_{\boldsymbol{S}}(\boldsymbol{w}_{t+1}) - R_{\boldsymbol{S}}(\boldsymbol{w}_t) \le \sigma \langle R_{\boldsymbol{S}}(\boldsymbol{w}_t), \boldsymbol{u}_t - \boldsymbol{w}_t \rangle + \frac{\sigma^2}{2}(\boldsymbol{u}_t - \boldsymbol{w}_t)^T \nabla^2 R_{\boldsymbol{S}}(\boldsymbol{w}_t)(\boldsymbol{u}_t - \boldsymbol{w}_t) + \frac{\sigma^3 L_2}{6}\|\boldsymbol{u}_t - \boldsymbol{w}_t\|^3$$

$$\le \sigma\|R_{\boldsymbol{S}}(\boldsymbol{w}_t)\|\|\boldsymbol{u}_t - \boldsymbol{w}_t\| - \sigma^2\frac{\beta^2 \epsilon^{\frac{1}{3}}}{16L_1^2} + \frac{\sigma^3 L_2}{6}\left(\frac{\beta}{2L_1}\right)^3$$

$$\overset{a}{\le} \sigma\frac{\beta\epsilon}{2L_1} - \sigma^2\frac{\beta^2 \epsilon^{\frac{1}{3}}}{16L_1^2} + \sigma^3\frac{L_2\beta^3}{48L_1^3}$$

$$\le \frac{3\epsilon^{\frac{4}{3}}}{4L_2} - \frac{9\epsilon}{128L_2^2}$$

$$\le -\frac{9\epsilon}{256L_2^2}, \tag{160}$$

where $a$ is from the value of $\boldsymbol{u}_t$, and the last two inequality is due to the choice of $\sigma$ and $\epsilon$. Thus, combining this with (153) and (154), we see the Algorithm break after at most

$$2M \max\left\{\frac{4L_1}{\epsilon^2}, \frac{256L_2^2}{9\epsilon}\right\} = \mathcal{O}(\epsilon^{-2}) \tag{161}$$

iterations. $\qquad\square$

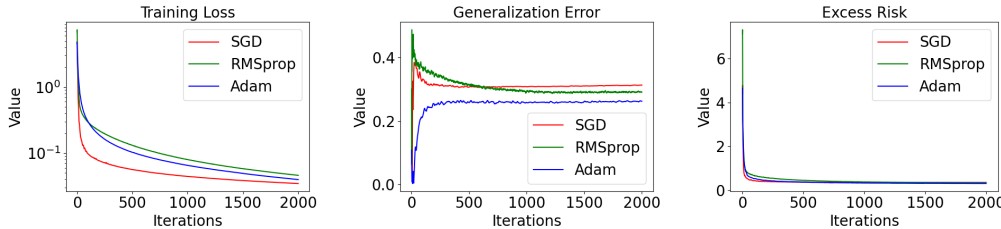

Figure 1: Results of digits dataset under cross entropy loss. From the left to right are respectively training loss, generalization error, and excess risk.

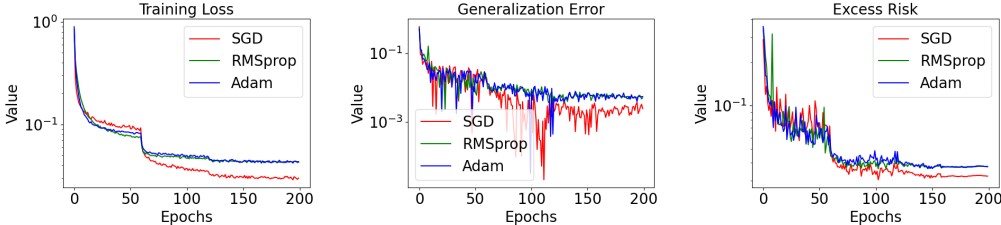

Figure 2: Results of MNIST dataset on LeNet5. From the left to right are respectively training loss, generalization error and excess risk.

From the result, we see that PGD approximates some $(\epsilon, \epsilon^{\frac{1}{3}})$ second-order stationary point at a computational cost of $\mathcal{O}(\epsilon^{-2})$.

## D.1 Excess Risk Under Non-convex problems

We have the following corollary about the expected excess risk of the proposed PGD Algorithm 1. This corollary is proved when we respectively plug $\zeta(t) = \max\left\{2\sqrt{ML_1/t}, 512L_2^2/9t\right\}$, $\rho(t) = \zeta(t)^{\frac{1}{3}}$ and $\delta = 0$ into the Theorem 4.

**Corollary 3.** *Under Assumption 1, 2, 4, and 5. For $t$ satisfies with*

$$\max\left\{2\sqrt{\frac{ML_1}{t}}, \frac{512L_2^2}{9t}\right\} \leq \min\left\{\frac{8\beta^3 L_2^3}{27L_1^3}, \frac{27}{64^3 L_2^3}, \frac{\beta}{2}, \frac{\alpha^2}{2L_0}, \frac{\lambda^3}{8}\right\} \tag{162}$$

*we have*

$$
\begin{aligned}
\min_{1\leq s\leq t}\left|\mathbb{E}_{\mathcal{A},\boldsymbol{S}}\left[R(\boldsymbol{w}_s) - R(\boldsymbol{w}^*)\right]\right| &\leq \frac{2L_0}{\lambda\sqrt{n}} + \frac{4L_0}{\lambda}\max\left\{2\sqrt{\frac{ML_1}{t}}, \frac{512L_2^2}{9t}\right\} + \frac{2KM}{\sqrt{n}} \\
&\quad + \frac{8KL_0^2}{n\lambda} + \left(L_0\min\left\{6, \frac{3\lambda}{2L_2}\right\} + 2M\right)\xi_{n,1} + 2M\xi_{n,2} \\
&\quad + \mathbb{E}_{\mathcal{A},\boldsymbol{S}}[R_{\boldsymbol{S}}(\mathcal{P}_{\mathcal{M}_{\boldsymbol{S}}}(\boldsymbol{w}_t)) - R_{\boldsymbol{S}}(\boldsymbol{w}_{\boldsymbol{S}}^*)].
\end{aligned}
\tag{163}
$$

*where $\boldsymbol{w}_t$ is updated by PGD, $\xi_{n,1}$ and $\xi_{n,2}$ are respectively defined in Theorem 4 with $D = 2$.*

## E Experiments

In this section, we empirically verify our theoretical results in this paper. The experiments are respectively conducted for convex and non-convex problems. We choose SGD [61]; RMSprop [68], and Adam [44] as three proper algorithms which are widely used in the field of machine learning. Since we can not access the exact population risk $R(\boldsymbol{w}_t)$ as well as $\inf_{\boldsymbol{w}} R(\boldsymbol{w})$ during training. Hence, we use the loss on test set to represent the excess risk. Our experiments are conducted on a server with single NVIDIA V100 GPU. All the reported results are the average over five independent runs.

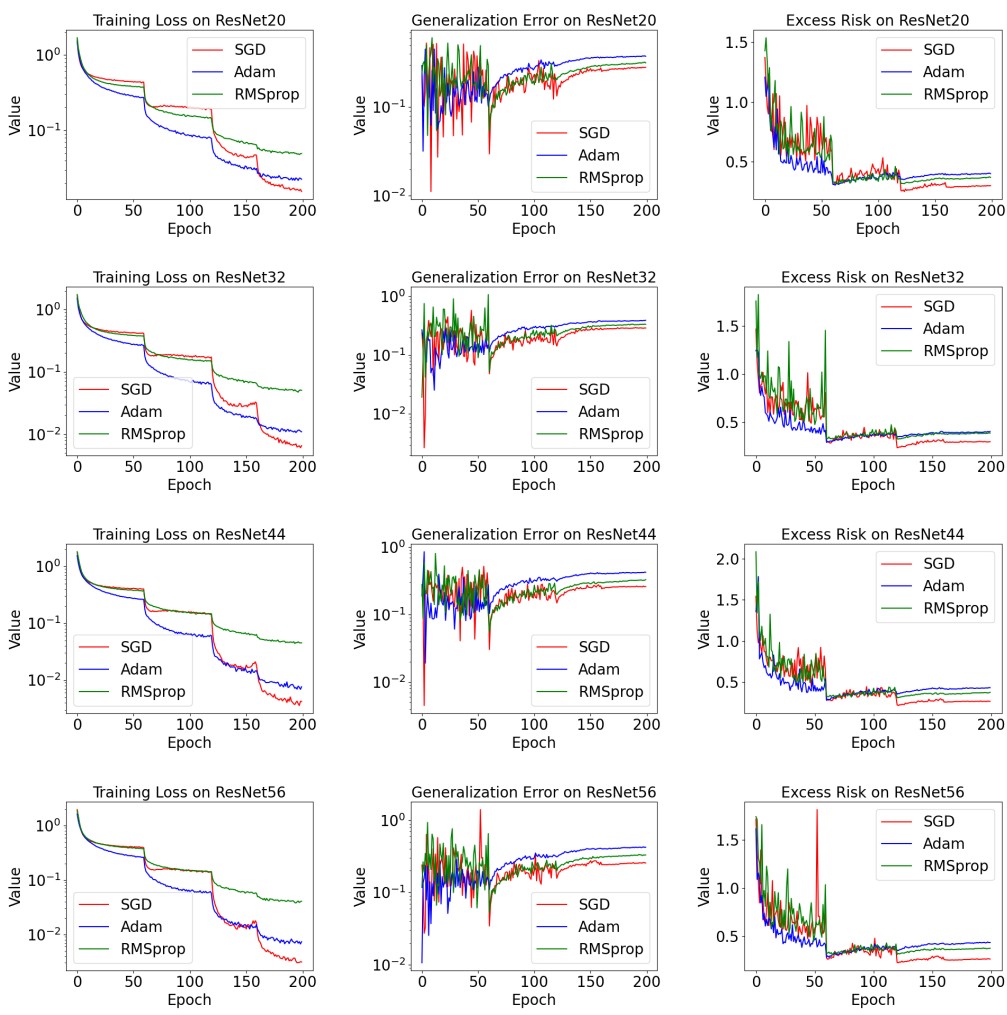

Figure 3: Results of CIFAR10 dataset on various structures of ResNet i.e., $20, 32, 44, 56$. From the left to right are respectively training loss, generalization error and excess risk.

## E.1 convex problems

We conduct the experiments on multi-class logistic regression to verify our results for convex problems. We use the dataset *digits* which is a set with $1800$ samples from $10$ classes. The dataset is available on package *sklearn* [60].

We split $70\%$ data as the training set and the others are used as the test set. We follow the training strategy that all the experiments are conducted for 2000 steps, the learning rates are respectively $0.1$, $0.001$, and $0.001$ for SGD, RMSprop, and Adam. They are decayed with the inverse square root of update steps. The results are summarized in the Figure 1.

From the results, we see that training loss for the three proper algorithms converge close to zero, while the generalization error and excess risk converge to a constant. The observation is consistent with our theoretical conclusion in Section 3.

## E.2 Non-convex problems on Neural Network

For the non-convex problem, we conduct experiments on image classification with various neural network models. Specifically, we use convolutional neural networks LeNet5 [46] and ResNet [33].

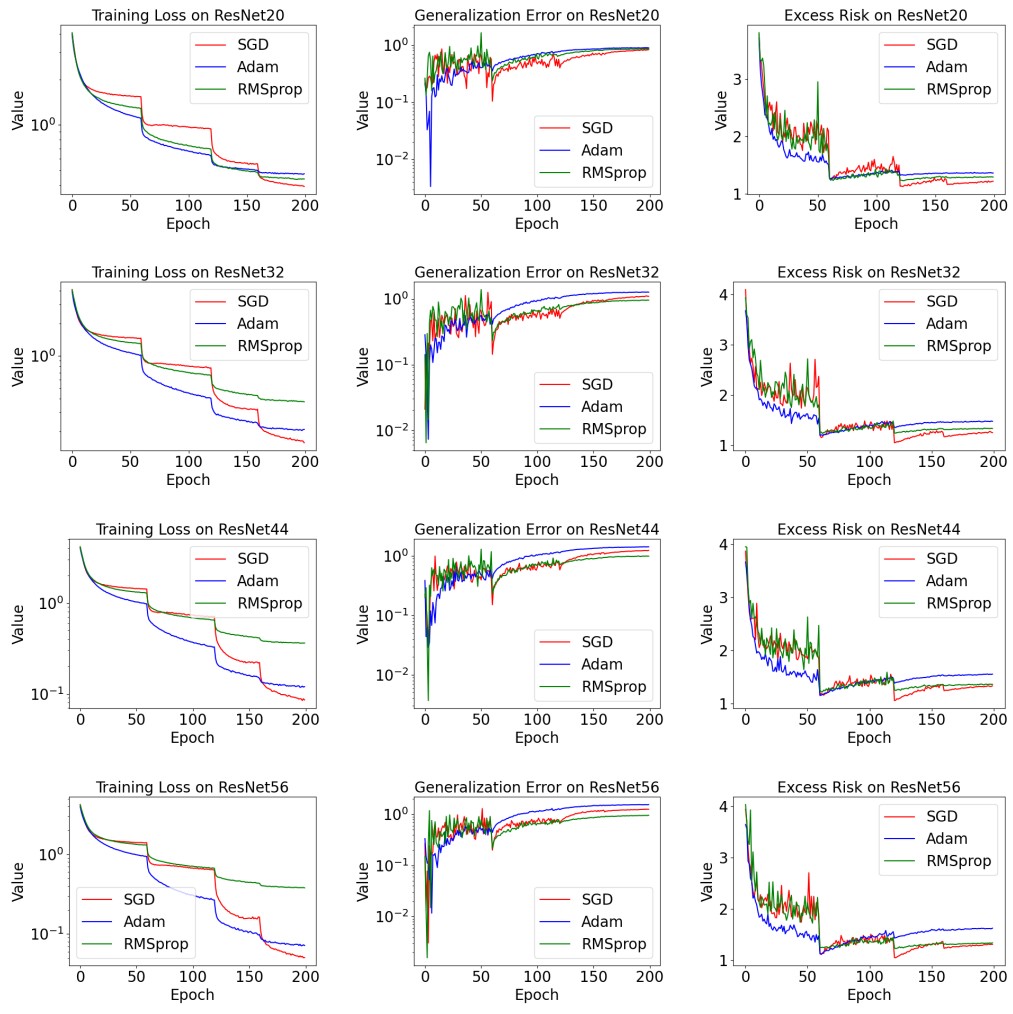

Figure 4: Results of CIFAR100 dataset on various structures of ResNet i.e., $20, 32, 44, 56$. From the left to right are respectively training loss, generalization error and excess risk.

The two structures are widely used in the image classification tasks, and they are leveraged to verify our conclusions for non-convex problems with model parameters in the same order of $n$ and much larger than $n$.

For both structures, we follow the classical training strategy. All the experiments are conducted for 200 epochs with cross entropy loss. The learning rates are set to be $0.1, 0.002, 0.001$ respectively for SGD, RMSprop, and Adam. More ever, the learning rates are decayed by a factor $0.2$ at epoch $60, 120, 160$. We use a uniform batch size $128$ and weight decay $0.0005$.

### E.2.1 Model Parameters in the Same Order of Training Samples

**Data.** The dataset is MNIST [46] which contain binary images of handwritten digits with $50000$ training samples and $10000$ test samples.

**Model.** The model is LeNet5 which is a five layer convolutional neural network with nearly $60,000$ number of parameters.

**Main Results.** The results are summarized in Figure 2. Our code is based on `https://github. com/activatedgeek/LeNet-5`. From the results, we see that the training loss monotonically

decreases with the update steps, while both the generalization error and excess risk tend to converge to some constant. This is consistent with our theoretical results in Section 4.2 when $d$ is in the same order of $n$.

### E.3 Model Parameters Larger than the Order of Training Samples

**Data.** The datasets are CIFAR10 and CIFAR100 [45], which are two benchmark datasets of colorful images both with 50000 training samples, 10000 testing samples but from 10 and 100 object classes respectively.

**Model.** The model we used is ResNet in various depths i.e., $20, 32, 44, 56$. The four structures respectively have nearly $0.27, 0.46, 0.66$, and $0.85$ millions of parameters.

**Main Results.** The experimental results for CIFAR10 and CIFAR100 are respectively in Figure 3 and 4. Our code is based on `https://github.com/kuangliu/pytorch-cifar`. The results show the optimization error, generalization error, and excess risk exhibit similar trends as the results on MNIST dataset. Thus, although our bounds in Section 4 are non-vacuous when $d$ is in the same order of $n$. The empirical verification on the over-parameterized neural network indicates that our results potentially can be applied to the regime of $d \gg n$.

## F   Examples

In this Section, we present three examples satisfies our assumptions imposed in this paper. Let us start with a linear regression problem for convex optimization.

**Example 1** (Linear Regression)**.** *Let* $\boldsymbol{z} = (\boldsymbol{x}, y)$, $y = \boldsymbol{x}^\top \boldsymbol{w}^* + \epsilon$ *for independent noise* $\epsilon$, *and* $f(\boldsymbol{w}, \boldsymbol{z}) = (y - \boldsymbol{w}^\top \boldsymbol{x})^2$.

For any $\boldsymbol{z}$, the quadratic loss $f(\boldsymbol{w}, \boldsymbol{z})$ is convex, and satisfies our smoothness condition Assumption 1. Obviously, when the Hessian of population risk $E[\boldsymbol{x}\boldsymbol{x}^\top]$ is positively definite, the population risk is local (global) strongly convex, thus Assumptions 1, 2, and 3 are satisfied. However, for any instantaneous loss $f(\boldsymbol{w}, z)$ has Hessian of $\boldsymbol{x}\boldsymbol{x}^\top$ which means $f(\boldsymbol{w}, \boldsymbol{z})$ is not necessarily strongly convex with respect to $\boldsymbol{w}$ for any $\boldsymbol{z}$. Thus, we can only treat it as a convex loss function when applying the technique in [32], and get the excess risk bound of order $O(\sqrt{1/n})$. However, the empirical minimizer has a excess risk of order $O(1/n)$ which matches our result. By the way, the technique in [81] also can be applied here, while they require the number of data is sufficiently large, while we do not have such requirement.

The above example has a globally strongly convex population risk, let us consider the following example with locally but not globally strongly convex population risk.

**Example 2** (Robust Regression)**.** *Let* $\boldsymbol{z} = (\boldsymbol{x}, y)$, $y = \boldsymbol{x}^\top \boldsymbol{w}^* + \epsilon$ *for independent noise* $\epsilon$, *and* $f(\boldsymbol{w}, \boldsymbol{z}) = \phi(y - \boldsymbol{w}^\top \boldsymbol{x})$, *with*

$$\phi(u) = \begin{cases} u^2 - \frac{1}{3}u^3 & 0 \le u \le 1, \\ u^2 + \frac{1}{3}u^3 & 0 \le u \le 1, \\ |u| & |u| \ge 1. \end{cases} \tag{164}$$

By computing the gradient and Hessian, one can verify that for any $\boldsymbol{z}$, our robust regression loss $f(\boldsymbol{w}, \boldsymbol{z})$ is convex, and satisfies our smoothness condition Assumption 1. Again, when the matrix $E[\boldsymbol{x}\boldsymbol{x}^\top]$ is positively definite, the population risk of this example is locally but not globally strongly convex. Then the example satisfies our Assumption 1-3. One can also show that the empirical risk minimizer has the generalization bound of order $\mathcal{O}(1/n)$ when $\mathbb{E}[\epsilon^2]$ is small enough. The error also matches our generalization bound in Theorem 2.

Finally, we consider an example of non-convex loss that satisfies our imposed Assumptions 1 and 4.

**Example 3.** *Let* $\boldsymbol{z}_i$ *be mixture Gaussian data such that* $\boldsymbol{z}_i \sim \frac{1}{2}\mathcal{N}(\boldsymbol{w}_1^*, \boldsymbol{I}) + \frac{1}{2}\mathcal{N}(\boldsymbol{w}_2^*, \boldsymbol{I}) = p_{\boldsymbol{w}^*}(\cdot)$. *The maximizing likelihood loss is* $f(\boldsymbol{w}, \boldsymbol{z}) = -\log p_{\boldsymbol{w}}(\boldsymbol{z})$.

By checking the gradient and Hessian, the loss function $f(\boldsymbol{w}, \boldsymbol{z})$ satisfies smoothness Assumption 1. The population risk $R(\boldsymbol{w}) = -E_{\boldsymbol{z} \sim p_{\boldsymbol{w}^*}}[\log p_{\boldsymbol{w}}(\boldsymbol{z})]$, which has two global minima

$(\boldsymbol{w}_1^*, \boldsymbol{w}_2^*), (\boldsymbol{w}_2^*, \boldsymbol{w}_1^*)$, and a saddle point $((\boldsymbol{w}_1^* + \boldsymbol{w}_2^*)/2, (\boldsymbol{w}_1^* + \boldsymbol{w}_2^*)/2)$. Thus, this problem violates the PL-inequality which says that every local minima are global minima. However, by Lemma 16 in [49], we can compute the Hessian to check that the two population global minima are all strict local minima, while the saddle point is strict saddle point. Thus, the example satisfies our Assumptions 1 and 4.