# OpenReview forum: "Characterization of Excess Risk for Locally Strongly Convex Population Risk"
_NeurIPS.cc/2022/Conference — NeurIPS 2022 Accept_

### Official Review · Reviewer_MRcT · 2022-07-10

**Rating:** 7
**Confidence:** 4
**Soundness:** 3 good
**Presentation:** 3 good
**Contribution:** 3 good

**Summary:**

This paper studies the excess risk for stochastic optimization problem where the objective function is locally strongly convex. The roadmap is to decompose the excess risk as the generalization error and the optimization error. Instead of demonstrating algorithm specific generalization bounds for certain algorithms, the paper bounds the generalization error via approximation of empirical problem to the population problem.

**Questions:**

1. In Corollary 1 and 2, why the optimization error of GD for convex problem is of $1/sqrt(t)$ and of SGD for convex problem is $1/t^4$? It contradicts with what is mentioned in Line 184 from reference [12,63]. Which ones are correct? Cor 1 & seems to be far from optimal.

2. Why the paper adopts a new definition for the uniformly stable (Def 2)?

3. In equation (13), it uses an infeasible algorithm to establish the algorithmic stability bound of $w_t$ in order to get the generalization error of $w_{S,k}^*$. I would assume that in (15), the $1/n$ term comes from the stability bound of strongly convex objectives?

4. Why in Theorem 4 and Theorem 5, the generalization bounds depends linearly on the number of local solutions, $K$? Is it from the proof technique?

5. Could the author further explains under what condition there would be only finite number of local solutions? This seems to be a key to Theorem 4 and 5 to hold.





### Minor Comments
1. Line 212, add a bracket for lambda / 4L_2.
2. Line 176, the sentence seems to be incomplete.
3. It would be nice if the author can add some figure in the final version about the roadmap of the analysis idea. It could highlight the importance of this framework.
4. Lemma 1, Theorem 4, 5 could have been more informative if adding some discussions about where does each term come from.

**Ethics Review Area:**

["I don’t know"]

**Limitations:**

The improvements are based on the key assumption that the objective is locally strongly convex. As a result, in the nonconvex case, there is a one-to-one correspondence between the local solution of the empirical problem and the population problem. It remains unclear how to obtain dimension free bounds beyond this regime. Please add some comments if possible.

**Strengths And Weaknesses:**

## Strengths

The main contribution of the paper is a new roadmap to analyze the generalization bounds for stochastic optimization.
1. The obtained generalization bound is dimension insensitive (improving the generalization bounds derived from uniform convergence arguments).
2.  The obtained generalization bound has no restrictions on specific algorithms and particular stepsizes choices (improving over the generalization bounds derived from algorithmic specific stability arguments).
3. The paper identifies a key issue that the generalization bound in [32] will explode when iteration number goes to infinite, which does not align with some of the numerical experiments, and provides a more precise generalization bound characterization for locally strongly convex objectives.

## Weaknesses

See questions below.

---

> ### Author Response · Authors · 2022-07-30
> **To Reviewer MRcT**
>
> We thank you for your valuable comments, and we address your concerns as follows.
>
> Q1: “In Corollary 1 and 2, why the optimization error of GD for the convex problem is of $1 / \sqrt{t}$ and of SGD for the convex problem is $1/t^{4}$? It contradicts with what is mentioned in Line 184 from reference [12,63]. Which ones are correct? Cor 1 & seems to be far from optimal.”
>
> A1: Yes, as you said, the convergence rate of GD and SGD under smooth convex problem is respectively $O(1 / t)$ and $O(1 / \sqrt{t})$. However, our bound of excess risk in equation (7) is of order $\tilde{O}(\sqrt{\epsilon(t)} + 1 / n)$, which matches our results in Corollary 1 and 2.
>
> Q2: “Why the paper adopts a new definition for the uniformly stable (Def 2)?”
>
> A2: As we have clarified in line 119, our uniformly stable in Definition 2 is different from the one in (Hardt et al., 2016), which does not take expectation over training sets $S$ and $S^{\prime}$. We adopt the new definition because our proof of generalization is based on such new uniformly stability. Theoretically speaking, the additional expectation taken over the training set here is used to upper bound the probability of good events as in equation (49) in the Appendix, so that we can upper bound the generalization error.
>
> Q3: “In equation (13), it uses an infeasible algorithm to establish the algorithmic stability bound of $w_{t}$ in order to get the generalization error of $w_{s,K}^{*}$. I would assume that in (15), the $O(1 / n)$ term comes from the stability bound of strongly convex objectives?”
>
> A3: Yes, you are correct. The $O(1 / n)$ is similarly derived as in the convex cases, which is from the stability bound of strongly convex objectives. More details can be found in Appendix C.1.1.
>
> Q4: “Why in Theorem 4 and Theorem 5, the generalization bounds depend linearly on the number of local solutions $K$? Is it from the proof technique?”
>
> A4: Yes, our generalization bound for non-convex problems has a linear dependence on $K$. This is indeed from the proof technique, as we upper bound the probability of the event “no extra local minima” as in Lemma 2. The term related $K$ in equation (17) is obtained from an application of Bonferroni’s inequality to equation (14). More details can be found in equation (111) in Appendix C.2.1. By the way, we have made a discussion to $K$ in line 209 which says that for many important non-convex problems, the $K$ can be quite small.
>
> Q5: ”Could the author further explains under what condition there would be only finite number of local solutions? This seems to be a key to Theorem 4 and 5 to hold.”
>
> A5: As we have clarified in line 205, the Proposition 1 in Appendix C.1 shows that there are only finite population local minima. The proof indicates that under our Assumptions 1 and 2, the non-convex loss has a finite number of local minima.
>
> Q6: “Line 212, add a bracket for $\lambda / 4L_2$, Line 176, the sentence seems to be incomplete.”
>
> A6: Thanks for pointing out these typos, we have fixed them in the revised version.
>
> Q7: “It would be nice if the author can add some figure in the final version about the roadmap of the analysis idea. It could highlight the importance of this framework.”
>
> A7: Thanks for such a valuable comment, we do not add such a figure in the current version due to the limited space. However, we will consider adding it the final version after reorganizing the paper.
>
> Q8: “Lemma 1, Theorem 4, 5 could have been more informative if adding some discussions about where does each term come from.”
>
> A8: Thanks for your suggestion, the discussion of these terms is as follows.
>
> In equation (6), the $\sqrt{\epsilon(t)}$ term is from the distance between iterates and the empirical global minimum. The after $O(1 / n)$ term is from the sum of the probability of “good events” (equation (28)) and the generalization error of strongly convex loss.
>
> In the bound (19) in Theorem 4, the first two terms are from optimization error, the $O(1/\sqrt{n})$ term is from our technique to handle the spurious local minima, and the $O(1 /n) + \xi_{n,1} + \xi_{n,2}$ term is from the probability bound of “no extra local minima” and the generalization error of strongly convex loss.
>
> The bound (22) in Theorem 4 is similar to the one of equation (19), except for there is no $O(1/\sqrt{n})$ term which originates from the spurious local minima.
>
> The terms in bounds appeared in Theorem 5 are similar to the ones in Theorem 4, excepted for the $E_{A, S}[R_{S}(P_{M_{S}}(w_{t})) – R_{S}(w_{S}^{*})]$ in equation (25). This is from the global optimization error for the non-convex problem as we explained in line 305.

---

> > ### Author Response · Authors · 2022-07-30
> > **To Reviewer MRcT**
> >
> > Q9: “The improvements are based on the key assumption that the objective is locally strongly convex. As a result, in the nonconvex case, there is a one-to-one correspondence between the local solution of the empirical problem and the population problem. It remains unclear how to obtain dimension-free bounds beyond this regime. Please add some comments if possible.”
> >
> > A9: Obtaining dimension-free generalization bound for the non-convex problem without some nice regularity conditions (e.g., PL-inequality or locally strongly convex) is a very hard problem in statistical learning theory. Practically, these nice properties only hold for the population risk, as we assumed in this paper. Usually, we should generalize these nice properties to empirical risk which is a key step to obtaining a generalization bound. However, during this step, it usually involves taking a union probability over the whole parameter space which is inevitably related to the dimension of parameter space. Hence, the dimension-free bound seems to be inaccessible in this regime. However, if such nice properties are directly held for empirical risk, the dimension-free bound has been obtained, e.g. in Theorem 1 in (Goen et al., 2017).
> >
> > Ref:
> > Hardt et al., 2016, Train Faster Generalize Better.
> > Goen et al., 2017, Fast Rates for Empirical Risk Minimization of Strict Saddle Problems.

---

> > > ### Comment · Reviewer_MRcT · 2022-08-09
> > > **Thanks for the explanation**
> > >
> > > I would like to thank the authors for the detailed explanation. I have no further questions.

---

### Official Review · Reviewer_7m5U · 2022-07-12

**Rating:** 5
**Confidence:** 3
**Soundness:** 3 good
**Presentation:** 2 fair
**Contribution:** 2 fair

**Summary:**

This paper derives improved generalization and excess risk bounds for convex and non-convex problems with the local strong convexity assumption. In the convex case, previous results had the bound $O(1/\sqrt{n})$ with just convexity and $O(1/n)$ with strong convexity or global strong-convexity type conditions, where $n$ is the number of data points. This paper obtains $\tilde{O}(1/n)$ with local strong convexity. In the nonconvex case, the authors derive when the number of model parameters $d$ is in a certain range and when empirical risk has no spurious local minima, the authors show $\tilde{O}(1/n)$. Without the latter assumption in the nonconvex case, the authors show the bound $O(1/\sqrt{n})$.

**Questions:**

- After Assumption 2, the authors state this assumption implies the population risk is locally strongly convex, however I cannot find this rigorously proved. Please give a reference or proof. eq. (32) in the proof seems to use strong convexity directly, please show how this follows from Assumption 2 with the region $S_2(w^*, \lambda/(4L^2))$ used in eq. (32).

- line 108: Please put the definition of excess risk in a display formula, not an inline one since the paper is about bounding this quantity.

- line 12: what does dimensional insensitive mean? Line 304 seems to have an exponential term which can dominate if $d$ is too large so that $d/n\geq c_1$. What happens if $d$ is too large to make the exponent positive here?

- line 28, 29: the discussion here with the previous work seems rather single-sided. It gives the impression that the authors strictly improve the previous work under the same assumptions, which is not true. There is the additional local strong convexity assumption. Please also discuss limitations of your approach.

- line 70, 71: this sentence seems broken, please consider rewriting.

- line 87, 88: this sentence seems broken, please consider rewriting.

- line 110: what is random in $\inf_w R(w)$? Why does expectation involve this term?

- Thm 1 takes expectation over $S, S'$ instead of the supremum in previous work, in the proof of this theorem, please explain how the second inequality from the last follow?

- Isn't Theorem 2 supposed to hold with high probability? Is this statement correct?

- line 155, 176: The way that the authors talk about their "proof technique" is very vague. Can you please provide pointers to the proof or explain better what in the proof technique results in this behavior so that your readers can understand your proof technique?

- Paragraph starting with line 168: What happens when global strong convexity holds? Does this paper recover the existing best-known dependence?

- Corollary 2: How does this bound compare to [32]? It seems when $T=n$, the bound in [32] can be better. Please provide a full comparison and highlight in which regime the bound in this paper is better and why this regime is important.

- Lemma 1: How to compute the iterate $w^*_{S, k}$ since the guarantee is on this iterate? The definition on eq. (12) requires knowing $w_k^*$ which are local minima. Is there a way to get a result on a realistic iterate that does not require knowledge of local minima?

- line 232, 233: please provide clear examples and clearly show why they satisfy the assumptions made in the paper.

- Theorem 4, 5 gives the result for $w_t$ whereas  Lemma 1 required iterates as eq. (12). Is the sequence $w_t$ generated as eq. (13)? If so, why is this feasible? If not, how is it possible to remove this requirement since Lemma 1 requires iterates of the form eq. (12)?

**Limitations:**

Limitations are not discussed in the paper. The authors always talk about how the results are better than the existing work. Authors need to clarify if it really is the case that there is no limitations on this result?

**Strengths And Weaknesses:**

Originality/significance: To my knowledge, the paper's results are capture a regime that was not done in the previous work. The authors show that local strong convexity is enough to improve the bound of excess risk from $O(1/\sqrt{n})$ to $\tilde{O}(1/n)$. The authors seem to have done a good job if citing the previous work and highlighting the differences of their result. On the other hand, I must say that the authors didn't discuss any limitations of their approach. The writing of the paper suggests that the paper gets a free lunch, which is not the case from what I understand. The analysis seems to be significantly more involved than the previous work such as [32] and the improvement in the convex case for example is not strict, the additional local strong convexity assumption is needed. The authors cite several papers in numerous places in the paper for when this assumption holds (for example line 100), however no precise application is given. Can the authors clearly describe some applications and clearly show how they satisfy their assumption?


Clarity: The paper is provided mostly clear comparisons with the related works. However, the presentation should be improved since there are many unclear parts that I included in the "Questions" section of my review.

---

> ### Author Response · Authors · 2022-07-30
> **To Reviewer 7m5U**
>
> We thank you for your valuable comments, and we address your concerns as follows.
>
> Q1: “I must say that the authors didn't discuss any limitations of their approach. The writing of the paper suggests that the paper gets a free lunch, which is not the case from what I understand.”
>
> A1: Our paper has limitations, and we have clarified them in the paper. For convex loss function, as in the line 168, under the $\textbf{extra}$ local strong convexity assumption, our result improves the bound of order $O(1 / \sqrt{n})$. The bound also matches (in order) the result under the global strongly convex loss function. In addition, for the non-convex case, we require the number of data to be in the same order of the parameter dimension to obtain our bound. The imposed local strong convexity assumption indicates that we do not get a free lunch.
>
> Q2: “The analysis seems to be significantly more involved than the previous work such as [32] and the improvement in the convex case for example is not strict, the additional local strong convexity assumption is needed.”
>
> A2: Regarding our improvements under convex loss function, compared to the results in (Hardt et al., 2016), as we have clarified in lines 68-74, under the locally strongly convex assumption, our improvements include the order of upper bounds, the dependence to the number of iterations and algorithms. To obtain these better results, we think it is reasonable under an extra mild assumption. Besides that, compared to the results in (Hardt et al., 2016) under the strongly convex case, our results (in the same order) are obtained under the weaker locally strongly convex condition.
>
> Regarding the improvements under the non-convex loss function, our results also significantly improve the existing results as summarized in line 278.
>
> Q3: “The authors cite several papers in numerous places in the paper for when this assumption holds (for example line 100), however, no precise application is given. Can the authors clearly describe some applications and clearly show how they satisfy their assumption?”
>
> A3: Thank you for pointing out this, we have three examples that satisfy our assumptions in Appendix F in the revised version. The three examples include linear regression, robust regression, and the Gaussian mixture model.
>
> Q4: “After Assumption 2, the authors state this assumption implies the population risk is locally strongly convex, however I cannot find this rigorously proved. Please give a reference or proof.”
>
> A4: As we have clarified in Assumption 1 (smoothness of Hessian) and 2 together imply locally strongly convex. First, if the Hessian of the loss function is positive definite at point $w_{0}$, then the loss function is strongly convex at $w_{0}$ (Bubeck, 2014). Under the assumption that $\nabla^{2} R(w_{0}) \succeq \lambda$ for some specific $w_{0}$, due to the smoothness Assumption 1 of $\nabla^{2} R(w)$, $\nabla^{2} R(w) \succeq \lambda / 2$ for any $w$ in a neighborhood of $w_{0}$.
>
> Q5: “eq. (32) in the proof seems to use strong convexity directly, please show how this follows from Assumption 2 with the region $S_{2}(w^{*}, \lambda / (4L^{2}))$ used in eq. (32).”
>
> A5: We have proved before equation (32) that when good events $E_{1}$ and $E_{2}$ happen (which happens with high probability), the empirical risk is strongly convex in the region $B_{2}(w^{*}, \lambda / (4L^{2}))$. Thus we the equation (32) holds.
>
> Q6: “line 108: Please put the definition of excess risk in a display formula, not an inline one since the paper is about bounding this quantity.
>
> A6: It has been defined in a display formula in the revised version.
>
> Q7: “line 12: what does dimensional insensitive mean? Line 304 seems to have an exponential term which can dominate if $d$ is too large so that $d/n\geq c_{1}$. What happens if $d$ is too large to make the exponent positive here?”
>
> A7: As we have clarified in line 61, the existing classical result is of order $O(\sqrt{d/n})$ which has polynomial dependence on $d$. However, in our result, if $d$ and $n$ are in the same order, our result is of order $O(\sqrt{1/n})$ or $O(1/n)$ (without spurious local minima), which has no dependence on $d$. Thus, dimensional insensitive compared with existing results. If $d$ is too large, the exponent positive will explode. However, to the best of our knowledge, completely dimension-free generalization has not appeared, without imposing some nice conditions e.g., PL-inequality. More discussions on the dimension of $d$ refer to A9 to Reviewer MRcT.

---

> > ### Author Response · Authors · 2022-07-30
> > **To Reviewer 7m5U Part II**
> >
> > Q8: “line 28, 29: the discussion here with the previous work seems rather single-sided. It gives the impression that the authors strictly improve the previous work under the same assumptions, which is not true. There is the additional local strong convexity assumption. Please also discuss limitations of your approach.”
> >
> > A8: We do not present any of our results here, the discussion is used to clarify the underlying problems of existing results. Our motivation is to fix these issues. We have clearly clarified our results are built upon the local strong convexity assumption in many places, e.g., lines 32, 68, and 71.
> >
> > Q9: “line 70, 71: this sentence seems broken, please consider rewriting.”
> >
> > A9: It has been revised in the new version.
> >
> > Q10: “line 87, 88: this sentence seems broken, please consider rewriting.”
> >
> > A10: It has been revised in the new version.
> >
> > Q11: “line 110: what is random in $\inf_{w}R(w)$? Why does expectation involve this term?”
> >
> > A11:  Please notice there is a bracket after it, the randomness is over $R(A(S)) - \inf_{w}R(w)$, i.e., E[R(A(S)) - \inf_{w}R(w)], which measures the average performance gap between the obtained $A(S)$ and the optimal one.
> >
> > Q12: “Thm 1 takes expectation over $S, S^{\prime}$ instead of the supremum in previous work, in the proof of this theorem, please explain how the second inequality from the last follow?”
> >
> > A12: Please check the notation of our uniform stability, we take expectation on data $S, S^{\prime}$ after taking supremum. For the last inequality in the proof of Theorem, the first “$\leq $” is due to $\{z_{i}\}$ and $\{z_{i}^{\prime}\}$ are all i.i.d. samples. The second “$\leq$ is due to the definition of uniform stability in Definition 1.
> >
> > Q13: “Isn't Theorem 2 supposed to hold with high probability? Is this statement correct?”
> >
> > A13: The $\epsilon_{stab}$ is the uniform stability that takes expectation over both data and algorithm. Thus, it has no randomness, which says Theorem 2 does not suppose to hold with high probability.
> >
> > Q14: “line 155, 176: The way that the authors talk about their "proof technique" is very vague. Can you please provide pointers to the proof or explain better what in the proof technique results in this behavior so that your readers can understand your proof technique?”
> >
> > A14: In (155), we say convergence rate related $\sqrt{\epsilon(t)}$ term originates from our proof technique, as first we upper bound the generalization on empirical global minima, then we upper bound the distance between iterates and empirical global minima via the convergence rate.
> >
> > In line 176, the footnote 2 means that when applying Markov’s inequality in equation (39), the dependence of $\lambda$ can be weaken to $1/\lambda^{2}$ via applying Markov’s inequality to $||\nabla R_{S}(w^{\*})||$ instead of $||\nabla R_{S}(w^{\*})||^{2}$. Concretely, the equation (39)
> >
> > $P(||\nabla R_{S}(w^{\*})||\geq \lambda^{2} / (16L_{2}) ) \leq \frac{256L_{2}^{2}}{\lambda^{4}}E[||\nabla R_{S}(w^{\*})||^{2}] \leq \frac{256L_{0}^{2}L_{2}^{2}}{n\lambda^{4}}$
> >
> > is changed as
> >
> > $P(||\nabla R_{S}(w^{\*})||\geq \lambda^{2} / (16L_{2}) ) \leq \frac{16L_{2}}{\lambda^{2}}E[||\nabla R_{S}(w^{\*})||] \leq \frac{16L_{2}}{\lambda^{2}}E[||\nabla R_{S}(w^{\*})||^{2}]^{1/2} \leq \frac{16L_{0}L_{2}}{\sqrt{n}\lambda^{2}}$.
> >
> > This results in a generalization bound of order $O(1 / \sqrt{n})$.
> >
> > Q15: “Paragraph starting with line 168: What happens when global strong convexity holds? Does this paper recover the existing best-known dependence?”
> >
> > A15: As we clarified in line 169, our bound $\tilde{O}(1 / n)$ under local strong convexity matches (in order) the one under global strong convexity. The $O(1/n)$ bound is proven to be optimal for globally strongly convex loss (Shalev et al., 2009).
> >
> > Q16: “Corollary 2: How does this bound compare to [32]? It seems when $T=n$, the bound in [32] can be better. Please provide a full comparison and highlight in which regime the bound in this paper is better and why this regime is important.”
> >
> > A16: In equation (5.1) of (Hardt et al., 2016), when $T = n$, their excess risk is of order $O(\sqrt{1/n})$ which is in the same order of than our bound $\tilde{O}(1 /\sqrt{t} + 1 / n)$in equation (10). However, the $T=n$ here means conducting training in only one epoch, which obviously violates the practical training strategy. Thus when $T \geq n$ their bound will explode while ours will converge to $\tilde{O}(1 / n)$. This has been discussed in line 139.

---

> > > ### Author Response · Authors · 2022-07-30
> > > **To Reviewer 7m5U Part III**
> > >
> > > Q17: “Lemma 1: How to compute the iterate $w_{S,k}^{\*}$ since the guarantee is on this iterate? The definition of eq. (12) requires knowing $w_{k}^{\*}$ which are local minima. Is there a way to get a result on a realistic iterate that does not require knowledge of local minima?”
> > >
> > > A17: We do not need to get any of $w_{S,k}^{\*}$ or $w_{k}^{\*}$. As we have clarified in line 214, the stability bound in Theorem 1 can be applied to any infeasible algorithm. Thus, we construct the infeasible auxiliary iterates in equation (13) to derive the generalization error on $w_{S,k}^{\*}$ via bounding the stability of the auxiliary iterates. This is a purely theoretical analysis, and does not require knowing any of $w_{S,k}^{\*}$ or $w_{k}^{\*}$.
> > >
> > > Q18: “line 232, 233: please provide clear examples and clearly show why they satisfy the assumptions made in the paper.”
> > >
> > > A18: We give the examples in Appendix F of the revised version.
> > >
> > > Q19: “Theorem 4, 5 gives the result for $w_{t}$ whereas Lemma 1 required iterates as eq. (12). Is the sequence $w_{t}$ generated as eq. (13)? If so, why is this feasible? If not, how is it possible to remove this requirement since Lemma 1 requires iterates of the form eq. (12)?”
> > >
> > > A19: As we have clarified in A18 and the 216 in our paper, the auxiliary iterates in (13) are used to get the generalization bound on $w_{S,k}^{\*}$ via analyzing its stability. $w_{t}$ is not generated as eq. (13). It can be the iterations of any proper algorithm.
> > >
> > > Q20: “Limitations are not discussed in the paper. The authors always talk about how the results are better than the existing work. Authors need to clarify if it really is the case that there is no limitations on this result?”
> > >
> > > A20: We do not claim that our results have no limitation. We have stated that our results are obtained under local strong convexity, it has appeared in our title. The comparisons with existing results are clearly stated in the paper, including the order and conditions.
> > >
> > > Ref:
> > > Hardt et al., 2016. Train Faster Generalize Better.
> > > Shalev et al., 2009. Stochastic convex optimization.

---

> > > > ### Comment · Reviewer_7m5U · 2022-08-08
> > > > **Thanks for the response!**
> > > >
> > > > Thank you for the response. I revised my score to accept. I am still not satisfied with the way that the authors are not acknowledging the limitations of the paper, but I agree with other reviewers that it passes the bar for acceptance. For example, regarding limitations: saying that there is an "extra" assumption does not equal to discussing limitations. Discussing limitations means showing when this extra assumption brings benefits and when it is not. Right now, the authors are doing a lot of the former but none of the latter.

---

> ### Author Response · Authors · 2022-07-30
> **Generarl Response**
>
> Thank you for your valuable comments. We have made a revision to our paper according to your suggestions. The revision is summarized as follows:
>
> 1) We give examples that satisfy our Assumptions in this paper, respectively under convex and non-convex loss functions. These examples are presented in Appendix F.
>
> 2) Fixing the typos you mentioned.
>
> 3) Rewriting some unclear sentences to make this paper more readable.

---

### Official Review · Reviewer_GY9U · 2022-07-12

**Rating:** 7
**Confidence:** 3
**Soundness:** 3 good
**Presentation:** 3 good
**Contribution:** 3 good

**Summary:**

This paper proposes upper bounds of the expected excess risk of locally strongly convex losses, removing global growth conditions.
Specifically, this paper bounds the generalization error and the optimization error to get better risk bounds.
For convex problems, the bound proposed by this paper is of the order $\mathcal{O}(1/n)$, better than previous $\mathcal{O}(1/\sqrt{n})$ ones.
For non-convex problems, this paper shows that for each population local minimum, there is an empirical local
minimum concentrated around it with high probability, and the bound proposed by this paper is of the order $\mathcal{O}(1/\sqrt{n})$, better than previous $\mathcal{O}(\sqrt{d}/\sqrt{n})$ ones.

**Questions:**

Please see the weaknesses part.

**Limitations:**

As I can see, this paper has no potential negative societal impact.

**Strengths And Weaknesses:**

Merits:
1. This paper generalizes previous global growth conditions to locally strongly convexity, which extends the application scenarios.
2. Under convex condition, the result given in this paper is better than [Hardt et al., 2016; Shamir and Zhang, 2009; Zhang et al., 2017].
3. Under non-convex condition, this paper proves that with high probability, there is an empirical local
minimum concentrated around each population local minimum, and the excess risk given in this paper is better than [Shamir and Zhang, 2009].

Cons:
1. To bound the stability, a key part is to bound the gap between models derived from adjacent datasets, however, by defining the 'good events', the proofs are similar to previous ones, which reduces the novelty.
2. Some references should be added, such as [Li and Liu 2022, High Probability Guarantees for Nonconvex Stochastic Gradient Descent with Heavy Tails], in which some better results are given.
3. Comparing with PL condition, the superiority of locally strongly convexity (or the relationship between them) should be given more detailed since some of examples given in this paper also satisfy the PL condition (e.g. robust regression).

---

> ### Author Response · Authors · 2022-07-30
> **To Reviewer GY9U**
>
> We thank you for your valuable comments, and we address your concerns as follows.
>
> Q1: “To bound the stability, a key part is to bound the gap between models derived from adjacent datasets, however, by defining the 'good events', the proofs are similar to previous ones, which reduces the novelty.”
>
> A1: Under the convex condition, our key technique is by defining the “good events” and controlling their probability. However, under non-convex condition, our key techniques include defining “good events”, proving “no extra local minima”, and constructing an auxiliary sequence in equation (13). To the best of our knowledge, it has never been seen in the existing literature to combine these three techniques in the proof.
>
> Q2: “Some references should be added, such as [Li and Liu 2022, High Probability Guarantees for Nonconvex Stochastic Gradient Descent with Heavy Tails], in which some better results are given.”
>
> A2: Thanks for pointing out this reference (Li and Liu 2022), we find that this paper is published in July 2022, which is after our submission in May 2022. Thus, we miss this reference. After reading this paper, we find they give a high-probability generalization bound of order $O(1 / n)$ for population risk satisfies PL-inequality, which has the same order of our result (without spurious local minima). They also require the number of data is larger than the order of dimension as we did in this paper. However, the key difference is their results are obtained under PL-inequality, while ours are under the weaker local strong convexity condition. We have added this reference in the revised version.
>
> Q3: “Comparing with PL condition, the superiority of locally strongly convexity (or the relationship between them) should be given more detailed since some of examples given in this paper also satisfy the PL condition (e.g. robust regression).”
>
> A3: The PL-inequality is a sufficient condition of no spurious local minima. The PL-inequality is violated when the loss function has saddle points, while the loss function that is locally strongly convex can have saddle points.
>
> To check it, let us see an example of the Gaussian mixture model. Consider the data $x_{i}$ drawn from a mixture of two Gaussians such that $x_{i}\sim \frac{1}{2}N(\theta_{1}^{\*}, I) +  \frac{1}{2}N(\theta_{2}^{\*}, I) = p_{\theta^{\*}}$. Then the population risk of maximizing the likelihood problem is $R(\theta_{1}, \theta_{2}) = -E_{x\sim p_{\theta^{*}}}[\log{p_{\theta}(x)}]$, which has two global minima $(\theta_{1}^{\*}, \theta_{2}^{\*}), (\theta_{2}^{\*}, \theta_{1}^{\*})$, and a saddle point $((\theta_{1}^{\*} + \theta_{2}^{\*}) / 2, (\theta_{1}^{\*} + \theta_{2}^{\*}) / 2)$. Thus, this problem violates the PL-inequality. However, we can compute the Hessian to check that the two global minima are all strict local minima, while the saddle point is a strict saddle point. Thus, the example satisfies our Assumption 4. More details of this example can be found in Lemma 16 in (Mei et al., 2018)., and our discussion in Appendix F.
>
> Ref: Mei et al., 2018, The Landscape of Empirical Risk for Non-convex Losses.
> Li and Liu, 2022,  High Probability Guarantees for Nonconvex Stochastic Gradient Descent with Heavy Tails.

---

> > ### Comment · Reviewer_GY9U · 2022-08-09
> > **Thanks for the response**
> >
> > Your response has fixed my concerns. I have no further questions.

---

### Official Review · Reviewer_HJpT · 2022-07-13

**Rating:** 7
**Confidence:** 3
**Soundness:** 4 excellent
**Presentation:** 3 good
**Contribution:** 3 good

**Summary:**

This paper studies the excess risk when the population loss is locally strongly convex. For convex problems, they obtain an upper bound of order $\tilde{O}(1/n)$ without assuming globally strong convex as in previous works, where $n$ is the number of training examples. For nonconvex problems, they obtain a bound of $\tilde{O}(1/\sqrt{n})$, which improves previous $\tilde{O}(\sqrt{d/n})$ result. Furthermore, assuming there are no spurious local minima with high probability, they can improve the bound to $\tilde{O}(1/n)$.

**Questions:**

My major questions and suggestions are in the weakness part, below are some additional comments:

1. The authors mentioned it is hard to improve their worse dependence on $\lambda$. Could they explain intuitively why it is hard?
2. Assumption 1 is used in previous works does not mean it is mild. I think it is kind of strong, but acceptable.

**Limitations:**

I do not think this paper has any potential negative societal impact as it is purely theoretical. Regarding the limitations of the results, please see the weakness and question parts.

**Strengths And Weaknesses:**

Strengths:
1. The theoretical results look rigorous and sound. Their results improve existing ones in several aspects.
2. The authors also clearly show how their theorems are proved with good intuition.
3. The analysis looks novel to me.

Weaknesses:
1. For convex problems, although I can see that locally strongly convex + globally convex is more general than globally strongly convex, this improvement looks a bit incremental. Also, they have a worse dependence on the local convexity parameter $\lambda$. The authors might want to clarify more why it is more challenging. But it might be fine if the focus of this paper is the nonconvex part.
2. For nonconvex problems, although the result does not depend on the dimension $d$, it depends on $K$, the number of local minima. However, if only considering Proposition 1, $K$ could be exponential in the dimension in the worst case, which means the result is better than previous ones only when $K$ is much smaller than $d$.
3. The theoretical results look a bit dense. I am fine with it but it is still better if the authors could make it more readable. For example, they could hide the constant factors. Theorem 3 also looks not necessary because it is just a direct corollary of theorem 2.

---

> ### Author Response · Authors · 2022-07-30
> **To Reviewer HJpT**
>
> We thank you for your valuable comments, and we address your concerns as follows.
>
> Q1: “For convex problems, although I can see that locally strongly convex + globally convex is more general than globally strongly convex, this improvement looks a bit incremental”
>
> A1: As you said, our condition “locally strongly convex + globally convex” is more general than the globally strongly convex condition. Besides that, please note that our globally strongly convexity assumption is imposed on the expected risk $E_{z}[f(w, z)]$ instead of the instantaneous loss (i.e., $f(w, z)$) as in the existing literature, e.g., (Hardt et al., 2016). This difference also makes our condition weaker. For example, consider a linear regression problem such that $z = (x, y)$, $y = x^{\top}w^{*} + \epsilon$ for independent noise $\epsilon$, and $f(w, z) = (y - w^{\top}x)^{2}$. Obviously, when the Hessian of population risk $E[xx^{\top}]$ is positively definite, the population risk is local (global) strongly convex, and all of the assumptions in our paper are satisfied. However, for any instantaneous loss $f(w, z)$ has Hessian of $xx^{\top}$ which means $f(w, z)$ is not necessarily strongly convex with respect to $w$ for any $z$. Thus, we can only treat it as a convex loss function when applying the technique in (Hardt et al., 2016), and get the excess risk bound of order $O(\sqrt{1/n})$. However, the empirical minimizer has an excess risk of order $O(1/n)$ which matches our result. By the way, the technique in (Zhang et al., 2017) also can be applied here, while they require the number of data is sufficiently large, and we do not have such a requirement.
>
> Q2:” Also, they have a worse dependence on the local convexity parameter”
>
> A2: As we clarified in 172, our bound is not necessarily looser than the global strongly convex based results, as the $\lambda$ in our bound is the local strongly convex parameter around the global minimum, instead of the global strongly convex parameter. However, in the worst case, our bound is looser (but with weak conditions) than the existing ones. We have made efforts to improve the dependence of $\lambda$. However, under our technique, it seems hard to do this without sacrificing the order of $n$. More details are in A3.
>
> Q3: “The authors mentioned it is hard to improve their worse dependence on $\lambda$. Could they explain intuitively why it is hard?”
>
> A3: If you check our proofs in Appendix B.1, you can find our bound involves a term that estimates the probability of “good events”, i.e., equation (29). The probability is estimated via Markov’s inequality in equation (39). The dependence of $\lambda$ can be weakened to $1/\lambda^{2}$ via applying Markov’s inequality to $||\nabla R_{S}(w^{*})||$  instead of its square as we did in this paper. However, if doing so, the probability bound will be of order $O(\sqrt{1/\lambda^{2}n})$ which has a worse dependence on $n$.
>
> Q4: “For nonconvex problems, although the result does not depend on the dimension $d$, it depends on $K$, the number of local minima. However, if only considering Proposition 1, $K$ could be exponential in the dimension in the worst case, which means the result is better than previous ones only when $K$ is much smaller than $d$.”
>
> A4: Yes, you are correct, our bound has a linear dependence on the number of local minima $K$. Though in a very special case, i.e., a special single neuron neural network (Auer et al., 1996), the loss function has an exponential number of local minima. The result is obtained under a very special constructed training set. Thus, we consider such a special case is somehow impractical, as many of the important non-convex problems have small $K$ as we clarified in line 209. On the other hand, the dependence on $K$ can be removed, when imposing some implicit regularization conditions e.g., the obtained iterates converge to the empirical local minimum around a specific population local minimum (e.g., max-margin solution) (Lyu et al., 2020).
>
> Q5: “The theoretical results look a bit dense. I am fine with it but it is still better if the authors could make it more readable. For example, they could hide the constant factors.”
>
> A5: Thanks for pointing out this, although we have clarified the order of our bounds after presenting them, we find your suggestion makes our paper more readable. We have made a revision according to your suggestion in the revised version.
>
> Ref:
> Auer et al., 1996 Exponentially many local minima for single neurons.
> Lyu et al., 2020 Gradient Descent Maximizes the Margin of Homogeneous Neural Networks.

---

> > ### Comment · Reviewer_HJpT · 2022-08-08
> > **Thank you for your detailed explanations**
> >
> > Thank you for the detailed explanations. I think the comments well address my concerns about the contribution of this paper. The results in both convex and non-convex parts seem stronger than I thought. So I would like to increase the contribution score from 2 to 3 and the overall rating from 6 to 7.

---

### Author Response · Authors · 2022-07-30
**General Response**

We thank reviewers for their valuable comments. We have submitted a revised version according to their comments. All of the revised parts are marked as blue. Details are summarized as follows:

1) According to the comments of Reviewer 7m5U. We have added three examples in Appendix E which satisfy our Assumptions imposed in this paper.

2) We have fixed some typos and unclear sentences according to reviewers’ comments.

---

### Meta-Review · Area_Chair_XvwL · 2022-08-26

**Recommendation:** Accept
**Confidence:** Certain

**Metareview:**

There is a consensus among reviewers that the results are strong with novel results on the rate of convergence of the excess risk of certain iterative algorithms for convex and non-convex problems, under a local strong convexity assumption.

For the camera-ready version, the authors are encouraged to provide a more detailed discussion of the limitation of the work as emphasized in the discussion with reviewer 7m5U.

**Award:**

No

---

### Decision · Program_Chairs · 2022-09-14

Accept